Osteology of Galeamopus pabsti sp. nov. (Sauropoda: Diplodocidae), with implications for neurocentral closure timing, and the cervico-dorsal transition in diplodocids

Tschopp Emanuel tschopp.e@gmail.com 1 2 3
Mateus Octávio 2 4
1 Dipartimento di Scienze della Terra, Università di Torino , Torino , Italy
2 GeoBioTec, Faculdade de Ciencia e Tecnologia, Universidade Nova de Lisboa , Caparica , Portugal
3 Museu da Lourinhã , Lourinhã , Portugal
4 GeoBioTec + Departamento de Ciências da Terra, Faculdade de Ciencia e Tecnologia, Universidade Nova de Lisboa , Caparica , Portugal
Farke Andrew
Electronic publication date: 2017 May 2
Publication date: 2017
Volume: 5
Electronic Location ID: e3179
Received 2016 May 6; Accepted 2017 Mar 14
Copyright: ©2017 Tschopp and Mateus
Copyright year: 2017
Copyright holder: Tschopp and Mateus
License: This is an open access article distributed under the terms of the Creative Commons Attribution License, which permits unrestricted use, distribution, reproduction and adaptation in any medium and for any purpose provided that it is properly attributed. For attribution, the original author(s), title, publication source (PeerJ) and either DOI or URL of the article must be cited.
License URL: https://creativecommons.org/licenses/by/4.0/

Keywords: Dinosauria, Sauropoda, Morrison Formation, Diplodocinae, Ontogeny, New species, Late Jurassic, Howe Ranch

Funding: Fundac˛ão para a Ciência e a Tecnologia of the Ministério de Educação e Ciência (FCT-MEC) SFRH/BD/66209/2009 Synthesys DE-TAF-1150 Volkswagen Stiftung Europasaurus Project European Union’s Seventh Framework programme Marie Skłodowska-Curie grant 609402 - 2020 researchers: Train to Move (T2M) This research was funded through the doctoral fellowship SFRH/BD/66209/2009 awarded by the the Fundação para a Ciência e a Tecnologia of the Ministério de Educação e Ciência (FCT-MEC), Portugal, to E Tschopp. External funding for collection visits for comparative purposes were at least partially financed through a short course grant from FCT-MEC (for a visit of the HMNS), a Synthesys grant (DE-TAF-1150) (for a visit at MB.R.), and a Volkswagen Stiftung grant as part of the “Europasaurus Project” to E Tschopp. E Tschopp is currently funded by the European Union’s Seventh Framework programme for research and innovation under the Marie Skłodowska-Curie grant agreement No. 609402 - 2020 researchers: Train to Move (T2M). There was no additional external funding received for this study. The funders had no role in study design, data collection and analysis, decision to publish, or preparation of the manuscript.

==============================
Diplodocids are among the best known sauropod dinosaurs. Numerous specimens of currently 15 accepted species belonging to ten genera have been reported from the Late Jurassic to Early Cretaceous of North and South America, Europe, and Africa. The highest diversity is known from the Upper Jurassic Morrison Formation of the western United States: a recent review recognized 12 valid, named species, and possibly three additional, yet unnamed ones. One of these is herein described in detail and referred to the genus Galeamopus. The holotype specimen of Galeamopus pabsti sp. nov., SMA 0011, is represented by material from all body parts but the tail, and was found at the Howe-Scott Quarry in the northern Bighorn Basin in Wyoming, USA. Autapomorphic features of the new species include a horizontal canal on the maxilla that connects the posterior margin of the preantorbital and the ventral margin of the antorbital fenestrae, a vertical midline groove marking the sagittal nuchal crest, the presence of a large foramen connecting the postzygapophyseal centrodiapophyseal fossa and the spinopostzygapophyseal fossa of mid- and posterior cervical vertebrae, a very robust humerus, a laterally placed, rugose tubercle on the concave proximal portion of the anterior surface of the humerus, a relatively stout radius, the absence of a distinct ambiens process on the pubis, and a distinctly concave posteroventral margin of the ascending process of the astragalus. In addition to the holotype specimen SMA 0011, the skull USNM 2673 can also be referred to Galeamopus pabsti. Histology shows that the type specimen SMA 0011 is sexually mature, although neurocentral closure was not completed at the time of death. Because SMA 0011 has highly pneumatized cervical vertebrae, the development of the lamination appears a more important indicator for individual age than neurocentral fusion patterns. SMA 0011 is one of very few sauropod specimens that preserves the cervico-dorsal transition in both vertebrae and ribs. The association of ribs with their respective vertebrae shows that the transition between cervical and dorsal vertebrae is significantly different in Galeamopus pabsti than in Diplodocus carnegii or Apatosaurus louisae, being represented by a considerable shortening of the centra from the last cervical to the first dorsal vertebra. Diplodocids show a surprisingly high diversity in the Morrison Formation. This can possibly be explained by a combination of geographical and temporal segregation, and niche partitioning.

Introduction

Diplodocidae is one of the best known groups of sauropod dinosaurs. The anatomy and relationships of its members are well studied (e.g., Osborn, 1899; Hatcher, 1901; Holland, 1924; Gilmore, 1932; Gilmore, 1936; McIntosh & Berman, 1975; Berman & McIntosh, 1978; Gillette, 1991; Upchurch, Tomida & Barrett, 2004; McIntosh, 2005; Whitlock, 2011a; Mannion et al., 2012; Tschopp & Mateus, 2013b; Gallina et al., 2014; Tschopp, Mateus & Benson, 2015). Diplodocidae is subdivided into the two subgroups Apatosaurinae and Diplodocinae. Apatosaurinae includes the genera Apatosaurus and Brontosaurus, whereas diplodocines are more diverse (Tschopp, Mateus & Benson, 2015). The earliest confirmed report of a diplodocine occurs in the Oxfordian (Late Jurassic) of Georgia (Gabunia et al., 1998; Mannion et al., 2012). In the Kimmeridgian and Tithonian, diplodocids reached their highest diversity, and are known from deposits across the Western United States, Tanzania, Portugal, Spain, Argentina, Chile, and possibly Zimbabwe and England (Mannion et al., 2012; Rauhut, Carballido & Pol, 2015; Salgado et al., 2015; Tschopp, Mateus & Benson, 2015). The most recent occurrence is from the late Berriasian to early Valanginian of Argentina (Whitlock, D’Emic & Wilson, 2011; Gallina et al., 2014; Tschopp, Mateus & Benson, 2015).

The Upper Jurassic Morrison Formation of the western USA yielded the highest diversity of diplodocid sauropods worldwide. Although it has been studied since the 1870s, which led to the first descriptions of diplodocid sauropods (Amphicoelias Cope, 1877; Apatosaurus Marsh, 1877, Diplodocus Marsh, 1878; Brontosaurus Marsh, 1879), new species have continued to be discovered in the Morrison Formation until the present (Kaatedocus siberi; Tschopp & Mateus, 2013b). Recently, an extensive phylogenetic analysis of the clade Diplodocidae at the specimen-level recognized yet another genus, typified by a species previously included in Diplodocus: “D. ” hayi was found as the sister taxon to Diplodocus and more derived diplodocines by Tschopp, Mateus & Benson (2015), who created the new genus Galeamopus for the species, and referred three more specimens to the same genus, but not necessarily the same species: AMNH 969 (a skull, atlas and axis previously identified as Diplodocus), SMA 0011 (a semi-articulated skeleton including cranial, axial, and appendicular elements), and USNM 2673 (a partial skull previously referred to Diplodocus as well, and used as the basis for the skull attached to the mounted skeleton of the Diplodocus carnegii holotype CM 84; McIntosh, 1981). Here, we provide a detailed description of the specimen SMA 0011, thereby also illuminating the osteology of the genus Galeamopus. We show that differences between SMA 0011 and the holotype of Galeamopus hayi (HMNS 175) are numerous, thus supporting the claims of Tschopp, Mateus & Benson (2015) that SMA 0011 represents a second species within Galeamopus, which will be named G. pabsti sp. nov.

Howe Ranch: a rediscovered diplodocid El Dorado

The specimen SMA 0011 was found at the Howe-Scott Quarry on the Howe Ranch. The several sites on the ranch have produced a high number of partially to almost completely articulated dinosaur skeletons, sometimes even with soft tissue preservation (see Brinkmann & Siber, 1992; Ayer, 2000; Schwarz et al., 2007; Siber & Möckli, 2009; Christiansen & Tschopp, 2010; Tschopp & Mateus, 2013b; Tschopp et al., 2015). Three sites have proved particularly productive: the Howe Quarry, the Howe-Stephens Quarry, and the Howe-Scott Quarry (Fig. 1). The Howe Quarry was first worked by Barnum Brown for the American Museum of Natural History (New York, USA) in 1934, and was later relocated and completely excavated by a team from the Sauriermuseum Aathal (Switzerland), led by Hans-Jakob ‘Kirby’ Siber (Brown, 1935; Ayer, 2000; Michelis, 2004; Tschopp & Mateus, 2013b). The other two sites, as well as several smaller, less productive spots at various stratigraphic levels within the Morrison Formation, have since been discovered nearby and excavated by the SMA (Ayer, 2000; Siber & Möckli, 2009; Christiansen & Tschopp, 2010; Fig. 2). All three major sites yielded well-preserved and at least partially articulated diplodocid specimens of varying ontogenetic stages. Only one of these specimens has yet been formally described (even including the AMNH material from 1934), and now constitutes the holotype of Kaatedocus siberi (Tschopp & Mateus, 2013b). Herein, we provide the detailed description of a second diplodocid specimen from this locality.

Figure 1 Locality of the Howe Ranch.

The Ranch is situated in the vicinity of Shell, Wyoming (B, star), with a detailed map of the three most important sites on the Ranch (C). (B) is modified from Christiansen & Tschopp, 2010, (C) copyright by the Sauriermuseum Aathal, modified with permission.

Figure 2 Stratigraphy of the Morrison Formation at Howe Ranch.

The levels of the three most important quarries on the Howe Ranch. The red line marks the clay change which has been proposed as marker bed to correlate sites across the Morrison Formation. Copyright by Jacques Ayer (2005), modified with permission.

Material

Locality

The Howe-Scott Quarry, where SMA 0011 was found, is located between the better known Howe Quarry (Brown, 1935; Ayer, 2000; Michelis, 2004; Tschopp & Mateus, 2013b) and the Howe-Stephens Quarry (Ayer, 2000; Schwarz et al., 2007; Christiansen & Tschopp, 2010; Fig. 1). The site was found in 1995 by a team from the Sauriermuseum Aathal, Switzerland, and excavated in three periods (1995, 2000, 2002–2003). Stratigraphically, it lies just slightly above the Howe-Stephens Quarry, 30 m above the J-5, and 30 m below the K-1 unconformities, which define the lower and upper limits of the Morrison Formation, respectively (Michelis, 2004; Fig. 2). In addition to SMA 0011, five partial diplodocid specimens (mostly appendicular material), a possible brachiosaur hindlimb, two partly-to-almost complete Hesperosaurus (Ornithischia, Stegosauria), some Othnielosaurus bones (Ornithischia, Neornithischia), numerous shed theropod teeth, carbonized wood, and various freshwater shells were recovered at the Howe-Scott Quarry (Michelis, 2004; E Tschopp, pers. obs., 2003). However, none of these specimens has yet been formally described.

Specimen

The specimen SMA 0011 (nicknamed “MaX”) consists of an almost complete disarticulated skull, 13 cervical vertebrae (probably CV 1–10, and the three posterior-most cervical vertebrae, see below), dorsal vertebrae 1–2 and the last six presacral vertebrae (possibly DV 5–10), several cervical, dorsal, and sternal ribs, a partial sacrum, both scapulae and coracoids, both humeri, the left ulna, radius and manus, the right ilium, both pubes, the left proximal ischium, the left femur, tibia, fibula and nearly complete pes. The specimen was found in two parts: (1) skull and vertebral column from the atlas to DV 2, and (2) 6 dorsal vertebrae, sacrum, and appendicular elements (Fig. 3). It is interpreted to belong to a single individual due to matching size, no overlap of elements, and an extremely similar pattern of neurocentral closure in cervical and dorsal vertebrae (see below). Other elements found close to the bones belonging to the holotype can be excluded from the individual due to significant size differences and doubling of elements.

Figure 3 Quarry map of SMA 0011.

Note the separation of the cervical series and the skull from the dorsal column and the appendicular skeleton, and the articulated block of dorsal vertebrae that do not belong to SMA 0011 (see arrowhead between horizontal lines 15 and 16). Abb.: bc, braincase; co, coracoid; CR, cervical rib; CV, cervical vertebra; DR, dorsal ribs; DV, dorsal vertebra; fe, femur; fi, fibula; fl, forelimb; h, humerus; hl, hindlimb; il, ilium; is, ischium; ma, manus; pcg, pectoral girdle; pe, pes; pu, pubis; pvg, pelvic girdle; r, radius; sc, scapula; SR, sternal ribs; SV, sacral vertebrae; ti, tibia; u, ulna. Map drawn by Esther Premru, copyright by Sauriermuseum Aathal, modified with permission.

Systematic Paleontology

Dinosauria Owen, 1842	
Sauropoda Marsh, 1878	
Eusauropoda Upchurch, 1995	
Neosauropoda Bonaparte, 1986	
Diplodocoidea Marsh, 1884	
Flagellicaudata Harris & Dodson, 2004	
Diplodocidae Marsh, 1884	
Diplodocinae Marsh, 1884	
GaleamopusTschopp, Mateus & Benson, 2015	

Type species. Diplodocus hayi Holland, 1924

Revised diagnosis. Galeamopus is a diplodocid sauropod that can be diagnosed by nine autapomorphies. The phylogenetic analysis (see below) recovered three autapomorphies that were not shared with other diplodocine specimens: (1) the interpostzygapophyseal lamina of mid- and posterior cervical neural arches does not project beyond the posterior margin of the neural arch (unique among Diplodocinae; already proposed by Tschopp, Mateus & Benson, 2015); (2) an approximately right angle formed by the ventral margin of the preacetabular lobe of the ilium and the pubic peduncle (unique among Diplodocinae); and (3) the lateral edge of the proximal end of the tibia forms a pinched out projection, posterior to the cnemial crest (unique among Diplodocidae; proposed as diagnostic for the species G. hayi by Tschopp, Mateus & Benson, 2015, but see below).

Additional autapomorphies that could not be found directly by the analysis due to the lack of anatomical overlap with the sister clade in crucial skeletal regions as for instance the skull, or because of the absence of characters coding for that particular feature, include the following (most of them were already identified by Tschopp, Mateus & Benson, 2015): (4) teeth with paired wear facets (unique among Flagellicaudata; Tschopp, Mateus & Benson, 2015); (5) well-developed anteromedial processes on the atlantal neurapophyses, which are distinct from the posterior wing (unique among Diplodocoidea; Tschopp, Mateus & Benson, 2015); (6) the atlantal neural arch bears a small subtriangular, laterally projecting spur at its base (unique among Diplodocidae; Tschopp, Mateus & Benson, 2015); (7) the posterior wing of atlantal neurapophyses remains of subequal width along most of its length (unique among Diplodocidae; Tschopp, Mateus & Benson, 2015); (8) the axial prespinal lamina develops a transversely expanded, knob-like tuberosity at its anteroventral extremity (unambiguous; Tschopp, Mateus & Benson, 2015); and (9) the loss of strong opisthocoely between dorsal centra 1 and 2 (unique among Diplodocidae).

Galeamopus hayiHolland, 1924	

Holotype. HMNS 175 (formerly CM 662).

Revised diagnosis. Some of the autapomorphies of the species Galeamopus hayi proposed by Tschopp, Mateus & Benson (2015) are actually also present in the second species named below (and were thus moved to the generic diagnosis), and some new apomorphic features were recognized during the present study (see ‘Discussion’). Autapomorphies recovered by the phylogenetic analysis but shared with other diplodocine specimens are not considered valid here. The revised list of autapomorphies of G. hayi includes the following four autapomorphies: (1) dorsoventral height of the parietal occipital process is low, subequal to less than the diameter of the foramen magnum (unique among Diplodocinae; Tschopp, Mateus & Benson, 2015); (2) an ulna to humerus length of more than 0.76 (unique within Diplodocoidea; Tschopp, Mateus & Benson, 2015); (3) distal articular surface for the ulna on the radius is reduced and relatively smooth (unique within Diplodocidae; Tschopp, Mateus & Benson, 2015); (4) a rhomboid outline of the proximal articular surface of metatarsal V (unique within Diplodocinae).

Referred specimen. AMNH 969, a nearly complete skull and articulated atlas and axis.

Locality and horizon. Galeamopus hayi is known from two quarries in the Upper Jurassic Morrison Formation of Wyoming: the Red Fork of the Powder River, Johnson County, (HMNS 175) on the eastern slopes of the Bighorn mountains, and the Bone Cabin Quarry in Albany County (AMNH 969).

Galeamopus pabsti sp. nov.	
Tschopp, Mateus & Benson (2015), figs. 1E, 2B, 3D, 7G, 36, 41B, 44B, 46C, 49B, 50B, 69B, 93A; Figs. 4–77.	

Holotype. SMA 0011: partial skull, 13 cervical vertebrae, 8 dorsal vertebrae, partial sacrum, cervical, dorsal, and sternal ribs, both scapulae and coracoids, both humeri, left ulna, radius, and manus (including one carpal element), right ilium and pubis, left ischium, left femur, tibia, fibula, astragalus, and pes.

Diagnosis. Galeamopus pabsti can be diagnosed by the following 14 autapomorphies: (1) horizontal canal connecting the posterior margin of the preantorbital and the ventral margin of the antorbital fenestra laterally on the maxilla (unambiguous); (2) the sagittal nuchal crest on the supraoccipital is marked by a vertical midline groove (unique among non-somphospondylian sauropods); (3) anterior cervical vertebrae are much higher than wide (>1.2; unique among Diplodocinae); (4) the posterior centrodiapophyseal and the postzygodiapophyseal laminae of mid- and posterior cervical vertebrae do not meet anteriorly at the base of the transverse process (unique among Diplodocinae); (5) mid- and posterior cervical vertebrae with a large opening connecting the postzygapophyseal centrodiapophyseal fossa and the spinopostzygapophyseal fossa (unambiguous); (6) a low EI of posterior cervical centra (<2.0; unique among Diplodocinae); (7) a low acromion height to scapular length ratio (<0.46; unique among Flagellicaudata); (8) a robust humerus (RI >  0.33; unique among Diplodocinae); (9) the lateral displacement of the distinct rugose tubercle on the concave proximal portion of the anterior surface of the humerus (unique within Diplodocidae); (10) the maximum diameter of the proximal end of the radius divided by its greatest length is 0.3 or greater (unique among Diplodocinae); (11) the longest metacarpal is at least 0.4 times the length of the radius (unique among Diplodocinae); (12) the proximal articular surface of metacarpal V is significantly larger than the surfaces of metacarpals III and IV (unique among Diplodocidae); (13) a subrectangular proximal articular surface of the tibia (unique among Diplodocinae); and (14) the ascending process of the astragalus has a concave posteroventral margin, resulting in the presence of two distinct, rounded posterior processes in ventral view (unique among Diplodocoidea).

Etymology. The species name “pabsti” honors the finder of the holotype specimen, Dr. Ben Pabst (born in Vienna, Austria, in January 26, 1949), who also created the skull reconstruction and led the repreparation of the specimen and its mount at SMA. Pabst has led several paleontological excavations in Switzerland and the USA, and is highly skilled in fossil preparation and skeleton mounting.

Referred specimens. USNM 2673, a partial skull.

Locality and horizon. Galeamopus pabsti is known from two quarries in the Upper Jurassic Morrison Formation of Wyoming and Colorado: the Howe-Scott Quarry (SMA 0011) on the western slopes of the Bighorn mountains, and Felch Quarry 1 near Garden Park, Fremont County, in Colorado (USNM 2673). Felch Quarry 1 has been dated to 152.29 ± 0.27 (Trujillo & Kowallis, 2015).

Comments. The holotype specimen SMA 0011 is housed at Sauriermuseum Aathal, Switzerland. This museum is open to the public, and specimens are available for study by researchers (see Schwarz et al., 2007; Klein & Sander, 2008; Christiansen & Tschopp, 2010; Carballido et al., 2012; Klein, Christian & Sander, 2012; Tschopp & Mateus, 2013a; Tschopp & Mateus, 2013b; Foth et al., 2015; Tschopp, Mateus & Benson, 2015). The excavations are very well documented, and the preparation of the material follows the latest scientific standards. The museum recognizes the scientific importance of holotype specimens, and takes all efforts to preserve them and provide permanent public access. The policy is publicly stated on their homepage (http://www.sauriermuseum.ch/de/museum/wissenschaft/wissenschaft.html). These efforts were recently acknowledged by the University of Zurich, Switzerland, through the attribution of a Dr. honoris causa to the founder and director of the Sauriermuseum Aathal, Hans-Jakob Siber.

The specimen itself is currently on display as a mounted skeleton. Completely prepared elements that are difficult to access in the mount were molded, and high-quality casts are stored in the SMA collections. A detailed account of the excavation, preparation, documentation, and mount will be published elsewhere.

The electronic version of this article in Portable Document Format (PDF) will represent a published work according to the International Commission on Zoological Nomenclature (ICZN), and hence the new names contained in the electronic version are effectively published under that Code from the electronic edition alone. This published work and the nomenclatural acts it contains have been registered in ZooBank, the online registration system for the ICZN. The ZooBank LSIDs (Life Science Identifiers) can be resolved and the associated information viewed through any standard web browser by appending the LSID to the prefix http://zoobank.org/. The LSID for this publication is: urn:lsid:zoobank.org:pub:93B626A1-BF8E-4865-A76E-551EE78C9D92. The online version of this work is archived and available from the following digital repositories: PeerJ, PubMed Central and CLOCKSS.

Description of SMA 0011

Terminology. Anatomical terms used here follow the traditional use of anterior and posterior instead of cranial and caudal (Wilson, 2006). Directional terms in the skull descriptions are used in relation to a horizontally oriented tooth-bearing edge of the maxilla. Terminology for axial and appendicular elements is explained in further detail below, given the extensive descriptive subsections.

Table 1 Skull measurements of Galeamopus pabsti SMA 0011 (in mm, asterisks indicates estimated dimensions).

Element	apL	aprL	vL	tbL	ocL	pprL	min apW	max apW	aW	minW	maxW	pW	dW	nW	minH	maxH	apH	
Premaxilla	R	325								47	13		46						
	L	320								47	12		45						
Maxilla	R	354		225	120		246										210		
Preantorbital fossa	R	73																	
Preantorbital fenestra	R	45																	
Prefrontal	R	63				53						37							
	L	67				43						34							
Frontal	R	73				41					69	90							
	L	74				44					68	84							
Postorbital	R	139								20		41					63	10	
	L	116								24		42					68	11	
Jugal	R	121					58										81		
	L	133					68										93		
Quadratojugal	R	182	154														59		
	L	149	106														51		
Lacrimal	L	62						12			13	15					10*	56	
Quadrate	R	148																	
Squamosal	R	>40										23					63		
	L	>59										22					60		
Parietal	R						69	3	19										
	L						62	6	15										
Supraoccipital	–													28			59		
Exoccipital– opisthotic complex	–											150							
Paroccipital process	R															23			
	L															23			
Occipital condyle	–	29									39	42							
Foramen magnum	–											27					17		
Posttemporal fenestra	R											23							
	L											24							
Basioccipital	–										30								
Basal tubera	–											42			8				
Basipterygoid process	R	64								16	11	17							
	L	def								19	10	16							
Orbitosphenoid	R												36						
	L												38						
Laterosphenoid	R																		
Prootic	R																		
	L																		
Dentary	R															22	62		
	L															37	67		
Surangular	R															44			
	L															41			
Angular	R	180																	
	L	170																	
Element	dH	vrH	dlpH	dmpH	apL paof	apL paofe	aopL	osrL	pp-fp	
Premaxilla	R										
	L										
Maxilla	R					73	45				
Preantorbital fossa	R										
Preantorbital fenestra	R										
Prefrontal	R									37	
	L									37	
Frontal	R										
	L										
Postorbital	R				36						
	L				36						
Jugal	R										
	L										
Quadratojugal	R										
	L										
Lacrimal	L										
Quadrate	R		29								
Squamosal	R										
	L										
Parietal	R			39							
	L			41							
Supraoccipital	–										
Exoccipital-opisthotic complex	–										
Paroccipital process	R	inc									
	L	34									
Occipital condyle	–										
Foramen magnum	–										
Posttemporal fenestra	R										
	L										
Basioccipital	–										
Basal tubera	–										
Basipterygoid process	R										
	L										
Orbitosphenoid	R										
	L										
Laterosphenoid	R							30			
Prootic	R								42		
	L								42		
Dentary	R										
	L										
Surangular	R										
	L										
Angular	R										
	L										
Notes.

Measurement protocols: maxilla pprL distance posterior process to anterior margin

vL measured along curvature

maxH distance posteroventral corner to posterodorsal corner

postorbital apH measured at lateral edge

jugal pprL measured from anteriormost point of orbit to posteriormost extension of jugal

quadratojugal maxH length dorsal process

lacrimal apL measured at dorsal end

quadrate apL distance from posterior-most point of shaft to anterior-most point of ventral ramus

squamosal anterior processes incomplete

parietal dlpH measured at base of process

pprL measure along dorsal edge of posterolateral process

min & max apW measured dorsally

supraoccipital maxH distance foramen magnum-parietal suture

exoccipital-opisthotic complex maxW measured across paroccipital processes

occipital condyle minW measured at neck

basal tubera maxW across paired tubera

basipterygoid process aW measured at distal tip

maxW measured at base, dorsoventrally

orbitosphenoid pW measured posterodorsally

dentary maxH at symphysis

Abb aopL length antotic process

apH dorsoventral height anterior process

apL anteroposterior length

apL paofe anteroposterior length preantorbital fenestra

apL paof anteroposterior length preantorbital fossa

aprL length anterior process

aW anterior width

def deformed

dH distal dorsoventral height

dlpH dorsoventral height dorsolateral process

dmpH dorsoventral height dorsomedial process

dW dorsal width

inc incomplete

max apW maximum anteroposterior width

maxH maximum dorsoventral height

maxW maximum transverse width

min apW minimum anteroposterior width

minH minimum dorsoventral height

minW minimum transverse width

nW width notch

ocL lateral length contributing to orbit

osrL length otosphenoidal ridge

pp-fp distance posterior process to frontoparietal suture

pprL length posterior process

pW posterior width

tbL length tooth-bearing portion

vL length ventral edge

vrH dorsoventral length ventral ramus

Cranial skeleton

Skull (Figs. 4–16; Table 1)

Preservation. The skull of Galeamopus pabsti SMA 0011 is nearly complete. The only bones lacking are the left maxilla and quadrate, the right lacrimal, and the bones from the palate with the exception of the right pterygoid. The skull has a typically diplodocid shape. It is elongate, with the external nares retracted and dorsally facing, and has slender, peg-like teeth (Figs. 4–7). Given the completeness of the skull, a reconstruction was created in cooperation with the Portuguese illustrator Simão Mateus (Fig. 7; Mateus & Tschopp, 2017). When compared with recent reconstructions of the skull of Diplodocus (Wilson & Sereno, 1998; Whitlock, 2011b), Galeamopus has a more triangular skull outline in lateral view, and more sinuous ventral maxillary edges in dorsal or ventral view (Fig. 7).

Figure 4 Skull bones of Galeamopus pabsti SMA 0011 before mounting.

Gray elements were lacking and reconstructed for the mounted skull. Abb.: an, angular; aof, antorbital fenestra; d, dentary; f, frontal; j, jugal; la, lacrimal; m, maxilla; na, nasal; oc, occipital condyle; p, parietal; pf, prefrontal; pm, premaxilla; popr, paroccipital process; pra, proatlas; q, quadrate; qj, quadratojugal; sa, surangular; so, supraoccipital; sq, squamosal; t, teeth. Scale bar = 10 cm. Photo by Urs Möckli and copyright by Sauriermuseum Aathal, modified with permission.

Premaxilla. The premaxillae are completely preserved. They are anteroposteriorly long and transversely narrow elements (Table 1) that contact each other medially and the maxillae laterally (Figs. 4–7). The posterior end of the premaxillae delimits the nasal opening anteriorly. In dorsal view, the elements are narrow in their central part and widen anteriorly and posteriorly. The anterior edge is straight to slightly convex, whereas the posterior margin is deeply concave, such that the two premaxillae together form a triangular process that enters the nasal opening anteromedially. The medial margin is straight, and the lateral one concave due to the central narrowing of the element. Some nutrient foramina are present on the anterior-most portion of the dorsal surface, as is a groove originating at the premaxillary-maxillary contact, and extending obliquely anteromedially (Figs. 5 and 7). The groove is faint and relatively short, not reaching either the anterior or the medial margin. Such a groove was usually interpreted as typical for dicraeosaurids (Remes, 2009; Whitlock, 2011a), but is also present in other diplodocids (Tschopp, Mateus & Benson, 2015). However, a fading out of this feature is uncommon in dicraeosaurids, where the groove is distinct (Janensch, 1935; Remes, 2009). Ventrally, the anterior portion of the premaxillae thickens slightly dorsoventrally in order to bear the replacement teeth, but not to the extent seen in the referred specimen USNM 2673 (Tschopp, Mateus & Benson, 2015). Five teeth are included in the mounted skull, but only four alveoli occur in the left element, whereas the right premaxilla appears to show five. The alveoli of the articulated premaxillae do not contact each other medially, such that there would be space for two more teeth in between, or a gap. The number of replacement teeth could not be discerned without a CT-scan. At the border with the maxilla, where the premaxilla narrows from the broader anterior part to the narrow central part, the two bones form an elongated fossa, which bears the subnarial and the anterior maxillary foramen. Both foramina lie on the medial edge of the maxilla, very close together.

Figure 5 Skull of Galeamopus pabsti SMA 0011 as usually figured.

The skull is shown as usually figured in dorsal (A), posterior (B), right lateral (C), and anterior views (D), following our terminology section. Dark, uniformely colored elements were lacking and reconstructed for the mounted skull. Note the shallow groove on the premaxilla, extending from the lateral margin anteromedially (1). Abb.: an, angular; aof, antorbital fenestra; bo, basioccipital; bpr, basipterygoid process; d, dentary; ex, exoccipital; f, frontal; j, jugal; ltf, laterotemporal fenestra; m, maxilla; n, external nares; na, nasal; o, orbit; os, orbitosphenoid; p, parietal; paof, preantorbital fossa; pf, prefrontal; pm, premaxilla; po, postorbital; popr, paroccipital process; pro, prootic; q, quadrate; qj, quadratojugal; sa, surangular; so, supraoccipital; sq, squamosal; stf, supratemporal fenestra. Scale bar =10 cm.

Maxilla. Only the right maxilla is preserved, and it is complete. The broad anterior portion bears a posterior process, which contacts the jugal and quadratojugal, and a posterodorsal process, which contacts the lacrimal, nasal, and the prefrontal (Figs. 4, 5, 7). The maxilla forms the dorsal, anterior, and anteroventral margins of the antorbital fenestra, and completely encloses the preantorbital fossa. Unlike Kaatedocus and Dicraeosaurus, the preantorbital fossa is pierced by a large fenestra. The fenestra is dorsally capped by a distinct ridge similar to Diplodocus, but unlike Apatosaurus. This distinct dorsal edge was previously thought to represent an autapomorphy of Diplodocus, but was shown to occur in other taxa as well (Tschopp & Mateus, 2013b). The preantorbital fenestra does not fill the entire preantorbital fossa (Table 1): the anterior-most area remains closed by a thin bony wall. The fossa is anterodorsally accompanied by a short, narrow groove more or less following the curvature of the anterior end of the dorsal rim of the fossa. The posterior end of the fossa is interconnected with the central portion of the antorbital fenestra by a distinct groove that extends posterodorsally to the dorsal corner of the posterior process (Fig. 8). This groove otherwise only occurs in the specimen USNM 2673 (Tschopp, Mateus & Benson, 2015). Remaining parts of the external surface of the maxilla do not bear other distinctive morphological features, with the exception of the anterior-most portion, where a few nutrient foramina can be seen, similar to those on the premaxilla. The number of maxillary teeth is difficult to discern in the mounted skull, but is approximately 12.

Figure 6 Skull of Galeamopus pabsti SMA 0011 in supposed habitual pose.

The skull is figured in posterodorsal (A), anterodorsal (B), left lateral (C), and posteroventral views (D), following our terminology section. Dark, uniformely colored elements were lacking and reconstructed for the mounted skull. Note the shallow groove on the premaxilla, extending from the lateral margin anteromedially (1), and the typical, flagellicaudatan ‘chin’ on the dentary (2). Abb.: an, angular; aof, antorbital fenestra; bo, basioccipital; bpr, basipterygoid process; bt, basal tubera; d, dentary; ex, exoccipital; f, frontal; fm, foramen magnum; j, jugal; ltf, laterotemporal fenestra; m, maxilla; n, external nares; na, nasal; o, orbit; os, orbitosphenoid; osr, otosphenoidal ridge; p, parietal; pf, prefrontal; pm, premaxilla; po, postorbital; popr, paroccipital process; pro, prootic; ptf, posttemporal fenestra; q, quadrate; qj, quadratojugal; sa, surangular; so, supraoccipital; sq, squamosal; stf, supratemporal fenestra. Scale bar = 10 cm.

Nasal. The right nasal is complete. It lies anterior to the frontal, and medial to the prefrontal (Figs. 4–7). A slender, anterior process connects to the maxilla. The nasal is a subtriangular element with a slightly concave anteromedial edge forming a part of the external naris, and posterior and lateral edges that include an angle of about 120°. The anteromedial edge is dorsoventrally thin, but the nasal suddenly gains thickness from there backwards and outwards. The medial corner does not reach the skull midline, such that the two nasals do not touch each other medially. The external naris thus extends posteriorly between the nasal bones into a notch between the frontals. A similar condition might be present in Kaatedocus, which has an anterior notch between the frontals as well, but no nasal is preserved in the holotypic skull, which would confirm the posterior extension of the naris (Tschopp & Mateus, 2013b).

Prefrontal. Both prefrontals are complete. They contact the frontals posteriorly, the nasals medially, the lacrimals posterolaterally, and the maxillae anterolaterally (Figs. 4–7; Table 1). The prefrontals are short, and have an anteroposteriorly convex dorsal surface. Their lateral margin is straight, the medial one is anteriorly and posteriorly concave for the articulation with the nasal and the frontal, respectively. A sharply pointed, medially projecting process separates the two concavities. The posterior edge is anterolaterally-posteromedially oriented, forming a hook-like posteromedial process as is typical for Diplodocidae (Wilson, 2002; Whitlock, 2011a). The process almost reaches the frontal midlength, as is the case in the diplodocine skulls CM 3452 and 11161 (Tschopp, Mateus & Benson, 2015). Anteriorly, the prefrontal tapers to a narrow tip, which is slightly dorsoventrally expanded. The left element bears a small nutrient foramen on the dorsal surface of the anterior part. The ventromedial edge is very distinct.

Frontal. Both frontals are completely preserved. They contact the prefrontal anterolaterally, the nasal anteriorly, each other medially, the parietal posteromedially, and the postorbital posterolaterally (Figs. 4–7). Ventrally, the frontal makes contact with the braincase, articulating with the orbitosphenoid. The frontals have a smooth dorsal surface, which is slightly convex posterolaterally-anteromedially. Their medial border is generally straight, but curves laterally at its posterior and anterior ends, leaving an opening between each other. However, the posterior curvature shows broken edges, so that it is uncertain how much of this opening is due to taphonomic breakage. Therefore, we did not indicate the presence of a pineal foramen in the reconstruction drawing (Fig. 7; Mateus & Tschopp, 2017), also because this is usually interpreted to be absent in diplodocids (Whitlock, 2011a; Whitlock, 2011b).

Figure 7 Skull reconstruction of Galeamopus pabsti.

The reconstruction is in dorsal (A) and lateral view (B), and was created by Simão Mateus (ML), and based on the holotypic skull of SMA 0011. Lacking bones were reconstructed after Diplodocus (Whitlock, 2011b). Only the bones preserved in the skull of SMA 0011 are labeled. Abb.: an, angular; bpr, basipterygoid process; d, dentary; f, frontal; j, jugal; la, lacrimal; m, maxilla; n, nasal; p, parietal; pf, prefrontal; pm, premaxilla; po, postorbital; popr, paroccipital process; q, quadrate; qj, quadratojugal; sa, surangular; sq, squamosal.

The anterior curvature forms an anterior notch between the frontals (length 18 mm), similar to the condition in Kaatedocus. The anterior notch is wider than in Spinophorosaurus (Knoll et al., 2012), and different from Kaatedocus in being V-shaped rather than U-shaped (Tschopp & Mateus, 2013b). This differs from the anterior midline projection formed by the frontals of Galeamopus hayi HMNS 175. The anterior margin of the frontal of G. pabsti SMA 0011 is strongly convex transversely in order to accommodate the posterior, hook-like process of the prefrontal anterolaterally. From the posterior-most point of the posterior process of the prefrontal, the frontal has a straight edge extending obliquely anterolaterally, until it reaches the lateral, orbital edge, with which it forms a very acute angle. The lateral border is distinctly concave in dorsal view, smooth in its anterior part, but becoming highly rugose posteriorly, close to where it articulates with the postorbital. Here, the lateral and posterior edges form an acute angle. The lateral portion of the posterior margin is slightly displaced anteriorly, compared to the medial portion, resulting in a somewhat sinuous posterior edge. Ventrally, the frontals are marked by a distinct ridge, extending obliquely from the anterolateral corner, below the posterior process of the prefrontal, to an elevated, broad area for the attachment of the braincase.

Postorbital. Both elements are complete. The postorbital is a triradiate bone with an anterior process articulating with the jugal, a posterior process overlapping the squamosal laterally, and a dorsomedial process covering the posterior edge of the frontal in posterior view and connecting to the anterolateral process of the parietal medially, thereby excluding the frontal from the margin of the supratemporal fenestra (Figs. 4–7). Anteromedially, the dorsomedial process abuts the antotic process of the braincase (Fig. 9). The anterior process has a subtriangular cross section, transversely elongate, with a narrow lateral and a very thin medial margin (Table 1). The dorsal surface of the anterior process is slightly concave transversely. Towards the anterior end, the process tapers to a point. The posterior process is short and triangular. At its base, one (on the right postorbital) or two (on the left element) nutrient foramina occur. The process is compressed transversely. The dorsomedial process is curved dorsoventrally, with the concave surface facing anteriorly. It is relatively high dorsoventrally, but narrow anteroposteriorly. It is anteroposteriorly broader laterally than medially. The anterior face of the dorsomedial process is marked by a horizontal ridge at its base. The ridge supports the posterior edge of the frontal.

Jugal. Both jugals are preserved and complete. The jugal is a flat, relatively large bone with a posterior process contacting the postorbital and a dorsal process articulating with the lacrimal (Figs. 4–7). The main portion connects to the quadratojugal ventrally and the maxilla anteriorly. The jugal forms the anteroventral rim of the orbit, the posteroventral border of the antorbital fenestra, and the anterodorsal edge of the laterotemporal fenestra. The bases of the dorsal and posterior processes are relatively broad, before they taper dorsally and posteriorly, respectively (Table 1). The dorsal process is bifid (Fig. 4). The anterior edge of the jugal is slightly concave, as is the anteroventral margin. These two edges form a rounded anteroventral corner.

Quadratojugal. The quadratojugals are both complete. They are transversely thin bones with a posterodorsal process overlying the quadrate laterally, and a long anterior ramus (Table 1) contacting the jugal dorsally and the maxilla anteriorly (Figs. 4–7). The quadratojugals form the anteroventral margins of the laterotemporal fenestrae, and the ventral borders of the skull. The anterior ramus of the quadratojugal is narrow at its base but expands dorsoventrally towards its anterior end. The ventral edge is almost straight; it is thus the concave dorsal margin of the anterior ramus that accounts mostly for this dorsoventral expansion. The shape of the anterior margin is not discernible in the mounted skull, but based on the photo taken before the mount, it bears a small dorsal projection that connects to the jugal and excludes the maxilla from the margin of the laterotemporal fenestra, and slightly tapers anteriorly towards the articulation with the maxilla. The posterodorsal process is less than half the length of the anterior process. It is inclined posterodorsally, as in all diplodocids (Upchurch, 1998; Wilson, 2002; Whitlock, 2011a). It is anteroposteriorly convex externally, relatively broad at its base, and tapers to a point dorsally, reaching about midlength of the quadrate shaft.

Lacrimal. Only the dorsal half of the left lacrimal is preserved. It is a narrow element expanding towards its dorsal end (Table 1), where it underlies the posterodorsal process of the maxilla anteriorly, and contacts the prefrontal dorsally, and possibly the nasal medially (Figs. 4, 6, 7). Ventrally, the lacrimal would contact the jugal, if this part of the bone were preserved. The lacrimal separates the orbit from the antorbital fenestra. It is anteroposteriorly narrow in its ventral half, with a triangular cross section, flat externally but bearing a distinct dorsoventral ridge internally. The anterior edge has a short, but dorsoventrally high, anterior process at its dorsal end. The posterior margin is generally straight, with only a weak bulge on its dorsal portion. The dorsal-most end curves backwards, below the prefrontal. The internal ridge becomes slightly more pronounced dorsally, posteriorly enclosing the lacrimal foramen, which is small and shallow in SMA 0011.

Figure 8 Maxillary canal in the skull of Galeamopus pabsti SMA 0011.

Skull and maxillary canal (arrow in the inset) are figured in right lateral view. The canal is herein interpreted as an autapomorphy of G. pabsti. Abb.: aof, antorbital fenestra; j, jugal; m, maxilla; paof, preantorbital fossa. Scale bar in skull overview = 10 cm.

Figure 9 Braincase of Galeamopus pabsti SMA 0011 in anterolateral view.

Note the contacts between the frontal, postorbital, and the antotic process (white lines). The parasphenoid rostrum is broken. Abb.: anp, antotic process; f, frontal; la, lacrimal; os, orbitosphenoid; po, postorbital; popr, paroccipital process; psr, parasphenoid rostrum.

Quadrate. Only the right quadrate is preserved, but it is complete. It has a complex anatomy, with a quadrate shaft articulating with the squamosal and the paroccipital process posterodorsally and posteroventrally, respectively; a pterygoid flange interconnecting the outer skull with the pterygoid medially; and a ventral ramus overlapped by the quadratojugal externally and bearing the articulating surface for the lower jaw ventrally (Figs. 4, 5, 7). The quadrate shaft is elongate posteriorly (Table 1), and has concave dorsal and ventrolateral surfaces. The lateral edge is a thin crest, where it is not capped by the squamosal or the quadratojugal. The posterior surface of the quadrate shaft and the ventral ramus is shallowly concave, forming the quadrate fossa. The pterygoid flange originates on the medial half of the quadrate shaft. It is very thin mediolaterally, but anteroposteriorly long, and curves medially at its dorsal tip. The dorsal edge of the flange is straight and more or less horizontally oriented. The medial side of the pterygoid flange is concave, but does not form such a distinct fossa like that present in Kaatedocus SMA 0004 (Tschopp & Mateus, 2013b). The ventral ramus of the quadrate of Galeamopus pabsti SMA 0011 is subtriangular in cross-section, with concave anterior and posterolateral surfaces. It has a thinner lateral than medial margin. The articular surface is subtriangular, with a concave anterior border, and a pointed posterior corner. The entire ventral ramus of the quadrate of SMA 0011 is posterodorsally inclined, as in all diplodocids (Upchurch, 1998; Wilson, 2002; Whitlock, 2011a).

Squamosal. Both squamosals are preserved, but lack a part of their anterior process (the right one more so than the left). The squamosals form the posterolateral corner of the skull. They have a complicated morphology, accommodating a variety of elements from the braincase and outer skull (Figs. 4–7). The anterior process overlies the posterior end of the quadrate. Dorsally, the squamosal is laterally covered by the posterior process of the postorbital and forms the external margin of the supratemporal fenestra. Posteriorly the squamosal contacts the paroccipital process and dorsoposteriorly the posterolateral process of the parietal. The squamosal is strongly curved posterolaterally. The anterior process appears to be the longest of all squamosal processes (Table 1), even though it is not preserved in its entire length. The ventral edge of the squamosal bears a short ventral projection at its posterior end, similar to, but much less distinct than the ventral prong present in advanced dicraeosaurids (Salgado & Calvo, 1992; Whitlock, 2011a). A concave area on the dorsolateral surface accommodates the posterior process of the postorbital. Other morphological features are difficult to observe in the articulated, reconstructed skull of SMA 0011.

Parietal. Both parietals are complete but slightly distorted. They are tightly sutured with the frontals anteriorly and have a short anterolateral process to contact the dorsomedial process of the postorbital, with which they form the anterior margin of the supratemporal fenestra (Figs. 4–7). The posterior face of the parietal contacts the exoccipital and the supraoccipital medioventrally. The posterolateral process of the parietal forms the posterior margin of the supratemporal fenestra and reaches the squamosal laterally. The dorsal portion of the parietal in SMA 0011 is very narrow anteroposteriorly (Table 1). The two elements do not touch each other medially, but this appears to be due to postmortem breakage of the extremely thin bone behind the parietal fenestra, which the parietals form together with the frontals. The dorsal portion is flat and not well separated from the posterior surface by a transverse nuchal crest (sensu Knoll et al., 2015) like that in Kaatedocus (Tschopp, Mateus & Benson, 2015). The parietal of Galeamopus pabsti SMA 0011 widens anteroposteriorly at its lateral end, where it develops a short anterolateral and a long and dorsoventrally deep posteroventral process. The parietal thus contributes most to the margin of the supratemporal fenestra. The posterior surface has an oblique ventromedial border, which has a very sinuous suture together with the supraoccipital. The dorsal margin of the posterolateral process is straight and does not cover the anterior border of the supratemporal fenestra in posterior view. The ventral edges are excluded from the posttemporal fenestra by the squamosal and a laterally projecting spur of the exoccipital.

Figure 10 Supraoccipital of Galeamopus pabsti SMA 0011 in posterodorsal view.

Note the unusual shape with the vertical groove on the sagittal area (inset). Abb.: f, frontal; fm, foramen magnum; oc, occipital condyle; p, parietal; pf, prefrontal; po, postorbital; popr, paroccipital process; q, quadrate; qj, quadratojugal; so, supraoccipital; sq, squamosal. Scale bar in skull overview = 10 cm.

Supraoccipital. The supraoccipital is complete and fused with the parietals and the exoccipital-opisthotic complex. The supraoccipital is a somewhat hexagonal bone, which contacts the parietals dorsolaterally, the exoccipital-opisthotic complex ventrolaterally, and borders the foramen magnum ventrally (Figs. 5, 6, 10). The suture with the exoccipital-opisthotic is barely visible. The dorsolateral edges of the supraoccipital are slightly concave. The ventrolateral edges are visible only laterally; further medially, the suture becomes obliterated up to the foramen magnum, but probably extended below the two distinct tubercles located dorsolateral to the foramen magnum. These tubercles served for the attachment of the proatlases. The tubercles are ellipsoid, oriented with their long axes extending dorsomedially-ventrolaterally. The elevation is much more distinct ventrally than dorsally. The dorsal portion of the supraoccipital bears a complex arrangement of ridges and concavities (Fig. 10). This complex structure is symmetrical and well-defined, arguing against a taphonomic or pathological origin. No distinct sagittal ridge occurs. In fact, the elevated area is marked by a vertical midline groove, which is otherwise only present in the skull USNM 2673 among diplodocids. A similar, but wider, sagittal groove occurs convergently in the titanosaurs Rapetosaurus krausei, Muyelensaurus pecheni, and Bonatitan reigi (Curry Rogers & Forster, 2004; Calvo, González Riga & Porfiri, 2007; Salgado Gallina, & Paulina-Carabajal, 2015). Given that the supraoccipital of Galeamopus hayi HMNS 175 does appear to bear a distinct sagittal nuchal crest, the groove is here interpreted to be an autapomorphy of the species Galeamopus pabsti. The supraoccipital has its greatest width slightly below midheight. No distinct foramina occur close to the border with the parietal, unlike in Kaatedocus (Tschopp & Mateus, 2013b). The dorsal-most portion of the supraoccipital of SMA 0011 tapers, not forming a distinct dorsal elevation as in Apatosaurus CM 11162 (Berman & McIntosh, 1978), or the indeterminate flagellicaudatan MB.R.2388 (Remes, 2009).

Exoccipital-opisthotic complex. The outer portion of the braincase is completely preserved. No sutures can be seen between the exoccipital and the opisthotic (the fused complex is sometimes called otoccipital; Knoll et al., 2012; Knoll et al., 2015; Royo-Torres & Upchurch, 2012). They bear two elongate paroccipital processes that extend lateroventrally to articulate with the squamosal and the posterior end of the quadrate (Figs. 5 and 6). Medially, the exoccipital-opisthotic borders almost the entire foramen magnum except for a small dorsal contribution of the supraoccipital. The exoccipital forms the dorsolateral corners of the occipital condyle. As in Suuwassea and Diplodocus CM 11161, the exoccipital almost excludes the basioccipital from the participation in the dorsal surface of the occipital condyle (Harris, 2006a). The lateral surface of the condylar neck is pierced by two foramina, which are the exits for cranial nerve XII (Fig. 11; Knoll et al., 2012), unlike the condition in Amargasaurus, which only has a single exit (Paulina Carabajal, Carballido & Currie, 2014). The two exits for cranial nerve XII are anteriorly bordered by the crista tuberalis, which separates them from the opening for cranial nerves IX–XI (the metotic foramen; Knoll et al., 2012; Royo-Torres & Upchurch, 2012), and extends from the ventral edge of the paroccipital process onto the basioccipital, of which it forms the posterolateral edge until it reaches the basal tubera (Fig. 11). The metotic foramen is anteriorly bordered by the crista interfenestralis, which separates two well developed fossae between the crista tuberalis and the otosphenoidal crest (Makovicky et al., 2003).

Figure 11 Braincase of Galeamopus pabsti SMA 0011 in posteroventral view.

Note that the basipterygoid processes are mounted dorsal to their actual position, and that the parasphenoid rostrum is broken off. The transverse width of the basal tubera is 42 mm. Abb.: bo, basioccipital; bpr, basipterygoid process; bs, basisphenoid; bt, basal tuber; cif, crista interfenestralis; cn, cranial nerve; ct, crista tuberalis; f, frontal; n, external naris; na, nasal; o, orbit; oc, occipital condyle; osr, otosphenoidal ridge; p, parietal; popr, paroccipital process; psr, parasphenoid rostrum; q, quadrate.

The paroccipital processes of Galeamopus pabsti SMA 0011 have slightly convex external surfaces, but do not bear a ridge as in Kaatedocus (Tschopp & Mateus, 2013b). The ventral edge of the paroccipital process is straight, only the dorsal corner of the distal end is expanded dorsally, resulting in a distinctly concave dorsal edge. The lateral margin of the paroccipital process is subtriangular, with a longer, vertically oriented dorsal portion, and a shorter, laterally inclined ventral part. In lateral view, it is straight, unlike the curved ends of the element in Suuwassea and Galeamopus hayi (Harris, 2006a; Tschopp, Mateus & Benson, 2015).

Basioccipital and basisphenoid. The basioccipital forms the main portion of the occipital condyle. It is relatively short and connects the articular surface of the occipital condyle with the basal tubera (Fig. 11), which are of about the same width (Table 1). The articular surface of the occipital condyle is offset from the condylar neck. Narrow ridges connect the midline of the ventral aspect of the condylar neck with the posteromedial corners of the basal tubera and the lateral face of the neck with the crista tuberalis. The ridges result in concave lateral surfaces of the basioccipital and concave posterior surfaces of the basal tubera. The concavity on the posterior surface of the tubera is anteroventrally confined by a distinct, transversely convex ridge, which separates the posterior and ventral surfaces of the tubera (Fig. 11). The basal tubera are box-like, and medially separated by a distinct, but relatively narrow notch. The ventral edges of the tubera form a nearly straight line in posterior view, whereas the anterior edges are angled in a wide V-shaped manner in ventral view. Anteriorly, the basipterygoid processes attach to the tubera. In the reconstructed skull, the processes are mounted slightly dorsal to their actual location, above the anteroventral end of the otosphenoidal crest (Fig. 12). When articulated properly, they would be elongate (5.3 times longer than wide; Table 1), straight, and would form a narrower angle than as mounted. This is important because shorter and more widely diverging basipterygoid processes are typical for Apatosaurus, whereas narrower angles are typical in Diplodocus (Berman & McIntosh, 1978). The processes are not as well connected at their base as is the case in Kaatedocus (Tschopp & Mateus, 2013b). The distal ends of the basipterygoid processes are expanded.

Figure 12 Braincase of Galeamopus pabsti SMA 0011 in left anterolateral view.

Note that the basipterygoid processes are mounted dorsal to their actual position, and that the parasphenoid rostrum is broken off. The transverse width of the distal end of the left basipterygoid process is 19 mm. Abb.: bpr, basipterygoid process; bt, basal tuber; cn, cranial nerve; oc, occipital condyle; ocv, orbitocerebral vein foramen; osr, otosphenoidal ridge; psr, parasphenoid rostrum.

Orbitosphenoid. The orbitosphenoids delimit the endocranial cavity anteriorly and attach to the frontals dorsally, each other medially, and the laterosphenoids posterolaterally. Each orbitosphenoid is relatively wide dorsally and has an anteroventral process, which is expanded at its end and separates the two openings for cranial nerves II medially (the optic foramen) and III laterally (the oculomotor foramen; Fig. 12; Janensch, 1935; Harris, 2006a; Balanoff, Bever & Ikejiri, 2010; Knoll et al., 2015). Unlike the condition in Suuwassea or Europasaurus (Harris, 2006a; Sander, Mateus & Laven, 2006), the optic foramen of Galeamopus is bridged over by bone medially. Anterodorsally, the two orbitosphenoids form the olfactory fenestra together with the frontals (the exit for cranial nerve I; Fig. 12; Janensch, 1935; Balanoff, Bever & Ikejiri, 2010), and posterolaterally, at the junction with the laterosphenoid, the foramen for cranial nerve IV (the trochlear foramen; Balanoff, Bever & Ikejiri, 2010) defines the outline of the orbitosphenoid.

Laterosphenoid. The laterosphenoid mainly consists of a crest that bears the antotic (or capitate; Knoll et al., 2012; Knoll et al., 2015) process posterodorsally and extends anteroventrally to join the otosphenoidal crest. It connects to the frontal and parietal posterodorsally, the orbitosphenoid anterodorsally, and the prootic posteroventrally. As for the orbitosphenoid, the laterosphenoid outline is defined by various openings: cranial nerves III and IV anteriorly at the junction with the orbitosphenoid, the trigeminal foramen posterodorsally (cranial nerve V; Balanoff, Bever & Ikejiri, 2010), as well as the oculomotor foramen and the abducens foramen anteroventrally (Fig. 12; Balanoff, Bever & Ikejiri, 2010). Dorsal to the opening for cranial nerve IV, there is a separate, small opening for the orbitocerebral vein, similar to the condition in Diplodocus and other sauropods, but different from Amargasaurus (Paulina Carabajal, 2012; Paulina Carabajal, Carballido & Currie, 2014). The antotic crest separates the trigeminal foramen from the other openings. The antotic process is dorsoventrally higher than anteroposteriorly long, and tapers laterally to a rounded tip, which contacts the postorbital.

Prootic. The prootic lies between the laterosphenoid anterodorsally, the parietal and paroccipital processes posterodorsally, and the basisphenoid anteroventrally. The prootic bears the well-developed otosphenoidal crest, which extends relatively far laterally, but is very thin dorsoventrally. It does not end in an additional transverse expansion anteriorly, as would be typical for dicraeosaurids (Janensch, 1935; Paulina Carabajal, Carballido & Currie, 2014). The otosphenoidal crest extends between the foramen for cranial nerve V more anteriorly and the ones for cranial nerves IX–XII more posteriorly (Paulina Carabajal, Carballido & Currie, 2014). Posterodorsally, the otosphenoidal crest extends to the base of the paroccipital processes, and bifurcates to enclose the foramen for cranial nerve VII. The two branches reunite before reaching the paroccipital process, similar to the condition in Amargasaurus (Paulina Carabajal, Carballido & Currie, 2014).

Pterygoid. The left pterygoid is only partly prepared (Fig. 13). The pterygoid connects the quadrate posterolaterally with the basipterygoid processes posteromedially, the ectopterygoid and palatine anterolaterally, and the vomer anteromedially. The two elements would join along the midline of the skull. The pterygoid of SMA 0011 resembles the same bone in the indeterminate diplodocine CM 3452 in its dorsoventrally deeper shape compared to Camarasaurus and Giraffatitan (McIntosh & Berman, 1975). A shallow articulation facet for the basipterygoid processes lacks the hook-like process present in dicraeosaurids and Camarasaurus (Wilson, 2002; Whitlock, 2011a).

Figure 13 Right pterygoid of Galeamopus pabsti SMA 0011.

The pterygoid is shown in lateral (A) and medial (B) views. The element is only partly prepared, the lighter color is matrix adhered to the darker bone. Abb.: ar, anterior ramus; er, ectopterygoid ramus; qr, quadrate ramus. Scale bar = 5 cm.

Ceratobranchial. Only the right ceratobranchial is preserved, but appears to be almost complete (Fig. 14). The ceratobranchial is a bone of the hyoid apparatus, with no bony connections to the rest of the skull (Wilson et al., 2016). It is a narrow bone, with a distinct upward curve at midlength. The anterior ramus becomes transversely flattened towards its anterior end, which bears a shallow longitudinal groove on the medial side. The ceratobranchial slightly widens dorsoventrally where it curves upwards and towards the squamosal, as was shown in Tapuiasaurus (Zaher et al., 2011; Wilson et al., 2016). The posterodorsal end is rounded and offset from the shaft by a distinct rim.

Figure 14 Right ceratobranchial of Galeamopus pabsti SMA 0011.

The ceratobranchial is shown in medial (A) and lateral (B) views. Abb.: ar, anterior ramus; sqr, squamosal ramus. Scale bar = 10 cm.

Mandible

The mandibles preserve both dentaries and surangulars, and the left angular. Two additional, thin bones might represent the prearticulars. No articular, splenial, and coronoid is preserved.

Dentary. Both dentaries are preserved. The dentary is the anterior-most bone of the lower jaw and the only one bearing teeth. Posteriorly, it is followed by the surangular dorsally and the angular ventrally (Figs. 4–7). Internally, it would be overlain by the splenial ventrally, but this is not visible due to the mount. The dentary is a thin bone, with a dorsoventrally high dentigerous portion (Table 1), having the typical, ventrally projecting ‘chin’ of flagellicaudatans (Fig. 6; Upchurch, 1998; Whitlock, 2011a). The anteromedial portion is marked by several small, irregularly placed pits. A relatively larger, distinct foramen pierces the lateral surface at midheight below the posterior-most tooth. The medial wall of the dentigerous portion of the dentary projects further dorsally than the medial wall. Posterior to the tooth bearing portion, the dentary tapers in dorsoventral height, the right one much more so than the left. The symphysis is oblong and strongly anteriorly inclined. There are at least eleven, possibly twelve, dentary teeth.

Surangular. Both surangulars are present. This bone is very flat transversely, curves ventrally at its posterior end and bears a foramen at its highest point, which is also the highest point of the entire lower jaw (Figs. 4–7). The jaw does not bear a coronoid eminence.

Angular. Both angulars are incomplete anteriorly. They are concave externally, due to the laterally curving ventral edge. They taper relatively continuously anteriorly, but abruptly at their posterior ends (Figs. 4–7), where they expand transversely in order to accommodate the articular, which is not preserved.

?Prearticular. Both prearticulars appear to be present, but are partly hidden in the mount or only partially prepared, and separately stored in the SMA collections (Fig. 15). They are thin, elongate bones that taper posteriorly. A very shallow groove marks the probable lingual surface, extending anteroposteriorly, following the somewhat sinuous curve of the dorsal edge of the bone. In its anterior half, the bone becomes slightly thicker mediolaterally and curves outwards.

Figure 15 Right ?prearticular of Galeamopus pabsti SMA 0011 in lingual view.

Note the shallow longitudinal canal (arrows). Scale bar = 5 cm.

Teeth

The teeth have the typical diplodocoid, peg-like shape, and have a Slenderness Index (SI) of approximately 4 (Fig. 16; Tschopp, Mateus & Benson, 2015: tab. S16). They are slightly wrinkled but do not have denticles. Worn teeth usually have a single wear facet at a low angle to the long axis of the tooth, but some teeth also show two facets that are conjoined apically. In these teeth, the lingual facet is more steeply inclined than the labial one. The crown tips are slightly wider than deep, which is especially visible in replacement and/or unworn teeth, which have a very weakly spatulate upper-most crown. The enamel is distributed evenly on all sides, and no grooves mark the lingual face. In the jaws, the teeth are inclined anteriorly relative to the long axis of the jaw, and set side-by-side without overlapping each other.

Figure 16 Teeth of Galeamopus pabsti SMA 0011 in lingual view.

They were found disarticulated from the skull. Abb.: tc, tooth crown; tr, tooth root. Scale bar = 2 cm.

Axial skeleton

Terminology. Vertebral laminae are described following the nomenclature of Wilson (1999), with the changes proposed by Wilson (2012), Tschopp & Mateus (2013b) and Carballido & Sander (2014), whereas fossa terminology follows the one of Wilson et al. (2011). The use of “pleurocoel” herein follows the definition of Carballido & Sander (2014: p. 337): “a lateral excavation with well-defined anterior, ventral and dorsal margins“.

Cervical vertebrae (Figs. 17–32; Table 2)

Preservation. Thirteen cervical vertebrae are present, as is the right proatlas. The cervical vertebrae were found partly articulated. The proatlas and atlas were recovered among the disarticulated skull elements. Axis to CV 5 were lying semi-articulated in close association, followed by the slightly disarticulated CV 6–8. After a short gap of 0.3 m, CV 9 and 10 were found articulated, and finally a block of five articulated elements including the cervico-dorsal transition was recovered at a distance of about 1 m from the next nearest vertebrae (Fig. 3). The gap between CV 8 and 9 is interpreted to be too short to accommodate yet another element, which in this area of the neck already reach lengths of at least 150% the distance of the gap. Also, measurements of the posterior cotyle of CV 8 and the anterior condyle of CV 9 more or less fit to each other, taking the deformation of CV 8 into account (Table 2). Thus, the only reasonable position, where cervical vertebrae could be missing, is between CV 10 and the block including the cervico-dorsal transition. None of the cervical ribs were fused to their centra, and certain anterior to middle ribs were found at some distance from the vertebrae. However, combining the positional information from the quarry maps and the size and side of the ribs, an attribution of most of them to their respective centra was possible. Five ribs belonging to the articulated cervico-dorsal transition were found in place, yielding crucial information about the changes in morphology from the neck to the back. Two pairs of them are transitional in shape, but can still be interpreted as cervical ribs due to the presence of an anterior process and their short posterior shaft (see below). They belong to the second and third articulated vertebra of the transitional block. One pair and a single rib are definitive dorsal ribs, and were found semi-articulated with the last two vertebrae in the block.

Table 2 Measurements of cervical vertebrae of Galeamopus pabsti SMA 0011 (in mm).

CV	apL	gH	cL	cmW	diW	prW	poW	ppL	ppH	ctW	ctH	
Atlas			25	47						49	28	
Axis	146	201	131				97  (comp)	86	31	30  (def)	55  (def)	
CV 3	240	251	198	25	110	94		140	41	45	72	
CV 4	330	347	268	27		111	72  (comp)	194	42	69	72	
CV 5	409	400	320	33	155  (def)	109  (def)	105  (def)	235	47	63	87	
CV 6	480	325	389	44	175  (def)	131		259	47	97	92	
CV 7	483	300	406	55	380  (est)	185	183  (def)	260	51	117	104	
CV 8	505	inc	435	74	440  (est)	213	235	256	20  (comp)	160  (def)	90  (comp)	
CV 9	523	368	405	80	310  (est)	174	250  (est)	259	60	156	141	
CV 10	500	420	387	70  (def)	360  (est)	193  (def)	185  (est)	250	62	162	165	
CV ?12	415	380	400	116		350  (est)		230	34		185	
CV ?13	450	475		95				185	57		195	
CV ?14		485	355	108				185	40		190	
CV	cdW	cdH	nsH	cL-cd	naH	Comments	
Atlas	49	16		25		Cotyle is anterior, condyle posterior	
Axis	49	57	134	115	66	ppL measured on right side, cL-cd measured at midheight	
CV 3	42	42	179	169	110	ppL measured on right side, cL-cd measured at midheight	
CV 4	36 (comp)	58	252	229	118	ppL and ppH are the mean of left and right sides	
CV 5	61	64	256	284	147		
CV 6	55	63	203	344	149 (est)		
CV 7	86	67	160	341	140		
CV 8	104 (def)	57 (comp)	inc	383	80 (comp)		
CV 9	150 (est)	114	200	362	153		
CV 10		150 (def)	230	354	140		
CV ?12	180	165	240	370	208		
CV ?13			325	310	222		
CV ?14		160	365	305	170 (est)		
Notes.

Abb. apL anteroposterior length

cdH height condyle

cdW width condyle

cL centrum length

cL-cd centrum length without condyle

cmW centrum minimum width

comp compressed

ctH height cotyle

ctW width cotyle

def deformed

diW width across diapophyses

est estimated

gH greatest height

inc incomplete

naH height neural arch (below poz)

nsH height neural spine

poW width across postzygapophyses

ppH pneumatopore height

ppL pneumatopore length

prW width across prezygapophyses

Proatlas. The right proatlas is preserved and complete (Fig. 17). It connects the braincase with the atlantal neurapophyses. The proatlas of SMA 0011 is strongly curved and tapers distally. The proximal articular surface is ovoid, with the largest width located in the dorsal half. The medial surface is concave, the lateral one convex. The proatlas of SMA 0011 is different from the element in Kaatedocus (see Tschopp & Mateus, 2013b: figs.  3–6) due to its much narrower distal tip.

Atlas. The atlantal centrum is not fused to the neurapophyses (Fig. 18). It has a well-developed anteroventral lip as is typical for diplodocids, and convergently present, although less evident in several other sauropods (Mannion, 2011; Whitlock, 2011a). A large foramen lies between the posterolateral projections at the posteroventral edge of the centrum. The lateral surface of the centrum is concave and bears a foramen as well, resembling an incipient pleurocoel. The neurapophyses have a relatively wide base, and turn upwards and backwards to articulate with the prezygapophyses of the axis. A wide medial process occurs anteriorly, as in the specimen AMNH 969 (Holland, 1906). This process articulates with the proatlas, and is much better developed than in Diplodocus USNM 2672 or Kaatedocus (Marsh, 1896; Hatcher, 1901; Tschopp & Mateus, 2013b). A small but distinct subtriangular process occurs on the opposite side of the medial process of the atlantal neurapophyses of SMA 0011, projecting laterally. The posterior wing of the neurapophysis does not taper as in Kaatedocus siberi (Tschopp & Mateus, 2013b), but remains subrectangular with a widely rounded distal end. This morphology was proposed as an unambiguous autapomorphy for the genus Galeamopus by Tschopp, Mateus & Benson (2015), but is also present in the dicraeosaurid Amargasaurus cazaui MACN-N 15 (Paulina Carabajal, Carballido & Currie, 2014). However, the wide distal ends of the neurapophyses remain autapomorphic for Galeamopus within Diplodocidae.

Figure 17 Right proatlas of Galeamopus pabsti SMA 0011.

The proatlas is shown in lateral (A) and medial (B) views. Note the elongate and narrow distal tip. Scale bar = 2 cm.

Figure 18 Atlas of Galeamopus pabsti SMA 0011.

Atlantal neurapophyses (A) and centrum (B) in medial (A1, A2), right lateral (A3, B2), left lateral (A4, B4), dorsal (B1), anterior (B3), posterior (B5), and ventral view (B6). Abb.: avl, anteroventral lip; dip, distal process; lsp, lateral spur; mp, medial process; ncs, neurocentral synchondrosis; pl, pleurocoel; plp, posterolateral process; pnf, pneumatic foramen. Scale bar = 10 cm.

Axis. The axis of SMA 0011 (Fig. 19) has a closed but still slightly visible neurocentral synostosis, and unfused cervical ribs. The centrum is opisthocoelous. The pleurocoel extends over almost the entire centrum, and contains horizontal ridges at its anterior and posterior end. No vertical subdivision of the pleurocoel occurs. Anteriorly, the pleurocoel extends onto the dorsal surface of the parapophysis. The ventral surface of the centrum bears a distinct longitudinal keel medially, which widens anteriorly and posteriorly, where it also becomes rugose. The centrum is diagenetically transversely compressed ventrally, but it is clear that the ventral surface was constricted at midlength, and it appears that the wider posterior part of the ventral keel was laterally accompanied by shallow depressions. The parapophysis is rounded and faces anterolaterally and slightly ventrally. The diapophysis projects somewhat posteriorly, but does not bear a distinct posterior process. The neural arch is high and weakly posteriorly inclined. The prezygapophyses are not preserved. The only well-defined laminae are the PODL and the PRSL. The PRSL is slightly expanded transversely at its anteroventral end, similar to, but not as distinct as in AMNH 969 (Tschopp, Mateus & Benson, 2015). In lateral view, the PRSL is slightly concave ventrally, and straight in the upper part. The spine top is rugose, weakly expanded transversely, and situated entirely anterior to the postzygapophyseal facets. This anterior placement of the summit is unusual for sauropods, but present in Diplodocus carnegii CM 84 (Hatcher, 1901). Unlike CM 84, however, the neural spine summit of SMA 0011 has a posterior projection, similar to the condition in Giraffatitan (Janensch, 1950). The margin of the SPOL is strongly concave in lateral view, becoming vertical in the upper part. Small epipophyses are present laterally above the postzygapophyses. They do not project posteriorly. A large rugose area is present on the lateral side of the spine, slightly above mid-height (Fig. 19). It is subtriangular, broader towards the SPOL, with a pointed, elongate tip towards the center of the SDF. This rugosity could be homologous to the distal lateral expansion in the axis of Camarasaurus and Suuwassea (Madsen, McIntosh & Berman, 1995; Harris, 2006b), but the neural spine top is much more elevated in SMA 0011. Such a rugosity appears to be absent in the axis of Diplodocus carnegii CM 84 (Hatcher, 1901). The postzygapophyses of the axis of SMA 0011 slightly overhang the centrum posteriorly, and bear subtriangular facets with a straight anterior border.

Figure 19 Axis of Galeamopus pabsti SMA 0011.

Axis shown in dorsal (A), posterior (B), right lateral (C), anterior (D), left lateral (E), and ventral (F) view. The round inset shows the right lateral side of the spine summit in posterolateral, and slightly dorsal view (not to scale). The prezygapophyses are not preserved. Note the short horizontal ridges in the pleurocoel (1), the depressions lateral to the ventral keel (2), the transverse expansion of the anteroventral extremity of the prsl (3), the anterior position of the neural spine summit, and its posterior projection (4), the rugose area on the lateral side of the neural spine (5). Abb.: di, diapophysis; epi, epipophysis; ncs, neurocentral synostosis; pap, parapophysis; pl, pleurocoel; podl, postzygodiapophyseal lamina; poz, postzygapophysis; prsl, prespinal lamina; sdf, spinodiapophyseal fossa; spof, spinopostzygapophyseal fossa; spol, spinopostzygapophyseal lamina. Scale bar = 10 cm.

Postaxial cervical vertebrae (Figs. 20–32). The cervical centra are all opisthocoelous and relatively elongate. As is typical for nearly all sauropods, the most elongate elements are the mid-cervical vertebrae (Table 2). All cervical centra have well-developed pleurocoels extending over almost the entire length of the centrum, also invading the dorsal surfaces of the parapophyses. The internal structure of the pleurocoel varies along the column: the anterior and posterior horizontal ridges described in the axis disappear by CV 4 and are present in only the right pleurocoel in CV 3 and 4 (Figs. 20 and 21). A vertical subdivision into anterior and posterior pneumatic fossae becomes visible in CV 3, and is pronounced from CV 5 backwards (Fig. 22). The subdividing ridge is oriented anterodorsally-posteroventrally, as in most sauropods. The posterior pneumatic fossae of CV 5–7 bear a large, slightly ellipsoid foramen at their anterior end, which pierces the median wall (Figs. 22–24). Whereas the median wall is thin posterior to this hole, it is transversely expanded anterior to the hole. The wider anterior margin of the hole bears a vertical groove that leads into a pneumatic foramen on the posterior face of its expanded portion (Fig. 23). Such a hole in the median wall is extremely rare in sauropods. Diplodocus carnegii CM 84 was reported to have confluent pleurocoels in posterior cervical vertebrae (Hatcher, 1901), “Morosaurus” agilis USNM 5384 shows this peculiarity in CV 3 (Gilmore, 1907), and a Camarasaurus axis has the same feature (AMNH 5761/X1, Osborn & Mook, 1921: pl. LXVII). Deep pneumatic openings are also present in mid-cervical centra of Galeamopus hayi HMNS 175, but these were left filled with sediment, and it remains unclear if these pierce the median wall or not (E Tschopp, pers. obs., 2010). The posterior pneumatic fossae of CV 5 and 6 of SMA 0011 become pointed posteriorly, due to the development of a shallow posteroventral fossa, which diagnoses most diplodocines (except Kaatedocus; Tschopp & Mateus, 2013b). From CV 6 backwards, the anterior pneumatic fossa becomes subdivided by a horizontal ridge at about mid-height. The ventral portion of the anterior fossa becomes vertically divided in CV 9 (Fig. 26). The latter is also the first element in the series to show a separation of the posterior-most portion of the posterior pneumatic fossa. Additionally, CV 10 has a horizontally subdivided posteroventral fossa (Fig. 27). In the first element of the articulated transitional series, the pleurocoel becomes less complex again (Fig. 28).

Figure 20 Cervical vertebra 3 of Galeamopus pabsti SMA 0011.

CV 3 shown in dorsal (A), posterior (B), right lateral (C), anterior (D), left lateral (E), and ventral (F) view. Note the horizontal ridge within the right pleurocoel (1) and the incipient vertical subdivision (2), the foramina lateral to the ventral keel (3), the deep anterior depression within the spinodiapophyseal fossa (4). Abb.:pap, parapophysis; podl, postzygodiapophyseal lamina; prz, prezygapophysis; spol, spinopostzygapophyseal lamina; sprl, spinoprezygapophyseal lamina; vk, ventral keel. Scale bar = 10 cm.

Figure 21 Cervical vertebra 4 of Galeamopus pabsti SMA 0011.

CV 4 shown in dorsal (A), posterior (B), right lateral (C), anterior (D), left lateral (E), and ventral (F) view. Note the horizontal ridge within the right pleurocoel (1), the posterior inclination of the spine summit (2). Abb.:pap, parapophysis; podl, postzygodiapophyseal lamina; prsl, prespinal lamina; prz, prezygapophysis; spol, spinopostzygapophyseal lamina; vk, ventral keel. Scale bar = 10 cm.

Figure 22 Cervical vertebra 5 of Galeamopus pabsti SMA 0011.

CV 5 shown in dorsal (A), posterior (B), right lateral (C), anterior (D), left lateral (E), and ventral (F) view. The round inset shows the left diapophysis in ventrolateral view (not to scale). Note the vertical subdivision of the pleurocoel (1), the foramen piercing the median wall (2), the dorsal subfossa within the spinodiapophyseal fossa (3), and the two parallel pcdl (4). Abb.: di, diapophysis; pap, parapophysis; pl, pleurocoel; ppf, posterior pneumatic fossa; prcdf, prezygapophyseal centrodiapophyseal fossa; prz, prezygapophysis; spol, spinopostzygapophyseal lamina; sprl, spinoprezygapophyseal lamina. Scale bar = 10 cm.

In the first preserved posterior cervical vertebra, the anterior condyle is damaged, so that it reveals the internal structure. The condyle is composed of large internal cavities, surrounded by 2–4 mm thick, relatively dense bony struts. The arrangement appears symmetric, with a subtriangular cavity dorsomedially, and two subcircular cavities following on both sides (Fig. 28).

Figure 23 Cervical vertebra 6 of Galeamopus pabsti SMA 0011.

CV 6 shown in posterior (A), right lateral (B), anterior (C), and left lateral (D) view. The round inset shows the right pleurocoel in posterolateral view (not to scale). Note the foramen piercing the median wall of the centrum (1), the horizontal subdivision of the anterior pneumatic fossa (2), the vertical accessory lamina subdividing the prezygapophyseal centrodiapophyseal fossa (3), the subfossae in the spinodiapophyseal fossa (4, 5), and the groove and foramen marking the anterior wall of the foramen piercing the median wall (6). Abb.: pap, parapophysis; pvfo, posteroventral fossa; spol, spinopostzygapophyseal lamina. Scale bar = 10 cm.

Figure 24 Cervical vertebra 7 of Galeamopus pabsti SMA 0011.

Neural arch (A) and centrum (B) shown in dorsal (A1, B5), ventral (A2, B6), posterior (A3, B1), right lateral (A4, B2), anterior (A5, B3), and left lateral view (A6, B4). The neural spine summit is not preserved. Note the large foramina piercing the (1) neural arch and (2) median wall of the pleurocoel. Abb.: cpol, centropostzygapophyseal lamina; cprl, centroprezygapophyseal lamina; cprl-f, centroprezygapophyseal lamina-fossa; epi, epipophysis; nc, neural canal; ncs, neurocentral synchondrosis; pap, parapophysis; pcdl, posterior centrodiapophyseal lamina; podl, postzygodiapophyseal lamina; prdl, prezygodiapophyseal lamina; pre, pre-epipophysis; prsl, prespinal lamina; pvf, posteroventral flange; pvfo, posteroventral fossa; tpol, interpostzygapophyseal lamina; tprl, interprezygapophyseal lamina. Scale bar = 10 cm.

Figure 25 Cervical vertebra 8 of Galeamopus pabsti SMA 0011.

Neural arch (A) and centrum (B) in dorsal (A1, B5), ventral (A2, B6), posterior (A3, B1), right lateral (A4, B2), anterior (A5, B3), and left lateral view (A6, B4). The neural spine summit is not preserved. Note the (1) bifid neural spine, (2) large neural arch foramen, and (3) anteriorly bifurcate pcdl. Abb.: cpol, centropostzygapophyseal lamina; cprl-f, centroprezygapophyseal lamina-fossa; epi, epipophysis; mt, median tubercle; ncs, neurocentral synchondrosis; pap, parapophysis; pcdl, posterior centrodiapophyseal lamina; poz, postzygapophysis; prdl, prezygodiapophyseal lamina; pre, pre-epipophysis; prz, prezygapophysis; pvf, posteroventral flange; spol, spinopostzygapophyseal lamina; tpol, interpostzygapophyseal lamina; tprl, interprezygapophyseal lamina. Scale bar = 10 cm.

Figure 26 Cervical vertebra 9 of Galeamopus pabsti SMA 0011 in right lateral view.

The vertebra is covered by the prezygapophysis of CV 10, because of they were not disassembled during the remounting. Note the vertical subdivisions of the anterior (1) and posterior pneumatic fossae (2), and the anteriorly bifurcated pcdl (3). Abb.: CV, cervical vertebra; pap, parapophysis; pre, pre-epipophysis. Scale bar = 10 cm.

Figure 27 Cervical vertebra 10 of Galeamopus pabsti SMA 0011 in right lateral view.

Note the subdivision of the pvfo (1), the posteriorly facing accessory lamina in the postzygapophyseal centrodiapophyseal fossa (2), the short, subvertical, accessory lamina in the spinodiapophyseal fossa (3), and the anteriorly bifurcated pcdl, with the dorsal branch being oriented nearly vertically (4). Abb.: pap, parapophysis; pre, pre-epipophysis; pvfo, posteroventral fossa. Scale bar = 10 cm.

Figure 28 Posterior cervical vertebra of Galeamopus pabsti SMA 0011 in right lateral and anterior view.

Articulated penultimate CV shaded. Note the internal structure apparent due to damage to the anterior articular condyle (1). Abb.: acdl, anterior centrodiapophyseal lamina; al, accessory lamina; cpol, centropostzygapophyseal lamina; cprl, centroprezygapophyseal lamina; di, diapophysis; ncs, neurocentral synostosis; pap, parapophysis; pcdl, posterior centrodiapophyseal lamina; pl, pleurocoel; podl, postzygodiapophyseal lamina; poz, postzygapophysis; prdl, prezygodiapophyseal lamina; prz, prezygapophysis; pvf, posteroventral flange; spol, spinopostzygapophyseal lamina; sprl, spinoprezygapophyseal lamina. Scale bar = 10 cm.

The parapophyses become slightly anteroposteriorly elongate in CV 3 and 4. These structures project ventrolaterally in all elements, but not to the degree present in Apatosaurinae (Gilmore, 1936; Upchurch, Tomida & Barrett, 2004; Tschopp, Mateus & Benson, 2015). The anterior surface of the parapophyseal ramus, which connects the facets and the anterior condyle, is very distinct, dorsoventrally expanded, and rugose. The fossa on the dorsal surface of the parapophysis is subdivided by a short, oblique ridge in CV 6 and more posterior elements. In CV 9 and 10, the parapophyseal facet is subtriangular, anteroposteriorly elongated, and wider posteriorly than anteriorly.

The ventral surface is hourglass-shaped and narrow in anterior and mid-cervical vertebrae, but becomes relatively wide in more posterior elements. The ventral surfaces of CV 3 and 4 bear a distinct longitudinal keel on their anterior halves, with prominent pneumatic foramina lateral to the keel in CV 3, and less prominent ones in CV 4. In CV 3, a shallow ventral ridge also occupies the posterior end. The ventral surfaces of CV 5 and more posterior vertebrae are concave without any traces of ridges or pneumatic foramina. Posteriorly, the ventral surfaces are bordered by distinct posteroventral flanges. These flanges become rugose ventrally in the posterior cervical vertebrae.

None of the centra are fused with the corresponding cervical ribs. The neurocentral synostosis is closed but visible in the anterior and posterior cervical vertebrae, whereas in posterior mid-cervical vertebrae it is completely open. Where it is closed, the zigzagging neurocentral synostosis is more visible anteriorly than posteriorly (Fig. 29). In the most anterior and posterior elements, the synostosis becomes extremely faint to completely obliterated posteriorly. It lies on top of the centrum, such that the entire pedicels of the neural arches are detached in the unfused elements. The synostosis line is highest in the anterior half and descends anteriorly and posteriorly.

Figure 29 Neurocentral synostosis in CV 5 of Galeamopus pabsti SMA 0011.

Detail of the vertebra in right lateral view. Note the higher degree of fusion in the posterior portion compared to the anterior part (arrows). Abb.: apf, anterior pneumatic fossa; cpol, centropostzygapophyseal lamina; di, diapophysis; pcdl, posterior centrodiapophyseal lamina; podl, postzygodiapophyseal lamina; poz, postzygapophysis; ppf, posterior pneumatic fossa; pvfo, posteroventral fossa.

The neural arch is high in anterior cervical vertebrae, but becomes lower posteriorly. In all elements, it appears very fragile and slender, with very thin but distinct lamination. In posterior cervical vertebrae, the neural arch is somewhat displaced anteriorly, reaching close to the anterior condyle, but being well distant from the posterior edge of the centrum. The displacement reaches its maximum in the posterior-most cervical vertebrae.

The prezygapophyses project anteriorly and slightly dorsally in most elements. Close to the cervico-dorsal transition, they become more elevated. They bear suboval facets in CV 3, with the long axis extending anteroposteriorly. From CV 4 onwards, the facets become subtriangular, with the tip located medially. The facets are transversely convex as in all diplodocines (McIntosh, 1990b; Wilson, 2002; Whitlock, 2011a). Only in CV 5 are they concave, but this appears to be due to taphonomic distortion. In CV 7 and 8, the articular facets are elevated on pedestals, but no transverse sulcus is present posteriorly, unlike in Kaatedocus (Tschopp & Mateus, 2013b). The prezygapophyses cap the PRCDF dorsally, which in CV 5 and 6 is subdivided by a vertical accessory lamina connecting ACDL and PRDL right at the diapophysis. Anteriorly, the prezygapophyses are ventrally supported by the CPRL, which is single in anterior cervical vertebrae. From CV 7 backwards, the CPRL is divided, with one distinct and few short, weak accessory lamina in the PRCDF. The accessory laminae subdividing the PRCDF become stronger in more posterior elements. The anterior-most portion of the lateral surface is marked by distinct pre-epipophyses in CV 4 and more posterior elements, however, they only extend considerably anterior to the prezygapophyseal facet in CV 9 and 10. Posteriorly on the prezygapophyseal process, the anterior portion of the SDF develops a deep, but not well defined fossa in CV 3.

In anterior cervical vertebrae, the SPRL is distinct on the prezygapophyseal process, disappears around midlength of the dorsal portion, and becomes visible again on the spine top. In mid-cervical vertebrae, the SPRL is weak to almost absent on the prezygapophyseal process, as is typical for Diplodocinae (Tschopp & Mateus, 2013b). Thus, the SPRF and SDF of anterior and mid-cervical vertebrae are not distinctly separated from each other at the base of the neural spine. In posterior cervical vertebrae, the SPRL is distinct. Due to a backwards curve of the spine top in anterior cervical vertebrae, the SPRL has a somewhat sinuous appearance in lateral view in these elements. Below the backwards curve, the SPRL extends almost vertically in CV 3 to 6, becomes slightly posteriorly inclined in CV 7 and 8, and anteriorly inclined in CV 9 and more posterior vertebrae. A PRSL is present at the base of the neural arch in unbifurcated spines, which reach back to CV 7.

The diapophysis is entirely located in the anterior half of the vertebra. The transverse processes of SMA 0011 do not form such distinct posterior processes as those present in Kaatedocus (Tschopp & Mateus, 2013b). The diapophysis is supported by distinct ACDL, PRDL, PODL, and PCDL. The ACDL and PRDL are separated along their entire length, a feature typical for apatosaurines, and usually absent in diplodocines (Tschopp, Mateus & Benson, 2015). The PCDL is almost horizontal, and the PODL steeply inclined in CV 3, but in CV 4 and more posterior elements, they approach each other, forming a more acute angle anteriorly. In anterior elements, the PODL and PCDL unite before curving laterally, but more posteriorly they remain separate as the ACDL and PRDL, and the POCDF is therefore extended onto the posterior surface of the diapophysis. In CV 5, two nearly parallel PCDL occur: the ventral one connects to the ventral edge of the diapophysis, whereas the dorsal one joins the PODL anteriorly (Fig. 22). In more posterior elements, the PCDL bifurcates anteriorly (Figs. 23 and 24). In the posterior CV, this bifurcation is very strong, with the more dorsal branch being nearly vertical (Figs. 25–27). The CDF lies directly ventral to the diapophyseal process. In the posterior cervical vertebrae of SMA 0011, a short but stout accessory lamina occupies the posterior portion of the fossa. In CV 10, there is a vertical accessory lamina posterior to the dorsal branch of the PCDL, subdividing the POCDF (Fig. 27). Dorsomedial to the accessory lamina, the POCDF is pierced by a large foramen, such that the POCDF is interconnected with the SPOF (Figs. 27 and 30). A similar state appears to be present in the anterior cervical vertebrae of Dicraeosaurus hansemanni MB.R.4886 (E Tschopp, pers. obs., 2011), a partial mid-cervical vertebra of Suuwassea emilieae ANS 21122 (Harris, 2006b: fig. 8B), and Brontosaurus yahnahpin Tate-001, but in these taxa, the borders of the opening seem to be broken. Fossae at the same location occur in many taxa, including Diplodocus or Supersaurus (Hatcher, 1901; E Tschopp, pers. obs., 2013), but none of them opens up into a large foramen as in SMA 0011 (Fig. 30).

Figure 30 Neural arch foramina in CV 8 of Galeamopus pabsti SMA 0011, in posterodorsal view.

The foramina are highlighted with the semi-transparent overlay. Abb.: bns, bifid neural spine; epi, epipophysis; mt, median tubercle; naf, neural arch foramen; pap, parapophysis; poz, postzygapophysis; ppf, posterior pneumatic fossa; prdl, prezygodiapophyseal lamina; prz, prezygapophysis; pvfo, posteroventral fossa; spol, spinopostzygapophyseal lamina; sprl, spinoprezygapophyseal lamina. Scale bar = 10 cm.

The SDF is of generally simple morphology. In CV 5 and 6, a shallow but dorsally well delimited fossa is located close to the spine summit. In CV 6 and 7, the SDF bears a distinct, dorsoventrally elongate fossa posterolateral to the SPRL, at about mid-height of the metapophysis. From CV 7 backwards, a vertical accessory lamina follows the SPRL posteriorly, as in Diplodocus carnegii CM 84 (Hatcher, 1901). No subfossae are present in the SDF of posterior cervical vertebrae, but in mid- and posterior cervical vertebrae, the SDF becomes clearly delimited dorsally, just below the anteroposterior narrowing of the spine top. CV 10 furthermore bears a stout, slightly anteriorly inclined lamina where the SDF is deepest, but the lamina does not connect to any surrounding lamina.

The neural spine undergoes distinct changes in development and orientation from anterior to posterior. In anterior cervical vertebrae, it is vertical, and dorsoventrally tall, reaching well above the postzygapophyses. The axis, as well as CV 3 and 4 have a distinctly posteriorly curving spine summit, as can also be seen in the corresponding elements of Brontosaurus yahnahpin. There is an abrupt change in height from CV 5 to 6, resulting in a smaller total height of CV 6 compared to CV 5. Such a development has only been described in Dicraeosaurus (Janensch, 1929), but neural spines are often incomplete, where anterior cervical vertebrae have been found (e.g., Diplodocus carnegii CM 84, Apatosaurus louisae CM 3018; Hatcher, 1901; Gilmore, 1936), which makes a thorough assessment of this character difficult. However, SMA 0011 is clearly different from the state in Kaatedocus siberi AMNH 7530 and SMA 0004, in Barosaurus sp. AMNH 7535, and in the indeterminate diplodocine CM 3452, where the anterior cervical neural spines are low, and total vertebral height continuously increases throughout the vertebral column (Tschopp & Mateus, 2013b; Tschopp, Mateus & Benson, 2015). From CV 6 backwards, the cervical neural spines of SMA 0011 decrease in relative height, compared to pedicel height (Table 2), and become anteriorly inclined. Towards the cervico-dorsal transition, neural spine height increases again, such that the posterior cervical vertebrae have highly elevated spine summits. In the first two vertebrae of the transitional block, the spine summits are most strongly anteriorly inclined, and the dorsal-most parts of the neural spines are anteroposteriorly short but elongated dorsoventrally. Bifurcation of the spine is present only from CV 8 backwards, which is more posterior compared to Diplodocus or Apatosaurus (Wedel & Taylor, 2013), but not as posterior as in Barosaurus (McIntosh, 2005). Unbifurcated neural spines slightly expand transversely towards their distal end, similar to the state in Suuwassea emilieae (Harris, 2006b). Posteriorly, the SPOLs are thin but project far posterodorsally, and connect to each other across the spine summit. Therefore, they enclose a distinct, wide and deep SPOF. Elements with bifid neural spines have a median tubercle. The lateral surface of the neural spine summits becomes rugose in posterior vertebrae. CV 9 has a distinct dorsoventral ridge on the medial side of the metapophysis, which connects the summit with the median tubercle, as in Kaatedocus siberi SMA 0004 (Tschopp & Mateus, 2013b).

Following the changing orientation and elevation of the spine, the SPOL also has a quite variable morphology from anterior to posterior cervical vertebrae: the lamina is strongly concave in CV 3, and less so in CV 4, due to the more expressed backwards leaning of the spine top in CV 3. The SPOL is gently curved in CV 5, but strongly concave in CV 6, where it forms a 90° angle. Due to the increasing anterior inclination of the spine, the SPOL becomes more gently concave in CV 7 and more posterior elements. Its posterior portion, where it unites with the epipophysis, is almost horizontal. The epipophysis is well developed in all cervical vertebrae, often overhanging the postzygapophyses. It constitutes the posterior end of the SPOL, and is often pointed. The postzygapophyseal facets are suboval to subcircular in the anterior cervical vertebrae, but become subtriangular more posteriorly, with the tip pointing medially. They are concave and thus face both downwards and outwards. They are ventrally supported by a vertical, single CPOL.

Penultimate and posterior-most cervical vertebra

Preservation. The two posterior-most vertebrae are the second and third elements in the block preserving the cervico-dorsal transition. They are still embedded in matrix, and only the right sides are prepared (Figs. 31 and 32). The diapophysis is not preserved in either vertebra, and the posterior-most element also lacks the right metapophysis and postzygapophysis. The anterodorsal part of the right lateral surface of the centrum of the posterior-most vertebra is reconstructed, including the neurocentral synostosis.

Figure 31 Penultimate cervical vertebra of Galeamopus pabsti SMA 0011 in right lateral view.

Articulated cervical vertebrae shaded. Abb.: acdl, anterior centrodiapophyseal lamina; al, accessory lamina; cpol, centropostzygapophyseal lamina; cprl, centroprezygapophyseal lamina; pap, parapophysis; pcdl, posterior centrodiapophyseal lamina; pl, pleurocoel; podl, postzygodiapophyseal lamina; poz, postzygapophysis; prdl, prezygodiapophyseal lamina; prz, prezygapophysis; pvf, posteroventral flange; spol, spinopostzygapophyseal lamina; sprl, spinoprezygapophyseal lamina. Scale bar = 10 cm.

Figure 32 Posterior-most cervical vertebra of Galeamopus pabsti SMA 0011 in right lateral view.

Articulated penultimate CV and DV 1 shaded. The right metapophysis is lacking, only the medial face of the left one is visible. Note the broken diapophysis that reveals the inner structure. Abb.: cpol, centropostzygapophyseal lamina; cprl, centroprezygapophyseal lamina; di, diapophysis; pap, parapophysis; pl, pleurocoel; poz, postzygapophysis; prdl, prezygodiapophyseal lamina; prz, prezygapophysis; spol, spinopostzygapophyseal lamina; sprl, spinoprezygapophyseal lamina. Scale bar = 10 cm.

Compared to more anterior cervical vertebra, the two posterior-most vertebrae have a considerably taller diapophysis, and less distinct epipophyses. Their centra are opisthocoelous and have an intermediate elongation compared to more anterior cervical vertebrae and the first dorsal vertebra (Table 2). The lateral surface is marked by elongate pleurocoels that occupy the central and anterior portion of the centrum. In the posterior-most element, the pleurocoel is more restricted towards the anterior than in the penultimate one, being almost entirely situated above the parapophysis. The parapophysis lies ventrolateral to the pleurocoels, which extend onto its dorsal face. Posteroventral flanges are present, but become less distinct in the posterior-most centrum. The ventral surface is transversely concave and broad, with a shallow longitudinal ridge located anteriorly.

The neural arch height above the synostoses is more or less equal to centrum length, not counting the condyle (Table 2). As in anterior and posterior cervical vertebrae, the neurocentral synostosis is closed, but still visible in its anterior half. The neural spine is divided. The prezygapophyseal facet is broad, and projects slightly anterior to the condyle in both vertebrae, although the ramus is more vertically oriented in the posterior-most cervical vertebra than in the penultimate one. A weak pre-epipophysis is present, but does not extend beyond the prezygapophyseal facet. The SPRL is strongly concave, due to the strong anterior inclination of the spine top. The PRDL does not contact the ACDL directly, but they are interconnected by a vertical lamina below the diapophysis. The latter is thus slightly elevated above the centrum, and dorsoventrally high. The broken diapophysis of the posterior-most element reveals large open spaces internally that are surrounded by narrow laminae of relatively dense bone tissue. Both the ACDL and the PCDL are only slightly inclined, and connect to the ventral most part of what would have been a dorsoventrally tall diapophyseal ramus before it was broken off. The POCDF is subdivided by a strong, laterally facing, almost vertical accessory lamina, forming a posteroventral branch of the anterior end of the PODL. This differs from more anterior elements, where the accessory lamina in the POCDF faces posteriorly. Unlike the mid-cervical vertebrae, the posterior elements do not have any fenestra connecting the POCDF with the SPOF. The spine summits are anteroposteriorly narrow, and inclined anteriorly, but the inclination decreases in more posterior elements. The lateral surface of the spine is marked by the SDF, which is well delimited dorsally, similar to the state in the first posterior cervical vertebra. From the top of the SDF, the spine of the posterior-most elements forms a narrow anterodorsal projection. The medial surface of the spine is slightly anteroposteriorly convex and smooth.

Dorsal vertebrae (Figs. 33–37; Table 3)

Preservation. The dorsal series of SMA 0011 was found in two parts, with one and a half dorsal vertebrae preserved with the neck vertebrae, and the posterior-most six elements preserved with the appendicular material. A third block including three anterior to mid-dorsal vertebrae with associated dorsal ribs was collected from a position between the two main parts as described above (Fig. 3), and was initially included as part of the specimen. However, these most probably do not belong to the holotype specimen due to different size, preservation, and an apparently older ontogenetic stage (based on neurocentral closure patterns).

Table 3 Measurements of dorsal vertebrae of Galeamopus pabsti SMA 0011 (in mm).

Dorsals	gH	cL	ppL	ppH	ctW	ctH	cdW	cdH	nsH	cL-cd	naH	dvH di	apD di	
DV 1	630		65		120	220  (est)		200	458  (def)	140	250  (est)			
DV 2			53  (def)				185  (def)	180  (def)						
DV 5?	730	157	50	83	160  (def)	195				150  (est)		107	60	
DV 6?	800		106	150		210			545  (est)	160	330			
DV 7?	810		110	160		225			605  (est)	170	345			
DV 8?	900		138	148		225			665	163	330			
DV 9?	900		112	130		212			665	160	290			
DV 10?			140	130		216				175	260  (est)			
Notes.

Abb. apD anteroposterior depth

cdH height condyle

cdW width condyle

cL centrum length

cL-cd centrum length without condyle

comp compressed

ctH height cotyle

ctW width cotyle

def deformed

di diapophysis

dvH dorsoventral height

est estimated

gH greatest height

naH height neural arch (below poz)

nsH height neural spine

ppH pneumatopore height

ppL pneumatopore length

Dorsal vertebrae 1 and 2. Both elements are broken and deformed such that it is difficult to understand their morphology in detail (Figs. 33 and 34). The first dorsal vertebra lacks the right diapophysis and neural spine, such that the medial surface of the left metapophysis is visible in the mount (Fig. 33). The dorsal portion of the centrum and ventral half of the neural arch are crushed, and various pieces of each became intermingled. The second dorsal element preserves a very deformed, anterior half of the centrum, which is not fused with the neural arch (Fig. 34). A part of the neural arch is severely crushed and intermingled with the fractured pieces of the first dorsal vertebra (Fig. 33).

Figure 33 Dorsal vertebra 1 of Galeamopus pabsti SMA 0011 in right lateral view.

Articulated posterior-most CV and partial DV 2 shaded. The right metapophysis of DV 1 is lacking, only the medial face of the left one is visible. The broken right prezygapophysis is present on top of the broken diapophysis. Abb.: DV, dorsal vertebra; ncs, neurocentral synostosis; pap, parapophysis; pl, pleurocoel; prz, prezygapophysis; spol, spinopostzygapophyseal lamina; sprl, spinoprezygapophyseal lamina; vk, ventral keel. Scale bar = 10 cm.

Figure 34 Dorsal vertebral centrum 2 of Galeamopus pabsti SMA 0011.

Centrum shown in dorsal (A), anterior (B), left lateral (C), and internal (D) view. The posterior half of the centrum is lacking due to pre-burial taphonomic processes (possibly scavenging), revealing the inner structure of the centrum in posterior view (D). The grey gradient indicates the position of the pleurocoel. Abb.: nc, neural canal; ncs, neurocentral synchondrosis; pl, pleurocoel. Scale bar = 10 cm.

The dorsal vertebrae are considerably shorter than the posterior-most cervical elements, but remain of about the same length along the dorsal column (not considering the condyle; Table 3). The first dorsal vertebra has a strongly opisthocoelous centrum, whereas DV 2 is only slightly opisthocoelous. Both vertebrae bear distinct pleurocoels on the anterodorsal corner of their lateral sides. These pleurocoels are shorter than the ones of the posterior-most cervical elements, and excavate the neural arch pedicels internally. The position of the parapophysis is difficult to see in both elements, but appears to be still on the centrum in DV 1 (anterodorsal to the pleurocoel), whereas the centrum of DV 2 does not show any traces of a parapophysis. The ventral side of DV 1 is well delimited by posterior ridges between the lateral and ventral surfaces. A broad, but distinct midline ridge marks the anterior half of the ventral side of the first dorsal centrum. The articulation surface of the second centrum for the neurocentral synchondrosis is broad and curved. The neural canal is narrowest at midlength of the centrum. The internal structure of the centrum consists of large chambers, separated from each other by thin, well-defined laminae, which are not symmetrical (Fig. 34D).

The neural arches of the dorsal vertebrae are higher, but more anteroposteriorly compressed, than in the posterior-most cervical elements. The prezygapophysis is relatively short. The SPRL is oriented almost vertically, and no strong anterior inclination of the neural spine is present anymore. The medial side of the first dorsal neural spine is gently convex, and slightly longer anteroposteriorly than in the posterior-most cervical vertebrae. Postzygapophyses are not preserved.

Mid- to posterior dorsal vertebrae (probably DV 5–10). Dorsal vertebra 5 lacks its right neural arch, diapophysis, and spine (Fig. 35). Dorsal vertebra 6 lacks the anterior part of the centrum, the right diapophysis, parapophysis, and prezygapophysis, and the spine top. In dorsal vertebra 7, the right diapophysis, parapophysis, and the spine top are missing. Dorsal vertebrae 8 and 9 lack the right diapophysis and parapophysis. The last dorsal vertebra lacks the neural spine process, whereas the arch below the postzygapophysis, the diapophysis, and the prezygapophyses are preserved (Fig. 36).

Figure 35 Dorsal vertebra 5 of Galeamopus pabsti SMA 0011.

DV 5 shown in posterolateral (A) and right lateral view (B). The element lacks the right half of the neural spine, and is partly mounted in matrix. Grey lines indicate the probable extensions of the right half. Note that the tip of the left diapophysis is reconstructed. The true diapophysis of DV 5 is figured in Fig. 37. Abb.: di, diapophysis; nc, neural canal; pap, parapophysis; pl, pleurocoel; poz, postzygapophysis; prz, prezygapophysis; spdl, spinodiapophyseal lamina; spol, spinopostzygapophyseal lamina; sprl, spinoprezygapophyseal lamina. Scale bar = 10 cm.

Figure 36 Dorsal vertebrae 6 to 10 of Galeamopus pabsti SMA 0011.

Vertebrae shown in right lateral (A), posterolateral (B), and anterolateral view (C). The elements are partly preserved in matrix. Note the open neurocentral synchondrosis in DV 7 to DV 10. Abb.: cpol, centropostzygapophyseal lamina; DV, dorsal vertebra; lspol, lateral spinopostzygapophyseal lamina; pap, parapophysis; pcdl, posterior centrodiapophyseal lamina; pcpl, posterior centroparapophyseal lamina; pl, pleurocoel; podl, postzygodiapophyseal lamina; posl, postspinal lamina; poz, postzygapophysis; prdl, prezygodiapophyseal lamina; prpl, prezygoparapophyseal lamina; prsl, prespinal lamina; spdl, spinodiapophyseal lamina; sprl, spinoprezygapophyseal lamina. Scale bar in A = 10 cm, DV 6 in A and C, and DV 10 in A and B are scaled to the same vertebral height.

The mid- and posterior dorsal centra are short, and generally amphiplatyan to amphicoelous. Only DV 5 shows a weak anterior condyle. The pleurocoel is largest in DV 6–8, occupies the dorsal half of the centrum and extends slightly onto the pedicels, below the neurocentral synchondrosis. The ventral surface is convex, and not well separated from the lateral side. The centrum is slightly shorter ventrally than at mid-height. In DV 6 and 7, a zigzagged line marks the neurocentral synostosis at the dorsal edge of the centrum. Dorsal vertebrae 8–10 have the centra and neural arches detached, but no obvious articulation surface is visible on either element, indicating that that closure has initiated but not entirely completed, such that centra and neural arches could be detached easily. The neural arch is high, with highly elevated postzygapophyses, resulting in longer pedicels than neural spines in at least DV 5–8. Pre- and postzygapophyses are on more or less a horizontal line. The pedicels below do not show a strong lamination, but the ACPL, PCDL, and CPOL can be well distinguished. Dorsal vertebrae 6–9 furthermore show a weakly developed PCPL. An accessory lamina can be found in DV 7, connecting the PCDL with the PODL, and in DV 8 between the PRPL and the PRDL. Only a single hyposphene is visible (in DV 5), relatively long dorsoventrally, and transversely expanded ventrally, resulting in a high and narrow trapezoid. The width of the ventral end (39 mm) is slightly more than twice the minimum width of the hyposphene (16 mm). The posterior surface of the hyposphene is transversely concave. It is ventrally supported by a single, vertical lamina. The parapophysis lies at mid-height on the pedicels in DV 6, at two thirds in DV 7 and at three fourths in DV 8. More posteriorly, the parapophysis seems to have been attached to the prezygapophysis.

A single transverse process is preserved completely (the left of DV 5; Fig. 37). It projects more or less straight laterally (although it was reconstructed as being strongly inclined, see Fig. 35), curving very gently ventrally towards its distal tip. The process is widest dorsally, and dorsoventrally concave both on its anterior and posterior sides. The diapophyseal facet points ventrolaterally and is strongly expanded posteriorly.

The spine is relatively low in DV 5–8, and only in DV 9 and probably 10 does it exceed the pedicel height. The spines are situated above the posterior-most portion of the centrum, and are vertically oriented. This differs from the strongly anteriorly inclined posterior dorsal neural spines of Diplodocus (Hatcher, 1901; Gilmore, 1932). The SPRL is vertical in DV 6, strongly dorsoventrally convex in DV 7 and 8, and slightly convex in DV 9. The SPDL is short and only expressed at its ventral end. Dorsally it merges with the SPOL, which extends onto the lateral surface of the spine. The POSL, or possibly medial SPOL, is straight and vertical. Due to the preservation and mounting (partly embedded in matrix), it cannot be distinguished at this point how far back the bifurcation proceeds. The last definitively bifid neural spines are present in DV 5.

Ribs

Cervical ribs (Figs. 38–49; Table 4). The cervical ribs are thin, fragile elements. None of them are fused with their respective centra. They are composed of a rib shaft, an anterior process, and the capitulum and tuberculum. The ribs are concave internally, with a lamina connecting the tuberculum with the capitulum internally, producing two separate fossae anteriorly and posteriorly.

The axial cervical rib has almost no tuberculum and is thus a straight, elongate, and dorsoventrally compressed sheet of bone, which becomes slightly higher around midshaft but tapers again posteriorly (Fig. 38). The capitulum is not offset from the posterior shaft, and faces anteromedially. The capitular facet is much longer than wide, such that it articulates with both the axial parapophysis, and to a small extent also with the posteroventral projections of the atlas.

Table 4 Measurements of cervical ribs of Galeamopus pabsti SMA 0011 (in mm).

CV	Side	apL	apL ap	minH	minW	maxL  cap	apL  caf	tW  caf	maxL  tub	apL  tuf	tW  tuf	Comments	
CR 2	R	197		8	13	21	24	9					
CR 3	L	250	11	5	15	27	24	17				Anterior tip missing, apL, apL ap incomplete	
	R			5	16								
CR 4	L	250	57	6	15	42	38	26				Central portion missing, apL incomplete	
	R	283	49	6	18	38	40	25	96	12	15		
CR 5	L		84			64	47	25				Incomplete	
	R	368	88		22	60	52	27	119	7	19	Deformed: apL, maxL tub, apL tuf too short	
CR 6	L		80			80	62	30				Anterior tip and posterior process lacking: apL ap too short	
	R					78	70	27				Anterior tip and posterior process lacking	
CR 7	L		106			89	67	39				Anterior tip and posterior process lacking: apL ap too short	
	R		116			100	67	36				Posterior process and tuberculum lacking	
CR 8	R	435	88		26	109	68	38	102			Anterior tip and tubercular facet missing: apL, apL ap, maxL tub too short	
CR 9	R		145			128	80	38				Posterior process and tuberculum lacking	
CR 10	R	382	50	5	43	133	81	39	124			Anterior and posterior tip and tubercular facet missing: apL, apL ap, maxL tub too short	
CR 11	R		117			137	79	35	172	19	25	Anterior and posterior process missing: apL ap too short	
	L		143			140	80	33	137			Posterior Process and tubercular facet missing: maxL tub incomplete	
CR ?12												Mounted, could not be measured	
CR ?13	L	465	117	10	35	178	75	43				Anterior and posterior tip and tubercular facet missing: apL, apL ap, maxL tub too short; minH, minW measured at posterior-most preserved point	
	R	400	144		34	159	88	37	152	52	26	Anterior and posterior tip missing: apL, apL ap too short; minW measured at posterior-most preserved point	
CR ?14	L	368	105			180	76		210	33	37	Caf incomplete, posterior process compressed transversely	
	R					158	64	60	220	49	23	Anterior tip and posterior process lacking	
Notes.

CR 2 is the first rib in the column, and attaches to both atlas and axis (see text). CR 13 and 14 are the two posterior-most ribs as described in the text.

Abb. ap anterior process (apL measured from transverse lamina between cap and tub to anterior tip)

apL anteroposterior length (measured in a straight line)

caf capiular facet

cap capitulum; maxL maximum length (measured from capitular or tubercular facets to ventrolateral edge of CR)

minH minimum dorsoventral height (around midlength of posterior shaft)

minW minimum transverse width (around midlength of posterior shaft)

tub tuberculum

tuf tubercular facet

tW transverse width

Figure 37 Left transverse process of DV 5 of Galeamopus pabsti SMA 0011.

The process is shown in dorsal (A), posterior (B), anterior (C), and ventral (D) view. Abb.: dif, diapophyseal facet. Scale bar = 10 cm.

Figure 38 Left axial rib of Galeamopus pabsti SMA 0011.

The rib is shown in lateral (A), ventral (B), medial (C), and dorsal (D) view. Abb.: caf, capitular facet. Scale bar = 10 cm.

Figure 39 Cervical ribs 3 of Galeamopus pabsti SMA 0011.

The left (A–D) and right (E–H) ribs are shown in dorsal (A, E), medial (B, F), ventral (C, G), and lateral (D, H) views. Abb.: ap, anterior process; cap, capitulum; tub, tuberculum. Scale bar = 10 cm.

Figure 40 Cervical ribs 4 of Galeamopus pabsti SMA 0011.

The left (A–C) and right (D–F) ribs are shown in lateral (A, D), dorsal (B, E), and medial (C, F) views. It is unclear how long the missing portion of the left rib shaft was, the space is thus only indicative. Note the transverse lamina connecting capitulum and tuberculum (1). Abb.: ap, anterior process; cap, capitulum; tub, tuberculum. Scale bar = 10 cm.

Figure 41 Right cervical rib 5 of Galeamopus pabsti SMA 0011.

The rib is shown in dorsal (A), medial (B), and ventromedial (C) views. The spur on the distal end of the rib shaft is a support for mounting. Abb.: ap, anterior process; cap, capitulum; tub, tuberculum. Scale bar = 10 cm.

Figure 42 Right cervical rib 6 of Galeamopus pabsti SMA 0011.

The rib is shown in dorsal (A), medial (B), and ventral (C) views. Note the pneumatic foramen on the capitulum (1). Abb.: cap, capitulum. Scale bar = 10 cm.

Figure 43 Right cervical rib 7 of Galeamopus pabsti SMA 0011.

The rib is shown in dorsal (A), medial (B), and ventral (C) views. The tuberculum is not preserved. Abb.: cap, capitulum. Scale bar = 10 cm.

Figure 44 Right cervical rib 8 of Galeamopus pabsti SMA 0011.

The rib is shown in dorsal (A), medial (B), and ventral (C) views. The tuberculum is only partly preserved. Abb.: tub, tuberculum. Scale bar = 10 cm.

Anterior to mid-cervical ribs are longer than their corresponding centra, unlike the situation in Apatosaurus louisae CM 3018 (Gilmore, 1936), but they overlap only a small portion of the following vertebra. The anterior process is distinct but very short in CR 3, and pointed in CR 3–5 (Figs. 39–41). This process becomes very broad and rounded anteriorly in mid- and posterior cervical ribs (Figs. 42–49). At the base of the anterior process, mid- and posterior cervical ribs bear a dorsal lamina, which connects the capitulum with the tubercular edge of the anterior process (Fig. 45). Thereby, it forms the anteromedial rim of a deep triangular fossa, which is otherwise bordered by a transverse lamina between capitulum and tuberculum and the lateral margin of the anterior process. This fossa is further subdivided by a second oblique ridge, parallel to the first, in posterior cervical ribs (Fig. 47B). The tuberculum is posteriorly inclined in anterior cervical ribs, and triradiate in cross-section at midlength. The three axes are oriented anteriorly, posteriorly, and medially. The tubercular facet is generally wider than long. The capitulum bears a pneumatic foramen dorsally, posterior to the origin of the lamina connecting the capitulum with the tuberculum (Fig. 42). The capitular facet is ovoid in CR 3, with the wider end anteriorly. It becomes subrectangular to reniform in more posterior ribs, with the longer axis being oriented anteroposteriorly, and the sometimes concave margin being the dorsolateral one. The ventral surface of the cervical rib is marked by striations (Fig. 46), probably for muscle or tendon insertions.

Figure 45 Right cervical rib 9 of Galeamopus pabsti SMA 0011.

The rib is shown in dorsolateral (A), dorsomedial (B), ventromedial (C), and ventrolateral (D) views. The rib shaft is not preserved. Note the oblique lamina at the base of the anterior process (1). Abb.: tub, tuberculum. Scale bar = 10 cm.

Figure 46 Right cervical rib 10 of Galeamopus pabsti SMA 0011.

The rib is shown in dorsal (A), medial (B), and ventral (C) views. The anterior process is only partly preserved. Note the striations on the ventral surface of the rib (1). Abb.: cap, capitulum; tub, tuberculum. Scale bar = 10 cm.

Figure 47 Cervical ribs 11 of Galeamopus pabsti SMA 0011.

The left (A–C) and right (D–E) ribs are shown in medial (A, D), dorsomedial (B, E), and dorsolateral (C, E) views. The rib shafts are not completely preserved (the right one is reconstructed). Note the laminae subdividing the anterior process (1). Scale bar = 10 cm.

The two posterior-most cervical ribs (Figs. 48 and 49) bear progressively shorter anterior processes, compared to more anterior cervical ribs. The dorsal oblique lamina disappears, and also the transverse lamina connecting capitulum and tuberculum becomes less pronounced. The angle between capitulum and tuberculum widens considerably, approaching 90° in the posterior pair. The posterior process shortens and tapers strongly. A distinct longitudinal ridge marks the ventral surface, as in anterior dorsal ribs. One right posterior cervical rib (field number M 6/16-3) has a pronounced, anteriorly projecting spur close to the origin of the transverse lamina on the capitulum, which might be an ossified tendon insertion, and is absent on the left element of the pair (Fig. 48). The pneumatic fossa on the capitulum is reduced in the first pair of posterior-most cervical ribs, and totally absent in the second pair. The capitular facet becomes ovoid again, resembling the shape of the facet in CR 3. In the posterior-most pair of cervical ribs, the capitular facet is nearly circular, and supported by a strong, subtriangular capitular neck. The tubercular facet is longer than wide, and thus resembles rather dorsal ribs than cervical elements. In the posterior-most cervical ribs, the posterior process does curve slightly downwards, and not strictly posteriorly as in more anterior elements.

Figure 48 Cervical ribs ?13 of Galeamopus pabsti SMA 0011.

The left (A–C) and right (D–F) ribs are shown in lateral (A, D), dorsal (B, E), and medial (C, F) views. The rib shafts are not completely preserved. Note the anterior projection on the capitulum of the right CR 13 (1). Abb.: cap, capitulum. Scale bar = 10 cm.

Figure 49 Cervical ribs ?14 of Galeamopus pabsti SMA 0011.

The left (A–C) and right (D–F) ribs are shown in lateral (A), dorsal (B), medial (C, E), dorsomedial (D), and ventral (F) views. The capitulum of the left rib is dorsoventrally compressed. The right rib has a broken anterior process and an incomplete and strongly distorted posterior shaft. Note the short, tapering posterior shaft (1), and the circular capitular facet (2). Abb.: ap, anterior process; cap, capitulum; tub, tuberculum. Scale bar = 10 cm.

Table 5 Measurements of dorsal ribs of Galeamopus pabsti SMA 0011 (in mm).

Element	side	pdL	maxL  cap	apL  caf	dvH  caf	maxL  tub	apL  tuf	tW  tuf	Distance  between  facets	Angle  between  cap and tub	
DR 1	R		228	16	50	173	19	53	202	90	
	L		205	23	51	189	37	53	117	50	
DR 2	R	1,105	250	23	81	170				90	
	L	1,130	205	35	76	170	29	80	136		
DR 3	R?	1,025									
DR 4											
DR 5	R		215			125				70	
DR 6	R	1,052									
	L		250	21		150				70	
DR 7	R	1,150									
	L		340	21	109	160	42	73	230	60	
DR 8	R	1,250									
	L		325	32		150				60	
DR 9	R										
	L										
DR 10	L		170			80					
Element	side	maxD  midshaft	minD  midshaft	maxD  dist	minD  dist	Comments	
DR 1	R	38	16				
	L	40	20	17	11	apL caf estimated; distance and angle between facets deformed	
DR 2	R	48	28	50	11	pdL along curvature, preserved length; maxL tub is preserved length; maxD midshaft, dist estimated	
	L	49	32	39	9	pdL along curvature, preserved length; cap deformed; distal end estimated	
DR 3	R?	52	23	75		pdL along curvature, preserved length	
DR 4							
DR 5	R					maxL cap & tub, preserved length	
DR 6	R	65	31			pdL along curvature, preserved length; maxL tub is preserved length	
	L	80	26			max L tub: preserved length; midshaft measurements taken more distally	
DR 7	R	74	26			Only shaft preserved, pdL preserved length	
	L	66	38				
DR 8	R	44	34			pdL along curvature, preserved length	
	L	55					
DR 9	R	42	25				
	L						
DR 10	L					max L cap tub preserved lengths	
Notes.

Abb. apL anteroposterior length (measured in a straight line)

caf capiular facet

cap capitulum

dist distal

dvH dorsoventral height

maxD maximum diameter

maxL maximum length

minD minimum diameter

pdL proximodistal length

tub tuberculum

tuf tubercular facet

tW transverse width

Figure 50 Right dorsal rib 1 of Galeamopus pabsti SMA 0011.

The rib is shown in anterior (A), medial (B), and posterior (C) views. The rib shaft is slightly incomplete at its distal end. Note the ridges on the anterior (1, 2) and posterior (3) surfaces. Abb.: cap, capitulum; tub, tuberculum. Scale bar = 10 cm.

Figure 51 Right dorsal rib 2 of Galeamopus pabsti SMA 0011.

The rib is shown in posterior (A) and anterior (B) views. Note the longitudinal ridge on the anterior surface (1), which is wider than the one in DR 1. Abb.: tub, tuberculum. Scale bar = 10 cm.

Dorsal ribs (Figs. 50–56; Table 5). Several ribs have been recovered associated with the dorsal series, but whereas the sequence from anterior to posterior appears relatively clear, based on the quarry position, the exact position of the single elements can only be confidently determined for some elements.

Figure 52 Right dorsal rib ?3 of Galeamopus pabsti SMA 0011 in lateral view.

The rib head is not preserved. Note the distal expansion for the attachment of sternal cartilages (1). Scale bar = 10 cm.

Figure 53 Right dorsal rib ?5 of Galeamopus pabsti SMA 0011.

The rib is shown in posterior (A) and anterior (B) views. Only the rib head is preserved. Note the absence of an oblique ridge crossing the posterior surface. Abb.: tub, tuberculum. Scale bar = 10 cm.

Figure 54 Posterior right dorsal ribs of Galeamopus pabsti SMA 0011.

The ribs are estimated to be dorsal ribs 6 in posterior (A), 7 in lateral (B), 8 in anterior (C) and 9 in posterior (D), view. Rib head and distal tips of ribs 6, 7, and 9 are not preserved. The rib head of DR 6 is reconstructed. The rib 8 is preserved in three parts (fractures indicated by arrows). Abb.: tub, tuberculum. Scale bar = 20 cm.

DR 1 has a capitulum and a tuberculum which stand in a right angle to each other (Figs. 50 and 55). The anterior surface of the rib head bears a distinct, narrow, proximodistal ridge, which originates from the tubercular facet and extends in a nearly straight line distally onto the rib shaft (Fig. 50), where it fades out. At the base of the capitulum, a broader, slightly less distinct ridge separates from the narrow one and curves for a short distance onto the anterior surface of the capitulum, joining its proximal edge at about midlength (Fig. 50). Both the tubercular and capitular facets are anteroposteriorly compressed, rugose articular surfaces. The posterior surface of the capitulum is flat. The posterior surface of the tuberculum is marked by two longitudinal ridges: a longer, narrower medial one, and a shorter and broader lateral one (Fig. 50). Together, they form a distinct proximal fossa just below the tubercular facet, which fades out more distally towards the rib shaft. The rib shaft has a V-shaped cross-section at its base and flattens distally. The anterior side changes from being distinctly convex (due to the presence of the proximal longitudinal ridge) to even slightly concave once the ridge disappears. The distal end of the shaft tapers nearly to a point, and is marked by a sharp longitudinal ridge on the posterior surface, which extends from below midshaft to the tip and thus creates a distinctly triangular cross-section, with an inverted orientation compared to the cross-section at the base of the shaft.

Figure 55 Left dorsal ribs 1, 2, and 6–10 of Galeamopus pabsti SMA 0011.

The ribs are shown in anterior (ribs 1, 2, 8–10) and posterior (ribs 6, 7) view. Scale bar = 10 cm.

Dorsal rib 2 has a much shorter tuberculum, which is mainly due to the fact that the bony shelf connecting capitulum and tuberculum is more extensive in this element compared to the first dorsal rib (Fig. 51). The longitudinal ridge on the anterior surface of DR 2 is less pronounced and wider than in DR 1, and no perpendicular ridge occurs at the base of the capitulum. Also the short, longitudinal ridges on the posterior surface of the tuberculum of DR 1 do not occur on DR 2, so that the rib head is uniformly concave posteriorly. When articulated with the dorsal vertebra, the shaft of DR 2 curves backwards and tapers until about midlength. From here, the anterior and posterior edges remain subparallel, just to minimally expand distally towards the distal-most tip.

A probable DR 3 preserves only the shaft, which is wider and more triangular than circular in cross-section. The distal end is expanded (Fig. 52).

More posterior ribs continue the trends observed from DR 1 to DR 3. The shape of the rib head changes such that the capitulum projects obliquely dorsomedially instead of perpendicular to the long axis of the shaft. The capitular facet becomes gradually stronger throughout the series, whereas the tuberculum becomes shortened. The rib head thus has a subtriangular shape in axial view in more posterior elements. In at least the last three dorsal ribs (but maybe additional posterior dorsal elements are lacking), the capitulum curves dorsally at its end, such that the capitular facet comes to face dorsomedially instead of more strictly medially as in more anterior ribs. The relatively thin sheet of bone between capitulum and tuberculum remains flat internally throughout the entire series (contrary to the state in most other diplodocines, in which this area is marked by an oblique accessory ridge; Tschopp, Mateus & Benson, 2015). None of the ribs bear pneumatic foramina. The shafts are marked by a longitudinal groove on the lateral edge in mid- to posterior dorsal ribs, and have an ovoid to slightly subtriangular cross-section, with a transversely oriented long axis, and a slightly more angular posterior surface. The last three or more dorsal ribs decrease significantly in shaft width, compared to more anterior elements, and obtain a subcircular cross-section similar to DR 1.

The left dorsal rib 2 bears bite marks on its distal end (Fig. 56). The bite marks are eleven parallel, slightly curved grooves on the external side of the rib, which extend from the posterior edge anteroventrally. The distance between the marks on the posterior edge varies from 16 to 26 mm, with a mean distance of 20.75 mm.

Sternal ribs (Figs. 57 and 58). Several morphotype C elements (sensu Tschopp & Mateus, 2013a) were recovered associated with SMA 0011. They are rod-like, narrow bones (Fig. 57). Some have a rather circular, and others a laminar cross-section, and all have smooth margins. A single, flattened morphotype E element (field number M5/4-2) is expanded on one side, where it has rugose margins (Fig. 58). No additional information can be gleaned to date that would help to confirm or discard the interpretation of Claessens (2004) and Tschopp & Mateus (2013a) that these elements are sternal ribs.

Appendicular skeleton

Terminology. The scapulacoracoid is described as if it were oriented horizontally, with the scapular blade pointing posteriorly. Manus and pes are described as if the digits were held completely vertically, and arranged in a single line perpendicular to the axial column (following Bonnan, 2001). Therefore, the directional term “posterior” is used interchangeably with “palmar” in the manus and “plantar” in the pes.

Forelimb (Figs. 59–67; Table 6)

Scapulae. Both scapulae lack the dorsal part of the acromion and of the distal end of the blade (Fig. 59). The acromion and the blade form an acute angle, but the acromial ridge is only very slightly developed. The lateral surface anterior to the acromial ridge is concave. Medially, the acromion is concave. The glenoid surface is transversely concave and faces slightly more medially than laterally. The surface is widest anterodorsally, close to the articulation with the portion of the glenoid on the coracoid, and tapers posteroventrally. The ventral edge is mostly straight, and does not bear a triangular process as present in some Camarasaurus specimens, or Dystrophaeus (Osborn & Mook, 1921; McIntosh, 1997). The distal end of the blade is slightly expanded ventrally as in Brontosaurus excelsus YPM 1980 (Upchurch, Tomida & Barrett, 2004). The dorsal, or acromial edge of the scapula is much more concave than the ventral one, due to the stronger extensions of the dorsal portion of the acromion and the indicated, wider distal expansion of the shaft, which starts more anteriorly on this edge than on the ventral one. No oval rugose tubercle is present on the base of the shaft, unlike in Brontosaurus excelsus YPM 1980 (Upchurch, Tomida & Barrett, 2004), although a slightly elevated structure occurs in the left scapula.

The left scapula bears distinct bite marks medially along the broken posterodorsal edge of the acromion. There are at least ten subparallel grooves oriented perpendicular to the broken edge, and varying in length from 19 to 73 mm. Also on the lateral side, the left scapula bears short, subparallel grooves, which mark the slightly elevated structure at the base of the shaft. Seven grooves are present. Given that this structure was probably the attachment site for soft tissue (the M. scapulohumeralis cranialis, according to Remes, 2008), the theropod might have bitten only there in order to detach the muscle from the bone.

Coracoid. The right coracoid is preserved, which is only observable in lateral view due to the way it is mounted. The coracoid is somewhat tear-drop shaped (Fig. 59), with a concave anterodorsal edge, and a strongly, continuously convex, narrow dorsal margin, unlike the squared coracoids of apatosaurs (Riggs, 1903; Bakker, 1998). The coracoid foramen is completely enclosed, but the coracoid is not fused with the scapula. The bone is gently convex dorsoventrally. It curves slightly medially at its anterior margin. No distinct notch is present anterior to the glenoid surface. The glenoid is strongly transversely expanded at its center, and tapers posterodorsally and anteroventrally. The articular surface is barely visible in lateral view. The glenoid surface and the articulation surface with the scapula enclose an angle of about 155°.

Figure 56 Bite marks on the left dorsal rib 2.

The bite marks (arrowheads) occur on the distal end of DR 2, shown in lateral view. Scale bar = 5 cm.

Figure 57 Sternal rib of Galeamopus pabsti SMA 0011.

This bone can be attributed to morphotype C of Tschopp & Mateus (2013a). Orientation unknown, both ends lacking. Scale bar = 10 cm.

Figure 58 Sternal rib of Galeamopus pabsti SMA 0011.

This bone can be attributed to morphotype E of Tschopp & Mateus (2013a). Orientation unknown. Scale bar = 10 cm.

Humeri. The humeri are both complete but slightly compressed anteroposteriorly, the right humerus more so than the left (Figs. 60 and 61). The humeri are widely transversely expanded at their proximal ends, both laterally and medially. The distal ends are expanded as well, but less so. The proximal portion of the anterior side is concave transversely. A small, rugose tubercle marks this concavity, as in most diplodocids (Tschopp, Mateus & Benson, 2015), but it is more laterally positioned compared to the apatosaur AMNH 6114 or Galeamopus hayi HMNS 175 (Fig. 60). The deltopectoral crest of G. pabsti SMA 0011 does not extend to midshaft (Table 6). Its distal end is distinct and follows the lateral margin. It is not transversely expanded as would be typical for titanosaurians (Wilson, 2002; Curry Rogers, 2005). The lateral surface of the crest is concave anteroposteriorly, but this depression is probably exaggerated taphonomically. The humeral head is well offset from the shaft and centrally located. The posterior surface is transversely convex in its proximal half, but becomes concave distally, where it develops a shallow intercondylar groove. Two ridges mark the distal end anteriorly, indicating the extensions of the medial and lateral condyles. The ridges are relatively well visible and extend proximally up the shaft (Fig. 61B). The medial condyle is much wider than the lateral one.

Ulna. The ulna lacks the proximal-most portion of the anterior arm of the condylar processes. The bone is strongly anteroposteriorly compressed in its proximal half (Fig. 62). It is generally slender, with a triradiate proximal end. The anterior arm is considerably longer than the lateral one, even though this is enhanced due to compression. The ulna has concave posterolateral and posteromedial surfaces. The lateral arm is somewhat wider than the anterior one. The distal part of the anterior surface bears two strong and elevated, longitudinal ridges. They both taper distally and proximally, and have a smooth surface. Proximally, the more lateral of the two ridges extends above midlength (Fig. 62D). Distally, the more medial ridge is more pronounced, reaching the distal articular surface. The distal end is expanded medially and somewhat anteroposteriorly. The articular surface is subrectangular in outline.

Radius. The radius is complete, but its proximal end is compressed, and the distal end taphonomically sheared such that the entire bone appears sigmoid (Fig. 63). The proximal articular surface has thus a narrow, ellipsoid outline, but would probably be slightly more subcircular if undeformed. The shaft is subrectangular in cross-section. As in the ulna, also the distal end of the radius is slightly expanded transversely. The posterior surface bears two longitudinal ridges on its distal portion for the articulation with the ulna. The lateral ridge is stronger and marks the posterolateral edge of the radius. It extends from the distal articular surface about one third up the shaft. The more medial ridge is weakly developed and shorter. It does not reach the distal articular surface. The distal surface is subrectangular, with convex medial and lateral margins and weakly concave anterior and posterior borders. The lateral half of the distal articular surface is beveled.

Table 6 Forelimb measurements of Galeamopus pabsti SMA 0011, in mm.

	pdL	acL	min  sD	max  sD	min  apD	ptW	dtW	dpcL	aprL	lprL	papD	dapD	dvH	scaL	max  tW gl	dvH  gl	Comments	
Scapula, R	1,375	620  (inc)	178	240  (inc)													Max sD measured at distal-most preserved end	
Scapula, L	1,370	620   (inc)	180	295		130									121	222	Distal end incomplete, pdL preserved straight length; max sD measured at distal-most preserved end	
Coracoid, R	379												515	290	105	200		
Humerus, L	870		174		70  (est)	474	309	310										
Humerus, R	893		180		75	480	328	325			89	104						
Radius, L	601		89		44	165	140				87	74					Min apD measured laterally	
Ulna, L	628		73		78	200	123		275	177	165	95					Proximal surface deformed, ptW, papD estimated; aprL, lprL measured as preserved; papD measured from posterior-most point of radial fossa to posterior-most point of olecranon	
Carpal, L	77			137	39						78						Min apD measured medially; dapd measured laterally (is equal to max apD	
Mc I, L	194		62		46	79	112				75	57					Min apD, dapD measured medially; papD measured laterally	
Mc II, L	232		56		44	100	103				69	63					Min sW measured dorsally; min apD, dapD measured medially	
Mc III, L	242		43		41	47	85				90	48						
Mc IV, L	220		44		50	60	87				110	70					dapD estimated	
Mc V, L	193		46		47	76	87				114	62					Distal end reconstructed	
phm I-1, L	56		66	88	50	86	77				71	71					Min sW measured at midheight; max sW measured palmarly; dapD measured medially	
phm II-1, L	90		94		27	93	110				57	33					Min apD measured centrally at midshaft; dapD measured laterally	
phm III-1, L																		
phm IV-1, L	53				21	93					45	24						
phm V-1, L	39					83					49	9						
Manual ungual, L	193/254					45					114						pdl measured straight/along dorsal curvature	
phm II-2, L	20			42	31												Min apD = max apD	
Notes.

Abb. acL acromion length

apD anteroposterior depth

aprL anterior process length

dapD distal anteroposterior depth

dpcL length deltopectoral crest

dtW distal transverse width

dvH dorsoventral height

gl glenoid

lprL lateral process length

papD proximal anteroposterior depth

pdL proximodistal length

ptW proximal transverse width

scaL scapula-coracoid articular length

sD shaft diameter

sW shaft width

tW transverse width

Figure 59 Right scapula and coracoid of Galeamopus pabsti SMA 0011 in lateral view.

Lacking parts indicated with dashed lines. Abb.: acr, acromion ridge; co, coracoid; cof, coracoid foramen; gl, glenoid; sc, scapula. Scale bar = 20 cm.

Figure 60 Left humerus of Galeamopus pabsti SMA 0011.

The humerus is shown in anterior (A), lateral (B), posterior (C), medial (D), proximal (E), and distal (F) view. Cross-sections of the disassembled pieces are shown as well. Note the rugose tubercle on the anterior surface of the proximal half of the shaft (mt). Abb.: dpc, deltopectoral crest; hh, humeral head; mt, median tubercle. Scale bar = 20 cm.

Figure 61 Right humerus of Galeamopus pabsti SMA 0011.

The humerus is shown in posterior (A) and anterior (B) views. Abb.: dpc, deltopectoral crest; hh, humeral head; icg, intercondylar groove; lr, lateral ridge; mr, medial ridge. Scale bar = 10 cm.

Figure 62 Left ulna of Galeamopus pabsti SMA 0011.

The ulna is shown in proximal (A), anterior (B), medial (C), posterior (D), lateral (E), and distal (F) views. Abb.: ap, anterior process; lp, lateral process; lr, lateral ridge; mr, medial ridge. Scale bar = 10 cm.

Figure 63 Left radius of Galeamopus pabsti SMA 0011.

The radius is shown in proximal (A), anterior (B), medial (C), posterior (D), lateral (E), and distal (F) views. Note the laterally beveled distal articular surface (1). Abb.: lr, lateral ridge; mr, medial ridge. Scale bar = 10 cm.

Carpal. The carpal is an irregular, relatively thick element (Fig. 64). It does not bear distinct articular surfaces, and was found slightly disarticulated, such that an orientation of the carpal within the manus was not possible to definitely confirm. Only one element was found. The entire bone is relatively rugose and was found between the radius and mtc I–III. This is the same arrangement as found in the articulated manus of the indeterminate diplodocine WDC-FS001A (Bedell & Trexler, 2005; Tschopp, Mateus & Benson, 2015), but different from apatosaurines, where the carpal overlies mtc II–IV (CM 3018 and UW 15556; Hatcher, 1902; Gilmore, 1936). If the orientation of the carpal did not change during diagenesis, the surface articulating with the radius is strongly convex transversely, but some abrasion has occurred, and the internal bone structure is visible both medially and laterally. It is therefore possible that the complete element would be more block-like in shape, as known from other diplodocine specimens (WDC-FS001A, Bedell & Trexler, 2005). It is relatively narrow anteroposteriorly at its medial end. The lateral side is about double the anteroposterior length, thanks to a distolateral, posteriorly projecting process. Anterior and posterior surfaces are fairly smooth. Distally, there are no distinct articulation surfaces for the metacarpals, unlike the state in Camarasaurus (Tschopp et al., 2015). The carpal of SMA 0011 is taller proximodistally than the elements known from the apatosaurines CM 3018 and UW 15556 (Hatcher, 1902; Gilmore, 1936).

Figure 64 Left carpal of Galeamopus pabsti SMA 0011.

The carpal is tentatively oriented according to the situation in the quarry. The carpal is shown in probable proximal (A), anterior (B), medial (C), posterior (D), lateral (E), and distal (F) views. Scale bar = 5 cm.

Metacarpals. All metacarpals are complete and were found articulated. Metacarpal I was recovered flipped 180° such that the distal articular surface was at the level of the proximal articular surface of the remaining metacarpals. This displacement indicates that mtc I was not rigidly included in the columnar metacarpal structure adapted for weight-bearing. Given that digit I bears a large ungual, it did not have a primarily graviportal role, and was probably therefore not so strongly bound to the other metacarpals. The metacarpals of SMA 0011 are relatively elongate bones (Fig. 65), but less so than in Camarasaurus (Tschopp et al., 2015). Metacarpal III is the longest, followed by mtc II, IV, I, and V (Table 6). Metacarpal I and II have subrectangular to trapezoidal proximal articulation surfaces, contrasting with triangular ones in mtc III and IV.

Figure 65 Left metacarpals I to V of Galeamopus pabsti SMA 0011.

The metacarpals are shown in anterior, medial, palmar, lateral, proximal and distal view. Digits are indicated on the left with roman numbers. Portions covered by a semitransparent, white layer in proximal and distal view are visible, but do not belong to the articular surface. Scale bar = 10 cm.

Metacarpal I is relatively stout. The proximal surface is concave anteroposteriorly and flat transversely. It is slightly deeper laterally than medially. The lateral edge of the articular surface is strongly concave, whereas the medial one is somewhat convex in proximal view. The posterior surface bears two small but distinct nutritional foramina on the distal half. The distolateral portion of the shaft is crushed, resulting in a triangular lateral surface. The distal condyles are well separated from each other and anteroposteriorly convex. The lateral condyle is much longer proximodistally than the medial one. This results in a strongly inclined distal surface, such that the proximal phalanx projects posteromedially (and distally) in the articulated manus.

Metacarpal II has very distinct, straight anteromedial and anterolateral edges. The proximal and distal ends are slightly expanded in all directions. The proximal articular surface is wider anteriorly than palmarly and slightly convex. The shaft is thicker medially than laterally. The proximal portions of both the medial and lateral surfaces are concave, laterally more than medially. A slightly rugose, longitudinal ridge separates the medial from the palmar surface, and extends distally from the proximal end for about two thirds the length of mtc II. The distal surface slightly curves into the anterior surface. Its lateral and medial condyles are only visible in distal and posterior view. The medial condyle is larger than the lateral one.

Metacarpal III is the most elongate element of the manus. The proximal articular surface is subtriangular to ovoid. No distinct transition from the anterior onto the medial surface occurs on mtc III. The anterior and palmar faces unite laterally at a distinct ridge. The medial surface is concave proximally. The concavity is bordered by two distinct longitudinal, somewhat rugose ridges extending distally half way down the shaft. In the articulated manus, these ridges would face internally. The proximal and distal articular surfaces are slightly twisted. The distal surface is ovoid to subrectangular and does not extend considerably onto the anterior face. The articular facet is flat transversely and convex anteroposteriorly.

Metacarpal IV has a P-shaped proximal articulation surface, with a concave medial edge. As in mtc III, the shaft of mtc IV is twisted, and a distinction of the anterior face is not possible. A distinct ridge connects the posterior apex of the proximal articular surface with the posteromedial corner of the distal articular surface. The distal articular surface is subtriangular as well, with the apex anteriorly, and inclined medial and lateral edges. Two condyles are visible posteriorly. The apex of the distal articular surface curves onto the anterior face.

Metacarpal V is short and widely expanded both transversely and anteroposteriorly at its proximal end. It is somewhat drop-shaped in proximal view, with the tip facing palmarly. The shaft is twisted anti-clockwise, in proximal view. The medial surface is slightly concave for the reception of mtc IV. The free lateral face is gently convex. The medial and lateral surfaces meet at a ridge in their proximal halves. The distal end is partially reconstructed, but the preserved parts indicate that it is transversely expanded. A distinct concavity marking the posterolateral corner of the distal articular surface is of taphonomic origin, having collapsed while being closely attached to the distal articular surface of mtc IV during diagenesis.

Manual non-ungual phalanges. The manual non-ungual phalanges are relatively short and robust (Fig. 66). They are wider than long, as is typical for the eusauropod manus (Bonnan, 2003). The phalanges were found disarticulated, but closely associated with the metacarpals. A definitive assignation to distinct digits can be inferred for phm I–1 and II–1, but the identification of the other three non-ungual phalanges remains uncertain. Based on comparisons with the articulated manus of the Camarasaurus SMA 0002, we identified the elements as phm IV–1, V–1, and II–2. However, they could also be phm III–1, IV–1, and V–1, respectively. The latter arrangement would imply a clearly advanced stage in phalangeal reduction compared to Camarasaurus, but would be supported to some degree by the closer association of the nubbin-like phalanx with mtc IV and II than with mtc II or phm II–1. Nonetheless, given that the other phalanges were dislocated and scattered around the entire metacarpus, the burial location of the vestigial phalanx should not be taken as strong evidence for its articulated position.

Figure 66 Left manual phalanges of Galeamopus pabsti SMA 0011.

The phalanges are shown in anterior, medial, palmar, lateral, proximal, and distal view. Digits are indicated on the left with roman numbers. Abb.: pvl, posteroventral lip; pvlp, posterior ventrolateral process. Scale bar = 5 cm.

The proximal surface of manual phalanx I–1 is concave anteroposteriorly. The phalanx I–1 has a concave posterior surface, with a proximally projecting palmar lip. Its medial surface is shorter than the lateral one, enhancing the angulation of the ungual phalanx even more. The lateral surface is concave proximodistally. The lateral extension of the posterolateral edge forms a thin, short crest (Fig. 66). Nothing similar is present in the manus of Camarasaurus (Osborn, 1904; Tschopp et al., 2015), but too few articulated proximal manual phalanges are known in diplodocids in order to decide if this might be autapomorphic in SMA 0011 or is instead more widespread within the clade. A phalanx figured by Jensen (1985: fig. 1E) appears to show a similar development of the posterolateral edge, but has not been identified below Sauropoda indet (Jensen, 1985). The phm I–1 of SMA 0011 has well-developed medial and lateral distal condyles with a distinct intercondylar groove occurring palmarly. The entire distal surface is subtrapezoidal, being longest palmarly, than medially, laterally and finally dorsally.

Manual phalanx II–1 has a concave proximal surface, which is oval in outline. It is only minimally wider than the shaft. The medial surface is broader, but shorter than the lateral one. The anterior surface is convex transversely. The posterior surface is marked by a bulge at the center of its proximal portion, and a pit distal to it. The distal articular surface is expanded transversely, and the condyles extend onto the medial and lateral surfaces. In anterior view, the distal surface is nearly flat, whereas in distal view, the palmar margin is concave.

Manual phalanx II–2 (as identified herein) is a vestigial, suboval bony nubbin. A distinct ridge separates the proximal and distal surfaces, which are convex and rough.

The probable manual phalanges IV–1 and V–1 are very similar, with IV–1 being slightly larger. They have concave proximal articular surfaces, transversely more so than anteroposteriorly. The surfaces are suboval in outline, and their anterior margins are pronounced laterally. The anterior surfaces are concave proximodistally, but slightly convex transversely. Medial and lateral surfaces are very narrow. The distal surfaces are without condyles. They have a continuous, rounded surface in dorsal view, which curves proximally at its medial and lateral end, almost reaching the proximal articular surface. The medial and lateral surfaces are thus practically nonexistent. The lack of medial and lateral condyles implies that these elements were the terminal phalanges of these digits.

Manual ungual. One ungual is present, situated on the first digit (Fig. 66). It is a long, high, and transversely compressed element. The proximal surface is ovoid, with a narrow anterior tip, and a widened palmar portion, where the articular surface lies. Above the articular surface, the proximal surface projects somewhat proximally, and is rugose. This rugosity extends as a short ridge posteriorly, onto the articular surface. The articular surface is inclined such that when articulated, the ungual would be slightly laterally deflected, compared to the long axis of the preceding phalanx. The medial surface is convex anteroposteriorly. A short groove marks the distal-most portion, which is slightly elevated (about 1 mm) above the more proximal portion of the claw, and shows a different surface texture (Fig. 67). The latter might represent fossilized remnants of the keratinous sheet covering the claw. The lateral surface is almost flat, with a long, proximodistally extending, straight groove covering the distal half of the surface. The palmar surface is strongly convex proximally and flat distally.

Figure 67 Possible preservation of keratinous sheet on left manual ungual I-2 of Galeamopus pabsti SMA 0011 (medial view).

Note the different surface texture at the tip (arrow), compared to more posterior portions. Abb.: dg, distal groove; pas, proximal articular surface. Scale bar = 5 cm.

Hindlimb (Figs. 68–77; Table 7)

Terminology. The pubis and ischium are described as if they were oriented vertically and horizontally, respectively. The distal shaft of the ischium thus has long dorsal and ventral edges in lateral view.

Ilium. The right ilium is preserved, but was found in such a bad state that the medial side had to be covered immediately with plaster (B Pabst, pers. comm., 2014). Therefore, no morphological information can be gleaned from that side. The ilium lacks a large part of the posterodorsal portion of the iliac blade, and the distal-most end of the pubic peduncle (Fig. 68). The preacetabular process has a very pointed apex, which is directed anterolaterally, and is relatively broad transversely. The anterior portion is strongly concave, with the ventral margin facing ventrolaterally. The ventral preacetabular border and the pubic process form an angle of 90°. A triangular depression is located laterally at the base of the pubic process, with a horizontal and medio- and lateroventrally inclined sides. This is similar to the putative diplodocid ilium from Spain (CPT-1074; Royo-Torres & Cobos, 2004; E Tschopp, pers. obs., 2012), and has also been reported in other sauropod taxa (e.g., Cetiosaurus oxoniensis, Lirainosaurus astibiae, and Jobaria tiguidensis; Upchurch & Martin, 2003; Díez Díaz, Pereda Suberbiola & Sanz, 2013; Tschopp, Mateus & Benson, 2015). The pubic peduncle is distinctly concave transversely on its posterior face, but fractures indicate that the concavity is exaggerated and that the transverse width of the pubic peduncle would be slightly larger otherwise. The ischial tubercle faces ventrolaterally. The acetabular margin is thinnest just posterior to the pubic peduncle, and extends transversely towards the articulation surfaces of the ischium and pubis.

Table 7 Hindlimb measurements of Galeamopus pabsti SMA 0011, in mm.

	pdL	pp  pdL	pp  apL	pp  tW	prapL	min  sW	max  tW	isaL	puaL	ptW	dtW	aaL	ptr	papD	dapD	apD	Comments	
Ilium, R	885	340	105	120  (est)	335												prapL measured ventrally	
Pubis, R	845					71	142	320		98	66	200		320	235	97	aaL measured in straight distance between ischial and iliac articulation facets; papD measured horizontally from anterior-most point of ischial articular surface to ambiens process; apD measured laterally at min shaft apD	
Pubis, L	870					55				110	80				273			
Ischium, L									244	482		197						
Femur, L	1,160					208				435			657	145		142	Distal end lacking, pdL is preserved length; ptr: measured to distal end 4th trochanter; apD is min apD	
Tibia, L	845					133				255	238							
Fibula, L	850					85							350	218	145		ptr: measured to center of iliofibular trochanter	
Astragalus, L	142						250							79		105	sw corresponds to maximum transverse width	
mt I L	124					85				120	141			126	76	47	pdL, apD measured laterally	
mt II L	153					64				106	128			98	67	54	pdL measured laterally; papD, apD measured medially	
mt III L	164					30				100	94			76	58	38	pdL, apD measured laterally	
mt IV L	180					36				82	82			80	43	38	apD measured medially	
mt V L	178					52				124	86			72	64	40	apD measured medially	
php I-1 L	82					76				94	84			87	85	74	pdL measured medially	
php II-1 L	96					74				81	88			68	55	48	pdL, dapD, apD measured medially	
php III-1 L	85					52				70	80			62	53	40	pdL, dapD, apD measured medially	
php II-2 L	30									51	41			69			pdL measured medially	
pedal ungual I	170/247									47				150			pdL measured straight proximodistally/oblique from proximal-most to distal-most point	
pedal ungual II	150/205									33				91			pdL measured straight proximodistally/oblique from proximal-most to distal-most point	
pedal ungual III	138/183									51				65			pdL measured straight proximodistally/oblique from proximal-most to distal-most point	
Notes.

Abb. aaL acetabular articulation surface length

apD anteroposterior depth

dapD distal anteroposterior depth

dtW distal transverse width

est estimated

isaL ischial articular surface length

papD proximal anteroposterior depth

pdL proximodistal length

ppapD pubic peduncle anteroposterior depth

ppW pubic peduncle transverse width

prapL preacetabular process length

ptr vertical distance from proximal articular surface to trochanter

ptW proximal transverse width

puaL pubic articular surface length

pupL pubic peduncle length

sW shaft width

Figure 68 Right ilium of Galeamopus pabsti SMA 0011.

The ilium is shown in dorsal (A), posterior (B), lateral (C), anterior (D), and ventral (E) view. Note the triangular depression above the pubic peduncle (1). The yellowish area in B is artificial matrix on which the ilium was mounted for display. Abb.: ac, acetabular surface; isa, ischial articular surface; prap, preacetabular process; pup, pubic peduncle. Scale bar = 20 cm.

Pubes. Both pubes are almost complete, but lack a portion of the ischial articulation. The pubes are relatively slender (Fig. 69). The obturator foramen is completely enclosed and located in the proximal third of the ischial articulation. It is subtriangular in outline and oriented dorsomedially-ventrolaterally. Even though eroded, the anterodorsal corner of the pubis does not seem to bear a very pronounced, hook-like ambiens process, unlike the condition seen in Diplodocus or Supersaurus (Hatcher, 1901; Lovelace et al., 2007). This corner is laterally expanded, and from here, the pubis slightly tapers along the acetabular surface. The medial surface of the proximal half of the bone is proximodistally concave and transversely slightly convex. The latter convexity becomes more pronounced towards midlength, where the ventral margin curves back from the expanded ischial articulation to the narrow midshaft. The dorsal edge of the pubis is gently concave. Its anterior end is expanded both transversely and anteroposteriorly. The narrowest portion of the shaft lies below midshaft, at about two thirds of the entire length of the pubis. The ischiadic articulation is not preserved in its entire length, but broken surfaces indicate that a distinct ridge extended from the ischiadic facet along the ventromedial margin of the shaft to the distal articular surface. The reconstructed length of the ischiadic articulation is about 38% the total length of the pubis (Table 7). The distal end is convex, expanded anteroposteriorly, but not transversely. It is heavily rugose, concave laterally in distal view, and convex medially.

Figure 69 Right pubis of Galeamopus pabsti SMA 0011.

The pubis is shown in medial (A), posterior (B), and lateral (C) view. Abb.: ac, acetabular surface; ip, iliac peduncle; isa, ischial articular surface; of, obturator foramen. Scale bar = 20 cm.

Ischium. The ischium lacks the distal half of the shaft (Fig. 70). It is mounted on plaster, such that only the medial view is accessible. Its proximal portion is wide and concave. The acetabular surface is inclined, such that the medial border forms a thin crest. This crest is relatively straight in medial view, but concave and curved in proximal view. Unlike the state in rebbachisaurids, the acetabular surface does not expand towards the articulation surfaces for the ilium and the pubis (Mannion et al., 2012). The iliac process has no distinct neck and is relatively narrow. The pubic articulation is much longer, and slightly convex in medial view. It curves slightly medially towards its ventral end. The shaft is weakly convex at its base, separating the concave acetabular portion from the again shallowly concave posterior shaft. The dorsal and ventral margins are parallel, only the posterior-most preserved portion of the dorsal edge indicates a slight dorsal expansion towards the end, as is typical for diplodocids (McIntosh, 1990a; McIntosh, 1990b; Upchurch, 1998; Wilson, 2002). No distinct ridges or scars can be seen on the internal surface.

Figure 70 Left ischium of Galeamopus pabsti SMA 0011 in medial view.

The distal end of the ischium is reconstructed. Abb.: ac, acetabular surface; ip, iliac peduncle; pua, pubic articular surface. Scale bar = 20 cm.

Femur. The greater trochanter and the distal end are not preserved in the femur of SMA 0011 (Fig. 71). The medial edge is gently curved below the femoral head, not as distinct as in Dyslocosaurus (McIntosh et al., 1992). The head is separated from the shaft ventrally, but does not project far medially. It is slightly wider transversely than anteroposteriorly, and has a strongly rugose surface. The lateral margin of the shaft is slightly convex proximally, forming a weak lateral bulge, but no medial deflection of the proximal end occurs. The shaft is crushed at its center, but it is obvious that the medial side was anteroposteriorly wider than the lateral one. There is no indication of a large foramen opening at the center of the anterior surface, as occurs in some specimens of Diplodocus and Dicraeosaurus (Wilson, 2002; Whitlock, 2011a; Tschopp, Mateus & Benson, 2015), but some parts in that area are reconstructed. The fourth trochanter is entirely located on the posterior surface of the shaft, but close to the medial border proximally. The distal end of the fourth trochanter curves distinctly laterally towards the midline of the shaft. The fourth trochanter is medially accompanied by a shallow depression proximally and two rugose tubercles centrally and distally (Fig. 71D). The shaft is 1.5 times as wide as it is anteroposteriorly thick (Table 7). The more distally located tubercle of the two is the more developed. The preserved, distal-most part of the shaft slightly expands transversely.

Figure 71 Left femur of Galeamopus pabsti SMA 0011.

The femur is shown in proximal (A), anterior (B), lateral (C), posterior (D), and medial (E) view. Note the rugose tubercles on the posterior surface of the shaft (1). The distal end is restored, and parts of the distal anterior surface are covered with epoxy for the mounting (indicated by white lines). Abb.: fh, femoral head; ft, fourth trochanter; icg, intercondylar groove; lb, lateral bulge. Scale bar = 20 cm.

Tibia. The tibia is complete, but compressed anteroposteriorly (Fig. 72). It is slightly expanded at both ends. The proximal end is longer transversely than anteroposteriorly, but this is partly due to taphonomic compression. The outline of the proximal articular surface is subrectangular as in apatosaurines, and unlike the subtriangular state as in diplodocines (Lovelace et al., 2007). However, it is unclear how much this shape is influenced by the compression. The cnemial crest is somewhat displaced distally, and is thicker distally than proximally. It projects laterally. Posterior to the crest, a fossa occurs for the reception of the fibula, which is posteriorly bound by a wide longitudinal ridge of about the same length as the cnemial crest. The lateral side of the shaft is much narrower than the medial one. A small convexity marks the distal end of the lateral edge. The distal articular surface has the typical step-like arrangement as in all sauropods, for the articulation with the ascending process of the astragalus.

Figure 72 Left tibia of Galeamopus pabsti SMA 0011.

The tibia is shown in proximal (A), medial (B), anterior (C), lateral (D), and distal (E) view. Abb.: cc, cnemial crest. Scale bar = 10 cm.

Fibula. The fibula is a slender bone, with a strongly anteroposteriorly expanded proximal end, and a less expanded distal end (Fig. 73). The proximal end is transversely compressed. It has a pointed anterior end, which projects somewhat medially, similar to Diplodocus carnegii CM 94 (Hatcher, 1901). A distinct, but proximodistally short ridge extends from the posterior end down the shaft (Fig. 73B), for about 9 cm. The medial surface is marked by a subtriangular area with a striated rugosity, which covers about the proximal-most 20–25% of the shaft. The attachment site for the iliofibularis muscle, which is often called fibular trochanter, is situated slightly above midheight, as in Diplodocus (Whitlock, 2011a), and has an oval outline. Towards its distal end, the shaft expands more strongly transversely than anteroposteriorly, but the distal articular surface still remains anteroposteriorly longer than transversely wide. In particular the distomedial edge expands to articulate with the fibular facet on the astragalus. The distal articular surface has an oval outline.

Figure 73 Left fibula of Galeamopus pabsti SMA 0011.

The fibula is shown in proximal (A), lateral (B), posterior (C), medial (D), and distal (E) view. Note the anteromedially projecting process on the proximal articular surface (1), the short longitudinal ridge on the posterior surface (2), and the striated rugosity on the medial surface (3). Abb.: fit, fibular trochanter. Scale bar = 10 cm.

Astragalus. The astragalus is wedge-shaped in both anterior and proximal views (Fig. 74). The anteromedial corner is reduced. Posteriorly, the astragalus is marked by a high ridge connecting to the ascending process. The latter extends backwards to the posterior end. The high, 42 mm wide ridge separates the two fossae for the articulation with the tibia medially and the fibula laterally. The ridge itself is slightly concave transversely, and bound by two distinct, dorsoventrally extending margins. The two margins end in two pronounced, bulge-like posteroventral expansions. The two expansions are separated by a strongly concave posteroventral margin in ventral view, similar to the condition considered autapomorphic in Janenschia robusta (Bonaparte, Heinrich & Wild, 2000). The tibial fossa is larger than the fibular fossa and subdivided by a shallow, oblique, anteroposteriorly oriented ridge into a medial and a lateral portion. The medial portion is pierced by three large foramina. The fibular fossa is relatively uniform, with the anterior edge forming a distinct lip-like lateral extension. The fibular fossa is thus visible in posterior view, a diplodocoid synapomorphy convergently acquired by Jobaria (Whitlock, 2011a). The distal roller is flattened due to compression, and appears to be subdivided horizontally into three distinct parts: an anteriorly facing portion, an anteroventral face, and a ventral part.

Figure 74 Left astragalus of Galeamopus pabsti SMA 0011.

The astragalus is shown in proximal (A), medial (B), anterior (C), lateral (D), posterior (E), and distal (F) view. Note the autapomorphic, concave posteroventral edge, with a distinct bulge medial to it (1). Abb.: af, astragalar foramen; asp, ascending process; dro, distal roller; fif, fibular facet; tif, tibial facet. Scale bar = 5 cm.

Pes. The pes was found associated with the astragalus, tibia and fibula, but slightly out of articulation. The absence of a calcaneum might therefore be due to taphonomy. Metatarsals I and II were found somewhat separated from mts III–V, with the phalanges php I–1 and III–1 in between. The first ungual was lying above the astragalus, whereas digit II was found in articulation. No other phalanges were found associated, but a small left pedal ungual was recovered mingled with the skull elements, and was therefore used in the mount. It is here described, but attribution to SMA 0011 must be considered preliminary.

Metatarsals. All left metatarsals were recovered complete (Fig. 75). Metatarsals III and IV are the longest elements, and mts I and II the stoutest (Table 7).

Figure 75 Left metatarsals of Galeamopus pabsti SMA 0011.

The metatarsals are shown in anterior, medial, plantar, lateral (distal surface towards the bottom), proximal, and distal view (plantar surface towards the bottom). Digits are indicated on the left with roman numbers. Note the lip-like extension of the proximal articular surface that overhangs the plantar surface (1). Abb.: dlr, dorsolateral ridge; plp, posterolateral process. Scale bar = 10 cm.

Metatarsal I is very robust, and the bone surface collapsed diagenetically in two areas on the dorsal and the lateral surface. The first metatarsal has a D- to drop-shaped proximal surface, which is wider anteriorly than plantarly and has a concave lateral margin. The anterior surface is considerably shorter medially than laterally, resulting in angled proximal and distal surfaces, compared to the long axis of the shaft. The anterior surface bears few nutrient foramina, as is the case in Cetiosauriscus and Suuwassea, but not in camarasaurids (Harris, 2007; Tschopp, Mateus & Benson, 2015; Tschopp et al., 2015). The posterior surface is convex proximally and bears a small foramen centrally on its distal half. The medial surface is slightly convex anteroposteriorly, the lateral one concave for the reception of mts II. Distally, the lateral condyle projects much further than the medial, and develops a distinct posterolateral process, as is typical for diplodocids (McIntosh, 1990a; McIntosh, 1990b). The distal part of the anterolateral edge is marked by a rugose tubercle accompanied by a particular bone surface structure resembling a net of veins (Fig. 76). The distal articular surface bears a distinct intercondylar groove visible in anterior and plantar view.

Figure 76 Surface structure on metatarsal I.

The left metatarsal I of Galeamopus pabsti SMA 0011 in anterior view (A), with the particular bone surface texture shown in (B). Scale bar = 2 cm.

Metatarsal II has a more squared proximal surface compared to mts I, but with concave medial and lateral margins. The anterior surface is less trapezoidal than in mts I. However, the proximal and distal articular surfaces of mts II are still angled to the long-axis of the shaft. As in mts I, mts II has a strong posterolateral process. The distal portion of the anterolateral edge bears a distinct rugosity, which does not extend onto the anterior surface, unlike in Dyslocosaurus AC 663 or Cetiosauriscus NHMUK R3078 (McIntosh et al., 1992; Tschopp, Mateus & Benson, 2015). Metatarsal II of SMA 0011 has a very distinct anteromedial edge, but a less developed anterolateral one. No intercondylar groove can be seen between the distal condyles in anterior view, but a shallow groove occurs posteriorly.

Metatarsal III is elongate, with a narrow shaft and greatly expanded proximal and distal ends. The proximal and distal articular surfaces stand perpendicular to the shaft axis. The proximal articular surface is subtriangular, with anterior, lateral, and medioplantar margins. It is relatively flat, and does not show distally curving edges as in mts I and II. A strong, narrow projection occurs on the posteromedial corner. A weak, narrow rugosity marks the distal end of the anterolateral edge of the shaft. The proximal portions of the medial and lateral faces are anteroposteriorly concave. The distal articular surface is subtriangular, with the lateral side being much shorter than the medial. It is anteroposteriorly convex and transversely nearly flat.

Metatarsal IV is elongate like mts III, but the proximal expansion reaches further down the shaft. The proximal end is slightly twisted in respect to the long axis. It is subtriangular in outline, with a rather straight lateroplantar margin, unlike the shape of mts IV of the camarasaur SMA 0002 (Tschopp et al., 2015). The surface is flat, as in mts III. The shaft is smooth, and maintains the subtriangular shape of the proximal articular surface. It is concave transversely on its lateroplantar surface, and does not bear any distinct rugosities. The distal end has only incipient condyles, which are hardly recognizable in either anterior or distal view. In distal view, the articular surface is trapezoidal, with a shorter anterior than plantar margin.

Metatarsal V has the typical paddle-shaped outline known from almost all sauropods (Bonnan, 2005). The proximal articulation surface is subtriangular, with the apex pointing anteromedially. From there, a ridge extends distally, separating the proximal portion of the anterior surface from the medial one. The ridge disappears in the distal half. The shaft is smooth, unlike in mts V of the camarasaurid SMA 0002 (Tschopp et al., 2015). The posterior surface is flat transversely, but a lip-like posterior extension of the proximal surface overhangs the face (Fig. 75). The distal surface is a single, convex facet.

Pedal non-ungual phalanges. The left pes of SMA 0011 preserves three proximal non-ungual phalanges and the second non-ungual phalanx of the second digit (Fig. 77). They are relatively short bones with subsequently less well-developed distal condyles, from php I-1 to php III-1.

Figure 77 Left pedal phalanges of Galeamopus pabsti SMA 0011.

The phalanges are shown in anterior, medial, plantar, lateral (distal surface towards the top), proximal, and distal view (plantar surface towards the bottom). Digits are indicated on the left with roman numbers. Scale bar = 5 cm.

Pedal phalanx I–1 is slightly wedge-shaped, with a considerably shorter lateral than medial surface. Therefore, the distal condyles face laterodistally, resulting in the typical lateral deflection of the pedal unguals of eusauropods (Bonnan, 2005). The proximal articular surface is subtrapezoid, with two distinct, concave facets for the two distal condyles of mts I. In the medial facet, a deep pit is located close to the midline, and somewhat more anteriorly than plantarly. A similar pit was interpreted as the result of osteochondrosis in the camarasaurid SMA 0002 (Tschopp et al., 2016). The anterior surface is transversely narrower than the posterior surface. It is clearly separated from the medial surface, but grades continuously into the lateral one. The posterior surface is transversely concave, with a smooth transition into the distal articular surface. Laterally, the proximal and distal articular surfaces nearly meet in the plantar half. The distal condyles are in an angle to each other, with the medial one being oriented nearly vertically, whereas the lateral one is oblique, resulting in an articular facet that is anteriorly narrower than plantarly.

Pedal phalanges II-1 and III-1 are similar to each other in general shape. The former is slightly broader than php III-1, which has subequal widths and lengths (Table 7). The medial condyle of both phalanges is transversely compressed, but projects considerably further distally than the lateral one. The proximal articular surface of php II-1 bears a deep pit as in php I-1. Laterally, the proximal facets of both php II-1 and III-1 taper, such that the outline becomes subtriangular in proximal view.

The pedal phalanx II-2 is a proximodistally shortened element, which basically only consists of proximal and distal articular surfaces and a short medial face. The proximal articular surface has two facets for the condyles of php II-1. It is at an angle to the long-axis as indicated by the orientation of the short medial surface. The distal articular surface has a relatively wide medial condyle, and a thin and narrow lateral one. The orientation of the two condyles is subparallel.

Pedal unguals. Three left unguals are preserved and mounted in the left pes of SMA 0011 (Fig. 77). The third ungual was found at some distance to the associated pes, together with skull material, but would fit in size for digit III. As mounted, this amounts to a pedal phalangeal formula of 2-3-2-0-0. This, however, is most probably underestimated, as comparisons with other diplodocid feet indicate (Hatcher, 1901; Gilmore, 1936; Janensch, 1961; Bonnan, 2005). The pedal unguals are sickle-shaped and decrease in length from the first to the third. Ungual III is the most stout element, because the proximal width remains more or less the same from ungual I to III, whereas the length decreases. The pedal unguals I and II are strongly transversely compressed, but this is exaggerated due to taphonomy. The anterior edge is strongly curved and narrow. It is S-shaped in ungual I, because of deformation in the proximal-most part. The medial surfaces are convex, and the lateral sides are anteroposteriorly concave on their anterior portion and convex plantarly. The pedal unguals are wider transversely in their plantar half, especially at the proximal end, where the wider area bears the proximal articular surface. A groove marks the lateral surface, and follows more or less the curvature of the claw. The plantar surface of pedal ungual I is marked by a deep oblique groove, extending from the proximomedial corner to about midlength of the lateroplantar edge. Such a groove has not been described previously, and does not occur in the other two unguals of the same pes. The groove might be caused by taphonomy, because according to the quarry map, a sternal rib was found above it. During diagenesis, this rib could have been pressed onto the claw resulting in such a relatively wide, but elongate groove. The plantar surfaces of pedal unguals I and II bear a weak tubercle, resembling that of Tastavinsaurus sanzi Ars1-3 (Canudo, Royo-Torres & Cuenca-Bescós, 2008: figs. 10A–19B).

Discussion

Phylogenetic analysis

The phylogenetic position of SMA 0011 was determined using a species-level version of an updated matrix of Tschopp, Mateus & Benson (2015), who recovered it consistently within the genus Galeamopus, closely related to its type species G. hayi, but potentially specifically different. However, in their input file for the software TNT, the multistate character statements to be ordered were erroneously defined with their real character numbering, whereas TNT requires a character numbering initiating with “0”. This resulted in only two supposed ordered multistate character statements that were actually treated as ordered (their C49, C380), and one multistate character statement that should have been treated as unordered (according to Tschopp, Mateus & Benson, 2015), which was in fact treated as ordered (their C154).

The input file was therefore updated in several aspects: multistate characters were reanalyzed in detail, resulting in the split of several characters into two binary characters, following the reasoning of Sereno (2007) and Brazeau (2011). Ordering was only implemented for the remaining multistate characters if they described numerical or meristic features (as e.g., the number of cervical vertebrae), and if the states described an obvious, morphological transformational series. Character scores were updated based on personal observations of ET between 2014 and 2016 of specimens of the OTUs Jobaria tiguidensis, Brachiosaurus altithorax, Giraffatitan brancai, Haplocanthosaurus priscus, and the specimen described here, SMA 0011. Species-level OTUs were created following the identifications of Tschopp, Mateus & Benson (2015: tab. 5), excluding the specimens that could not be unambiguously referred to a diplodocid species, and using frequency scoring (Wiens, 1995; Wiens, 1998; Wiens, 2000) for polymorphic characters in species represented by more than two specimens, and polymorphic scores if the only two specimens of a species showed conflicting states. The outgroup was reduced to include only species with well-established phylogenetic positions. The final matrix included 489 character statements and 35 operational taxonomic units.

The analysis was performed under implied weights, with the concavity constant k set to 5. We used the new technology tree searches in TNT v. 1.1 (Goloboff, Farris & Nixon, 2008), with all algorithms enabled, and stabilizing the consensus tree five times with a factor of 75. One best tree was found with a score of 124.01986. A second iteration using TBR did not yield additional shortest trees.

The topology of the tree (Fig. 78) is nearly the same as in the combined cladogram proposed by Tschopp, Mateus & Benson (2015: fig. 120), when excluding the yet unnamed, potentially new species. The only difference is the inclusion of Amphicoelias altus in a clade with the species of Brontosaurus, and the more basal position of Kaatedocus siberi within Diplodocinae (Fig. 78). The placement of Amphicoelias among Brontosaurus was already recovered by the main analyses of Tschopp, Mateus & Benson (2015), but additional analyses showed that this position might be dubious and actually due to the limited amount of data in the holotype specimen AMNH 5764, which is the only one that can currently be referred to the species with certainty (Tschopp, Mateus & Benson, 2015). An updated, specimen-level analysis is in preparation, evaluating these issues in more detail, but is outside the scope of the current paper.

Figure 78 Single tree with best fit of the phylogenetic analysis under implied weighting (k = 5).

The analysis is based on an updated matrix of Tschopp, Mateus & Benson (2015), transformed to a species-level analysis. Galeamopus pabsti sp. n. is recovered within Diplodocinae, as sister taxon to Galeamopus hayi. Note that the position of Amphicoelias altus among the species of Brontosaurus is dubious (see text).

SMA 0011 is recovered as sister taxon to Galeamopus hayi (Fig. 78). Together, they form the sister taxon to a clade uniting Barosaurus and Diplodocus. TNT provided ten autapomorphies for G. hayi, and 18 for the new species G. pabsti (see Data S1–Data S2). Given the numerous morphological differences between SMA 0011 and G. hayi, specific separation of the two is warranted. The numerical values for generic distinction proposed by Tschopp, Mateus & Benson (2015) cannot be applied equally here because we changed both taxon and character sampling. Also, it remains unclear if a similar system of assessing variability can work with matrices at species level, where intraspecific variation has already been accounted for by using frequency and polymorphic scoring. We therefore prefer to distinguish the two at species level only, and revised the diagnoses accordingly (see above).

Comparison with Galeamopus hayi

Specific distinction of SMA 0011 from the type species Galeamopus hayi was already proposed by Tschopp, Mateus & Benson (2015). These authors recognized six features unique to the holotype specimen of G. hayi, HMNS 175: (1) a low posterolateral process of the parietal, compared to foramen magnum height; (2) basipterygoid processes that diverge more than 60°; (3) the ulna that is longer than 76% the length of the humerus; (4) a radius with relatively weak posterodistal ridges for articulation with the ulna; (5) the strongly beveled distal articular surface of the radius; and (6) the presence of a projection of the proximal articular surface of the tibia, behind the origin of the cnemial crest. Furthermore, Tschopp, Mateus & Benson (2015) found one unambiguous autapomorphy for SMA 0011, the presence of a neural arch foramen connecting the POCDF and the SPOF, and seven ambiguous ones: (1) anterior cervical vertebrae that are more than 1.2 times higher than wide; (2) PCDL and PODL of posterior cervical vertebrae that do not meet at the base of the transverse process; (3) strong opisthocoely of dorsal centra disappears between DV 2 and 3; (4) posterior dorsal neural spines that are longer than wide at their ventral base; (5) a very robust humerus, with an RI (sensu Wilson & Upchurch, 2003) of 0.37 (Table 6); (6) absence of a shallow tubercle in the center of the proximal half of the anterior surface of the humerus; and (7) a radius that has a proximal articular surface that is 0.3 times its proximodistal length.

A more detailed reevaluation of these characteristics in SMA 0011 shows that some were interpreted wrongly by Tschopp, Mateus & Benson (2015) and others are more widespread among diplodocids. In fact, the angle of the basipterygoid processes cannot be accurately assessed in SMA 0011, and the processes are broken and incomplete in the other two skulls referred to Galeamopus by Tschopp, Mateus & Benson (2015; AMNH 969, USNM 2673). A beveling of the distal surface of the radius also occurs in SMA 0011, but affects only the lateral half of the surface, whereas HMNS 175 has a nearly entirely beveled surface. The tibiae of the two specimens have a similar proximal articular surface, such that the presence of the projection behind the cnemial crest can be interpreted as an autapomorphy of the genus Galeamopus. Strong opisthocoely in dorsal vertebrae actually disappears between DV 1 and 2 in SMA 0011, which is even more anterior than what would already be unique in diplodocines (Tschopp, Mateus & Benson, 2015). Such an anteriorly located change from strongly opisthocoelous to relatively flat anterior condyles in dorsal centra would be unique among diplodocoids, but the state in HMNS 175 cannot be currently assessed due to the apparent lack of associated ribs. Given that the overall morphology of the centra from the cervico-dorsal transition of SMA 0011 and HMNS 175 is very similar, the anterior position of the first flat anterior articular surface in the dorsal column is more cautiously interpreted as an autapomorphic feature of the genus. The width of the base of the neural spines cannot be assessed on HMNS 175 with certainty due to extensive reconstruction. The humerus of SMA 0011 actually bears a marked, rugose area anteriorly in the center of the proximal half (Fig. 60), contrary to what was stated in Tschopp, Mateus & Benson (2015).

In addition to the autapomorphic features of Galeamopus hayi and G. pabsti mentioned in the diagnoses, SMA 0011 reveals several additional differences from HMNS 175. The distinction between dorsal and posterior surfaces of the parietal are less clear in HMNS 175 compared to SMA 0011. The distal ends of the paroccipital processes are straight in lateral view in SMA 0011 and curved in HMNS 175 (Tschopp, Mateus & Benson, 2015). The basisphenoid of SMA 0011 is marked by a pit between the occipital condyle and the basal tubera.

In the cervical vertebrae, transverse processes of SMA 0011 do not have any posterior projections. Bifurcation of the neural spines already occurs in CV 5 or more anteriorly in HMNS 175 (Tschopp, Mateus & Benson, 2015), whereas in SMA 0011 the first bifid element is CV 8. The spine summits of the bifid cervical vertebrae of HMNS 175 are more rounded than in SMA 0011, and do not bear a horizontal, rugose ridge below the neural spine summit on the lateral surfaces of posterior cervical neural spines. Mid-cervical centra of HMNS 175 are less elongate than the ones from SMA 0011, which also have pre-epipophyses that extend distinctly beyond the prezygapophyseal facets. Mid- and posterior cervical vertebrae of SMA 0011 have vertical SPRLs without any fossa ventrolateral to them on the prezygapophyseal ramus. In contrast to the state in HMNS 175, SMA 0011 has epipophyses of posterior cervical vertebrae that are dorsoventrally compressed, and pleurocoels of anterior and mid-dorsal vertebrae that invade the neural arches. The dorsal neural arches appear to be higher and with a less distinct lamination in SMA 0011 compared to HMNS 175, although the development of the lamination in SMA 0011 might be affected by taphonomy.

The acromial ridge of the scapula is better developed in HMNS 175 compared to SMA 0011, as is a ridge following the long axis of the distal blade. The latter results in a somewhat triangular cross-section of the distal blade in HMNS 175, whereas it is rather D-shaped in SMA 0011. The laterally projecting sheet of bone on the lateropalmar edge of phm I-1 in SMA 0011 does not have an equivalent structure in HMNS 175, but it remains unclear if this feature might be of pathological origin in SMA 0011.

Given all these differences, and the fact that Tschopp, Mateus & Benson (2015) already found strong evidence for specific separation of the two specimens, the erection of Galeamopus pabsti as second species of Galeamopus can be confidently justified (based on both morphological and phylogenetic species concepts). Based on the additional information from the articulated type specimens on differing skull morphology in the two species, also the two skulls referred to the genus by Tschopp, Mateus & Benson (2015) can be identified more precisely: AMNH 969 has a relatively narrow sagittal nuchal crest, curved distal ends of the paroccipital processes, and a slightly laterally expanded otosphenoidal crest, and can thus be referred to G. hayi. USNM 2673 appears to have a similarly shaped anterior notch between the frontals as SMA 0011, and a vertical median groove on the sagittal nuchal crest, favoring a referral to G. pabsti.

Ontogenetic implications

The specimen SMA 0011 shows a variety of features that have been previously reported to indicate a juvenile age for an animal. However, histology indicates that it reached sexual maturity (see below). Cranial ontogeny in diplodocids was extensively discussed by Whitlock, Wilson & Lamanna (2010), who proposed the following juvenile features in Diplodocus: a relatively rounded snout, with tooth rows that reach further back, and a large orbit. Whereas the latter is typical for most amniotes (Varricchio, 1997; Whitlock, Wilson & Lamanna, 2010), the first two characteristics also occur in subadults and adults of other diplodocines (Tschopp & Mateus, 2013b). The skull of SMA 0011 has an orbit of about the same relative size as the large diplodocine skull CM 11161, and thus relatively smaller than the juvenile diplodocine CM 11255 (Whitlock, Wilson & Lamanna, 2010). However, the snout of SMA 0011 is more rounded, with a premaxillary–maxillary index reaching only 72%, compared to more than 80% in CM 11161 (Whitlock, 2011b). Thus, whereas orbit size seems indeed to be controlled by ontogeny, snout curvature appears to be more variable and potentially phylogenetically informative.

Potential osteological characteristics of young age in the postcranial skeleton of SMA 0011 include unfused vertebral centra and neural arches (although only in mid-cervical and mid-dorsal vertebrae), unfused cervical ribs, and a separate scapula and coracoid (Gilmore, 1925; Janensch, 1961; McIntosh, 1990b; Wedel & Taylor, 2013). Other characteristics have often been proposed to be an indicator for a young age, but are absent in SMA 0011: unlike what is seen in juveniles, the coracoid and pubic foramina are completely enclosed in SMA 0011, and the articular surfaces of the long bones are strongly rugose in SMA 0011 (Hatcher, 1903; McIntosh, 1990b; Bonnan, 2003; Schwarz et al., 2007). Furthermore, the absence of fusion between sacral vertebrae was shown to reflect ontogeny (Riggs, 1903; Mook, 1917; Wedel & Taylor, 2013), and the sternal plates are thought to adopt their definitive shape in adult animals only (Wilhite, 2003; Wilhite, 2005), but neither the sacrum nor any sternal plate is preserved in SMA 0011. Carpenter & McIntosh (1994) also proposed that the longitudinal ridges on the distal shafts of radius and ulna develop during ontogeny, but this could also be a phylogenetically informative character, given that adult Dyslocosaurus and Diplodocus specimens appear to have them much less developed than Apatosaurus (E Tschopp, pers. obs., 2011). Several authors showed that vertebral lamination and pneumaticity increases during ontogeny (Wilson, 1999; Wedel, Cifelli & Sanders, 2000; Wedel, 2003; Bonnan, 2007; Schwarz et al., 2007; Carballido & Sander, 2014), but only the smallest neosauropod specimens show largely reduced pleurocoels and laminae (equivalent to the MOS 1; Schwarz et al., 2007; Carballido & Sander, 2014; CM 566, SMA 0009, E Tschopp, pers. obs., 2011). Wedel, Cifelli & Sanders (2000) reported an increase in cervical centrum elongation of 35–65% in Apatosaurus. However, their calculation was based on juvenile vertebrae from Oklahoma, identified as Apatosaurus by Carpenter & McIntosh (1994), but some of them might actually belong to Camarasaurus (Upchurch, Tomida & Barrett, 2004). Increase in centrum elongation was also shown to happen during ontogeny of Europasaurus (Carballido & Sander, 2014). Recently, it has furthermore been suggested that the bifurcation of the neural spine is ontogenetically controlled (Woodruff & Fowler, 2012), but this has been shown to be questionable (Wedel & Taylor, 2013).

Given the presence of both open neurocentral synchondroses and closed synostoses in some cervical and dorsal vertebrae of SMA 0011, the present specimen qualifies for the MOS 3 and 4 of Carballido & Sander (2014). Vertebrae of Europasaurus holgeri of these stages already show all phylogenetically significant characters of the species (Carballido & Sander, 2014). The same was hypothesized for Suuwassea emilieae ANS 21122 (Hedrick, Tumarkin-Deratzian & Dodson, 2014) and Bonitasaura salgadoi MPCA-460 (Gallina, 2011; Gallina, 2012), which are the only sauropod specimens for which information from long bone histology and neurocentral closure could be correlated until now. It therefore seems clear that unfused vertebrae with well-developed lamination as in SMA 0011 can be identified to species level, and that the several morphological differences between SMA 0011 and the type specimen of Galeamopus hayi qualify as species autapomorphies.

Histology. The histology of the scapula, humerus, and femur of SMA 0011 has been described by Klein & Sander (2008). This allows for an accurate comparison of morphological and histological ontogenetic markers. Both the humerus as well as the femur of SMA 0011 were classified within histological ontogenetic stage 9, whereas the scapula showed a varying degree of remodeling from medial to lateral (Klein & Sander, 2008). This is the same age proposed for Suuwassea (Hedrick, Tumarkin-Deratzian & Dodson, 2014) and Bonitasaura (Gallina, 2012), and is probably the stage where sexual maturity is reached (Klein & Sander, 2008), because it correlates with a decrease of growth rates (see also Scheyer, Klein & Sander, 2010).

Timing of neurocentral closure. The pattern of neurocentral closure is variable among archosaurs (Brochu, 1996; Irmis, 2007; Birkemeier, 2011; Ikejiri, 2012). Even within Sauropoda, varying patterns have been reported (Harris, 2006b; Irmis, 2007; Gallina, 2011; Carballido & Sander, 2014). The incomplete nature and rareity of immature specimens result in additional difficulties, and very little information is available from articulated or associated vertebral columns (Gilmore, 1925; Harris, 2006b; Schwarz et al., 2007; Gallina, 2011; Carballido et al., 2012). The current specimen is thus of special importance for the study of neurocentral closure in sauropods.

SMA 0011 has closed but visible neurocentral synostoses in anterior and posterior cervical vertebrae, and in anterior-most and mid- to posterior dorsal vertebrae. Mid-cervical and one mid-dorsal vertebrae of SMA 0011 have open neurocentral synchondroses, but not all mid-dorsal elements are preserved. No cervical rib is fused to its corresponding centrum. Given that long bone histology shows that SMA 0011 had already reached sexual maturity (Klein & Sander, 2008), it seems that open synchondroses still occurred in sexually mature sauropods, a fact already reported from flagellicaudatan and titanosaur specimens (Gallina, 2011; Gallina, 2012; Hedrick, Tumarkin-Deratzian & Dodson, 2014). In the flagellicaudatan Suuwassea emilieae ANS 21122, vertebral fusion was apparently already completed in the preserved presacral vertebrae, but not in caudal vertebrae (Harris, 2006b). However, only fragmentary mid- and posterior cervical, and no mid- and posterior dorsal vertebrae are preserved in ANS 21122, which are the only elements still showing unfused centra and neural arches in SMA 0011. As in SMA 0011, ANS 21122 also has unfused cervical ribs, a separate scapula and coracoid, but a closed coracoid foramen and relatively rugose articular surfaces of the longbones (Harris, 2006b; Harris, 2007; Hedrick,Tumarkin-Deratzian & Dodson, 2014). The two specimens therefore seem to be of about the same morphological ontogenetic stage. The titanosaur Bonitasaura MPCA-460 appears to show a slightly different pattern of neurocentral closure, with a completely fused axis, but open anterior cervical and anterior dorsal vertebrae, and closed posterior cervical and posterior dorsal elements (Gallina, 2011). However, MPCA-460 was shown to fit into HOS 9 (Gallina, 2012), like SMA 0011 (Klein & Sander, 2008). These three specimens therefore indicate that neurocentral closure was delayed and only partially completed by sexual maturity in sauropods, as is the case in some crocodiles and lizards (Brochu, 1996; Maisano, 2002; Ikejiri, 2012). They also show that the pattern of closure is not as simple as previously thought. Based on comparisons with crocodiles, and on specimens with open synchondroses and closed neurocentral synostoses, a posterior-to-anterior sequence was postulated (Brochu, 1996; Irmis, 2007; Birkemeier, 2011; Ikejiri, 2012; Tschopp & Mateus, 2013b). However, SMA 0011 shows that—at least in diplodocids—in both the cervical and the dorsal column, the middle elements fuse last, and that within one single vertebra, the fusion starts posteriorly and progresses anteriorly (Fig. 29). Moreover, a detached, and partly broken left prezygapophysis of a posterior cervical vertebra shows the typical surface of a synchondrosis at the ventral end of the CPRL, whereas the right portion of the neural arch is clearly fused with the centrum. This peculiar feature indicates that there might even be some left–right asymmetry in the fusion pattern. Adding the information from Suuwassea ANS 21122, anterior cervical vertebrae appear to fuse first (also in SMA 0011, these are the ones where the synchondroses are the least visible), followed by anterior and posterior dorsal and posterior cervical vertebrae, whereas mid-cervical, mid-dorsal, and anterior to mid-caudal vertebrae fuse last. This varies from the condition in Bonitasaura, where a posterior-to-anterior pattern was proposed in both the postaxial cervical and dorsal columns (Gallina, 2011). A general posterior-to-anterior fusion pattern also appears to be present in at least one specimen of Camarasaurus (Trujillo et al., 2011) and the small juvenile possible Brachiosaurus SMA 0009, which already have closed, but still visible, synchondroses in anterior caudal vertebrae (Schwarz et al., 2007; Trujillo et al., 2011; Carballido et al., 2012). Different fusion patterns might thus prove to be a phylogenetically informative character, with macronarians showing a faster neurocentral closure than diplodocoids, and following a more strict posterior-to-anterior pattern, at least in the single vertebral regions. However, too few specimens are known to date, in which neurocentral closure can be directly compared with histology, in order to evaluate this character statistically. Nonetheless, these finds have further implications for the individual age of the holotype specimen of Kaatedocus siberi, SMA 0004 (Tschopp & Mateus, 2013b), which does not show any traces of neurocentral synostoses in any cervical vertebra, and also has completely fused cervical ribs (Tschopp & Mateus, 2013b). Being a diplodocine, this implies that Tschopp & Mateus (2013b) were right in identifying SMA 0004 as at least a subadult specimen, which retained a relatively small size. Moreover, as Carballido & Sander (2014) showed for Europasaurus, sauropod vertebrae already show the majority of the phylogenetically informative characters of their respective species before the completion of neurocentral closure. These results corroborate the findings of Wedel & Taylor (2013) that the posterior onset of neural bifurcation in cervical and dorsal vertebrae is not clearly correlated with ontogeny (contra Woodruff & Fowler, 2012).

Cervico-dorsal transition in Diplodocidae

Vertebral segmentation is a complex phenomenon. According to Romer (1956, p. 228), “the study of segmentation is comparable to the study of the Apocalypse. That way lies madness”. Among sauropods, SMA 0011 is one of few specimens that preserves articulated posterior cervical and anterior dorsal vertebrae with closely associated ribs. Five vertebrae were found in articulation, with the first clearly being a posterior cervical, and the last two being anterior dorsal vertebrae. Several morphological changes occur in the two intermediate vertebrae, which are outlined above. The most important ones concern the shortening of the centrum, the loss of a distinct anterior condyle, and the changing position of the parapophysis.

Generally, the position of the parapophysis is considered to be ventral or anterior to the pleurocoel in the first two dorsal centra of diplodocid sauropods (Hatcher, 1901; Gilmore, 1936); whereas in the dicraeosaurid Brachytrachelopan, the macronarian Camarasaurus, and in the stegosaur Miragaia, the parapophysis of the first dorsal vertebra is situated on the anterodorsal corner of the centrum (McIntosh et al., 1996; Rauhut et al., 2005; Mateus, Maidment & Christiansen, 2009). A distinct shortening of the vertebral centrum, as occurs between the third and the fourth vertebra of the articulated transitional block in SMA 0011 was interpreted to happen between DV 2 and 3 in Diplodocus carnegii (Hatcher, 1901) and Barosaurus lentus (McIntosh, 2005). The first dorsal vertebra without a distinct anterior condyle was proposed to be DV 5 in D. carnegii (Hatcher, 1901), DV 3 in Apatosaurus louisae (Gilmore, 1936), and DV 4 in B. lentus (McIntosh, 2005).

Different researchers have used varying morphological indicators to distinguish cervical from dorsal vertebrae. Hatcher (1901) and Gilmore (1936) used the presence of fused or free ribs to define cervical or dorsal vertebrae, respectively. Furthermore, Hatcher (1901) noted that the first dorsal vertebrae had a convex ventral surface. Janensch (1929) stated that the transition from cervical to dorsal vertebrae is often gradual, and that only the vertebrae bearing ribs that are connected to the sternum can be regarded as dorsal vertebrae, following the definition of dorsal vertebrae given by Stannius (1846). In fact, the definition of Stannius (1846) appears to be the most universally applicable, and has therefore been applied in a wide variety of vertebrates (Hoffstetter & Gasc, 1969). In any case, it seems that it is not possible to consistently identify the vertebrae alone as either cervical or dorsal elements, a fact that is also exemplified by the difficulties in defining the exact cervico-dorsal transition in the macronarian Euhelopus, where the proposed first dorsal vertebra lacks ribs (Wilson & Upchurch, 2009). Ribs that are connected to the sternum usually have expanded and rugose distal ends (Schwarz et al., 2007). However, the ribs identified as the first dorsal ribs in Diplodocus carnegii and Apatosaurus louisae have tapering distal tips (Hatcher, 1901; Gilmore, 1936) and were mainly identified as dorsal elements due to the abrupt length increase and the differing orientation compared to the preceding, probable cervical rib (vertical rather than parallel to the vertebral centrum; Hatcher, 1901; Gilmore, 1936).

The complete set of associated ribs with the cervico-dorsal transition in specimen SMA 0011 also implies that the first dorsal rib has a tapering distal tip. Notwithstanding the gradual shape changes concerning the disappearing anterior process of the ribs, and the morphology of the articular facets of tuberculum and capitulum, the length and orientation of the ribs changes abruptly in Galeamopus pabsti SMA 0011. Here, this change in rib morphology is accompanied by a distinct shortening of the vertebral centrum, the elevation of the parapophysis to a position anterodorsal to the pleurocoel, and a more upright orientation of the neural spine. This transition is significantly different from the one in Diplodocus or Barosaurus, where the first two to three dorsal vertebrae are more similar to cervical elements (Hatcher, 1901; McIntosh, 2005). The dorsal position of the parapophysis on DV 1 is different from all other diplodocids where the transition is preserved and indicates that cervicalization of the anterior-most dorsal vertebrae was less developed in Galeamopus than in other forms.

Implications for the process of cervicalization. The shape of the transitional ribs also yield more information on the possible process of cervicalization within Diplodocidae. We consider cervicalization to represent an evolutionary process, in which an anterior dorsal vertebra loses its connection to the sternum through macroevolutionary processes, and becomes incorporated into the neck. The fact that the first dorsal rib of SMA 0011 does not have an expanded distal end indicates that its connection to the sternum was already weakened or entirely lost. The loss of the connection to the sternum was then followed by a reduction in length of the rib shaft, the change to a more horizontal orientation, the development of an anterior process, and an elongation of the vertebra.

Cervicalization also occurred in the long-necked stegosaur Miragaia longicollum, which has 17 cervical vertebrae, compared with only 12–13 in Stegosaurus (Mateus, Maidment & Christiansen, 2009). The most posterior preserved elements of Miragaia have the general aspect of stegosaur dorsal vertebrae (i.e., tall neural spines, a short centrum, and well separated capitulum and tuberculum), despite the low position of the parapophyses and short ribs typical of cervical elements. These features are not seen in Galeamopus pabsti and in other diplodocids, where cervicalization was mostly inferred to have occurred because of the number of cervical and dorsal vertebrae that differ from their sister-group Dicraeosauridae and other more distantly related sauropods like Camarasaurus, all of which have 25 presacral vertebrae in total (e.g., McIntosh, 2005).

Vertebral count. Diplodocid cervical series are generally considered to comprise 15 vertebrae (Hatcher, 1901; Gilmore, 1936; Upchurch, 1998; Wilson, 2002; Whitlock, 2011a), with the exception of Barosaurus, which was interpreted to have 16 cervical vertebrae (McIntosh, 2005). However, since only two nearly complete, and largely articulated diplodocid necks have been reported to date (Diplodocus carnegii CM 84, lacking the atlas, Hatcher, 1901; and Apatosaurus louisae CM 3018, Gilmore, 1936), this count may well have been different in other diplodocid genera. In SMA 0011, evidence suggests a maximum of 14 cervical vertebrae (based on the number of cervical ribs, and the lack of large gaps in both morphological sequence and preservation in the quarry; Fig. 3).

Diplodocid diversity in the Morrison Formation

The Morrison Formation shows the highest diversity of diplodocid sauropods worldwide, together with macronarian sauropods such as Camarasaurus and Brachiosaurus, and the diplodocoids Haplocanthosaurus and Suuwassea (Foster, 2003; Button, Rayfield & Barrett, 2014; Tschopp, Mateus & Benson, 2015). In fact, with 13 named species, Diplodocidae is the most species-rich family of vertebrates of the Morrison Biota. This diversity of megaherbivores might be surprising, but can probably be explained by a combination of extrinsic and intrinsic factors. Extrinsic factors include spatial segregation, which is supported by ecological conditions that changed with latitude throughout the extension of the Morrison Formation (Sellwood & Valdes, 2008). The high number of fragmentary specimens that do not preserve diagnostic bones precludes the identification of many remains at the species level, and thus a meaningful assessment of geographic species ranges throughout the Morrison Formation. However, there are some indications that the species Kaatedocus siberi only occurred in central to northern portions of the Morrison Formation, whereas the specimens referred to Diplodocus hallorum are restricted to more southern areas (Lucas et al., 2006; Tschopp, Mateus & Benson, 2015; Whitlock, Hanik & Trujillo, 2016). Barosaurus lentus, on the other hand, is known from both southern and north-eastern exposures (McIntosh, 2005; Tschopp, Mateus & Benson, 2015; Melstrom et al., 2016).

In addition, there is evidence for temporal segregation of diplodocid species (Tschopp, Giovanardi & Maidment, 2016), but also here, more detailed studies are needed to confirm those. Temporal resolution across the Morrison Formation is incompletely known, and long-distance correlations between quarries are only now becoming more accurate (Trujillo, 2006; Trujillo & Kowallis, 2015; Maidment, & Muxworthy, 2016). The entire duration of the deposition of the Morrison Formation, however, has been considered to represent between seven and eleven million years (Swierc & Johnson, 1996; Kowallis et al., 1998; Platt & Hasiotis, 2006), so it would seem reasonable to expect at least some temporal segregation of the species.

Finally, the open, savannah-like environment of large parts of the depositional basin of the Morrison Formation (Turner & Peterson, 2004, and references therein) might have provided favorable conditions for diplodocids. In fact, diplodocids show a high degree of specialization for grazing, as indicated by the relatively squared snout, tooth wear patterns (Whitlock, 2011b), and high tooth replacement rates (D’Emic et al., 2013). Niche partitioning must have been necessary in order to sustain such a high diversity, and has been shown to have occurred between major clades of sauropods (Barrett & Upchurch, 1994; Calvo, 1994; Fiorillo, 1998; Upchurch & Barrett, 2000), but little work has been done comparing diplodocid species until recently (Whitlock, 2011b; Button, Rayfield & Barrett, 2014). The latter studies, however, indicate that some partitioning occurred among diplodocids (Whitlock, 2011b; Button, Rayfield & Barrett, 2014), which explains at least in part the high diplodocid diversity in the Morrison Formation. Nonetheless, how this diversification exactly took place remains an open question that can only be assessed with more precise identifications and stratigraphic correlations.

Conclusions

We describe in detail a new specimen of diplodocine sauropod dinosaur, SMA 0011. Comparison with other diplodocine specimens shows that it constitutes a second species within the genus Galeamopus, which we name Galeamopus pabsti. The type specimen died at a particular ontogenetic stage, where histology indicates that it reached sexual maturity, but neurocentral fusion in cervical and dorsal vertebrae has not yet been completed. The lack of fusion between vertebral centra and neural arches can thus not be taken as definitive evidence for a juvenile ontogenetic stage. Furthermore, the specimen indicates that the number of vertebrae in the cervical column of diplodocids might have been more variable than previously assumed, and that the transition from cervical to dorsal elements was variable between genera. Although potentially surprising, the high diversity of sauropods in the Upper Jurassic Morrison Formation can be explained by a combination of extrinsic and intrinsic factors that allowed in particular a radiation of Diplodocidae. These include spatial and temporal segregation of the species and niche partitioning.

Supplemental Information

Data S1 Character list used in the phylogenetic analysis

Modified from Tschopp, Mateus & Benson (2015a))

Click here for additional data file.

Data S2 Apomorphies recovered by TNT

List of the autapomorphies and synapomorphies of the OTUs and nodes of the phylogenetic analysis.

Click here for additional data file.

Data S3 Phylogenetic matrix

TNT input file.

Click here for additional data file.

Because this paper was originally part of ET’s dissertation, important thanks go to the doctoral committee, which consisted of Octávio Mateus, Martin Sander (Bonn, Germany), João Pais, Rogerio de Rocha (both GeoBioTec), and Louis Jacobs (Dallas, USA). Additionally, and most importantly, we thank Hans-Jakob Siber and the entire team from the SMA, for the opportunity to study this crucial specimen. We particularly appreciate the work of Esther Premru for the drawing of the quarry maps, Ben Pabst for the stunning reconstruction of the holotypic skull of Galeamopus pabsti and the numerous discussions about osteology, and Martin Kistler and his team for logistic support. Initial preparation of that specimen was led by Yoli Schicker-Siber, together with Maya Siber, Esther Wolfensperger, E Tschopp, and Alicia Siber 2001 and 2002. Subsequent preparation and mounting was done by Ben Pabst, Rabea Lillich, and Ursina Bachmann in 2014 and 2015. Further thanks go to Simão Mateus (ML) for the beautiful reconstruction drawing of the skull of SMA 0011.

We thank the Willi Hennig Society for providing the phylogenetic software TNT for free online. Steve Brusatte (Univ. of Edinburgh) reviewed an earlier version of portions of this paper and provided corrections for the English. Fabien Knoll (Univ. of Manchester, UK) provided crucial comments on braincase morphology. Remo Forster (Zürich, Switzerland) and Francisco Costa (Museu da Lourinhã) helped with some figures of the specimen SMA 0011. We highly appreciated the thoughtful reviews of Phil Mannion (Imperial College London, UK), John Whitlock (Mt. Alosyus College, USA), Paul Upchurch (University College London, UK), Mike Taylor (Univ. of Bristol, UK), and the editorial comments and help of Andrew Farke (Raymond M. Alf Museum of Paleontology, USA), which greatly improved the quality of this paper.

Many collection visits for comparative purposes were possible thanks to the help and hospitality from the following people (in alphabetical order of their institutions): Kate Wellspring (AC), Carl Mehling, Mark Norell, and Alana Gishlick (AMNH), Ted Daeschler and Ned Gilmore (ANS), Amy Henrici, Matthew Lamanna, and Dan Pickering (CM), Rafael Royo-Torres and Edoardo Espilez (CPT), Daniela Schwarz (MB.R.), Virginia Tidwell and Logan Ivy (DMNS), Pete Makovicky and Bill Simpson (FMNH), David Temple (HMNS), Paul Barrett and Sandra Chapman (NHMUK), Hans-Jakob Siber and Thomas Bolliger (SMA), Ralf Kosma, Achim Ritter, and Ulrich Joger (Staatl. Naturhist. Mus. Braunschweig, Germany), Paul Sereno and Bob Masek (Univ. of Chicago, USA), Matt Carrano and Mike Brett-Surman (USNM), Bill Wahl and Malcolm Bedell (WDC), and Dan Brinkman and Marilyn Fox (YPM).

Institutional abbreviations

AC Beneski Museum of Natural History, Amherst College, Amherst, Massachusetts, USA

AMNH American Museum of Natural History, New York City, New York, USA

ANS Academy of Natural Sciences, Philadelphia, Pennsylvania, USA

CM Carnegie Museum of Natural History, Pittsburgh, Pennsylvania, USA

CPT Conjunto Paleontológico de Teruel, Dinópolis, Teruel, Spain

DMNS Denver Museum of Nature and Science, Denver, Colorado, USA

HMNS Houston Museum of Nature and Science, Houston, TX, USA

MACN Museo Argentino de Ciencias Naturales, Neuquén, Argentina

MB.R. Museum für Naturkunde, Berlin, Germany

ML Museu da Lourinhã, Lourinhã, Portugal

MPCA Museo Provincial Carlos Ameghino, Cipolletti, Río Negro, Argentina

NHMUK Natural History Museum, London, United Kingdom

NSMT National Museum if Nature and Science, Tokyo, Japan

SMA Sauriermuseum Aathal, Aathal, Switzerland

Tate Tate Geological Museum, Casper College, Casper, Wyoming, USA

USNM United States National Museum, Smithsonian Institution, Washington DC, USA

UW University of Wyoming Geological Museum, Laramie, Wyoming, USA

WDC Wyoming Dinosaur Center, Thermopolis, Wyoming, USA

YPM Yale Peabody Museum, New Haven, Connecticut, USA

Anatomical abbreviations

aaL acetabular articulation surface length

ac acetabular surface

ACDL anterior centrodiapophyseal lamina

acL acromion length

ACPL anterior centroparapophyseal lamina

acr acromial ridge

af astragalus foramen

al accessory lamina

an angular

anp antotic process

aof antorbital fenestra

aopL length antotic process

ap anterior process

apD anteroposterior depth

apf anterior pneumatic fossa

apH dorsoventral height anterior process

apL anteroposterior length

aprL anterior process length

apW anteroposterior width

ar anterior ramus

asp ascending process

at atlas

avl anteroventral lip

aW anterior width

ax axis

bc braincase

bns bifid neural spine

bo basioccipital

bpr basipterygoid process

bt basal tuber

caf capitular facet

cap capitulum

cc cnemial crest

CDF centrodiapophyseal fossa

cdH height condyle

cdW width condyle

cif crista interfenestralis

cL centrum length

cL-cd centrum length without condyle

cmW centrum minimum width

cn cranial nerve

co coracoid

cof coracoid foramen

comp compressed

CPOL centropostzygapophyseal lamina

CPRL centroprezygapophyseal lamina

CPRL-F centroprezygapophyseal lamina-fossa

CR cervical ribs

ct crista tuberalis

ctH height cotyle

ctW width cotyle

CV cervical vertebra

d dentary

dapD distal anteroposterior depth

def deformed

dg distal groove

dH distal dorsoventral height

di diapophysis

dip distal process

dist distal end

diW width across diapophyses

dlpH dorsoventral height dorsolateral process

dlr dorsolateral ridge

dmpH dorsoventral height dorsomedial process

dpc deltopectoral crest

dpcL length deltopectoral crest

DR dorsal ribs

dro distal roller

dtW distal transverse width

DV dorsal vertebra

dvH dorsoventral height

dW dorsal width

epi epipophysis

er ectopterygoid ramus

est estimated

ex exoccipital

f frontal

fe femur

fh femoral head

fi fibula

fif fibular facet

fit fibular trochanter

fl forelimb

fm foramen magnum

ft fourth trochanter

gH greatest height

gl glenoid

h humerus

hh humeral head

hl hindlimb

icg intercondylar groove

il ilium

inc incomplete

ip iliac peduncle

is ischium

isa ischial articular surface

isaL ischial articular surface length

j jugal

la lacrimal

lb lateral bulge

lprL lateral process length

lr lateral ridge

lsp lateral spur

lSPOL lateral spinopostzygapophyseal lamina

ltf laterotemporal fenestra

m maxilla

ma manus

maxD maximum diameter

maxH maximum dorsoventral height

maxL maximum length

maxW maximum transverse width

minD minimum diameter

minH minimum dorsoventral height

minW minimum transverse width

mp medial process

mr medial ridge

mt median tubercle

mtc metacarpal

mts metatarsal

n external nares

na nasal

naf neural arch foramen

naH height neural arch

nc neural canal

ncs neurocentral synostosis

nsH height neural spine

nW width notch

o orbit

oc occipital condyle

ocL lateral length contributing to orbit

ocv orbitocerebral vein foramen

of obturator foramen

os orbitosphenoid

osr otosphenoidal crest

osrL length otosphenoidal crest

p parietal

pap parapophysis

papD proximal anteroposterior depth

paof preantorbital fossa

paofe preantorbital fenestra

pas proximal articular surface

PCDL posterior centrodiapophyseal lamina

pcg pectoral girdle

PCPL posterior centroparapophyseal lamina

pdL proximodistal length

pe pes

pf prefrontal

phm manual phalanx

php pedal phalanx

pl pleurocoel

plp posterolateral process

pm premaxilla

pnf pneumatic foramina

po postorbital

POCDF postzygapophyseal centrodiapophyseal fossa

PODL postzygodiapophyseal lamina

popr paroccipital process

posl postspinal lamina

poW width across postzygapophyses

poz postzygapophysis

pp-fp distance posterior process to frontoparietal suture

ppapD pubic peduncle anteroposterior depth

ppf posterior pneumatic fossa

ppH pneumatopore height

ppL pneumatopore length

pprL length posterior process

ppW pubic peduncle transverse width

pra proatlas

prap preacetabular process

prapL preacetabular process length

PRCDF prezygapophyseal centrodiapophyseal fossa

PRDL prezygodiapophyseal lamina

pre pre-epipophysis

pro prootic

PRPL prezygoparapophyseal lamina

PRSL prespinal lamina

prW width across prezygapophyses

prz prezygapophysis

psr parasphenoid rostrum

ptf posttemporal fenestra

ptr vertical distance from proximal articular surface to trochanter

ptW proximal transverse width

pu pubis

pua pubic articular surface

puaL pubic articular surface length

pup pubic peduncle

pupL pubic peduncle length

pvf posteroventral flanges

pvfo posteroventral fossa

pvg pelvic girdle

pvl posteroventral lip

pvlp posterior ventrolateral process

pW posterior width

q quadrate

qj quadratojugal

qr quadrate ramus

r radius

sa surangular

sc scapula

scaL scapula-coracoid articular length

sD shaft diameter

SDF spinodiapophyseal fossa

so supraoccipital

SPDL spinodiapophyseal lamina

SPOF spinopostzygapophyseal fossa

SPOL spinopostzygapophyseal lamina

SPRF spinoprezygapophyseal fossa

SPRL spinoprezygapophyseal lamina

sq squamosal

sqr squamosal ramus

SR sternal ribs

stf supratemporal fenestra

SV sacral vertebrae

sW shaft width

t teeth

ti tibia

tbL length tooth-bearing portion

tc tooth crown

tif tibial facet

tpol interpostzygapophyseal lamina

tprl interprezygapophyseal lamina

tr tooth root

tub tuberculum

tuf tubercular facet

tW transverse width

u ulna

vk ventral keel

vL length ventral edge

vrH dorsoventral length ventral ramus

Other abbreviations

HOS histological ontogenetic stage

MOS morphological ontogenetic stage

Additional Information and Declarations

Competing Interests

Author Contributions

Data Availability

New Species Registration

The authors declare there are no competing interests.

Emanuel Tschopp conceived and designed the experiments, performed the experiments, analyzed the data, contributed reagents/materials/analysis tools, wrote the paper, prepared figures and/or tables, reviewed drafts of the paper.

Octávio Mateus conceived and designed the experiments, analyzed the data, contributed reagents/materials/analysis tools, prepared figures and/or tables, reviewed drafts of the paper.

The following information was supplied regarding data availability:

The raw data has been supplied as Supplementary Files.

The following information was supplied regarding the registration of a newly described species:

Publication LSID: urn:lsid:zoobank.org:pub:93B626A1-BF8E-4865-A76E-551EE78C9D92

New Species LSID: urn:lsid:zoobank.org:act:E55732A1-016C-43DA-8027-99FD5CD46E60

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
