# Peer review of "Osteology of Galeamopus pabsti sp. nov. (Sauropoda: Diplodocidae), with implications for neurocentral closure timing, and the cervico-dorsal transition in diplodocids"

_PeerJ, doi:10.7717/peerj.3179_

## Round 0.1 · original submission · Minor Revisions

The reviewers have returned very thorough, and overall positive, comments on the manuscript. This is an admirable piece of descriptive work that is certain to be of great utility for future researchers. In addition to the more minor points raised by the reviewers, I note the following larger issue to address in revision:

The updated phylogenetic analysis must be addressed in more detail in the revision. Although the position of SMA 0011 is stated to have not changed drastically in the corrected analysis, no evidence for this is presented directly in the paper. Thus, I do feel that it is important to include an updated tree--the reviewers (particularly Reviewer 1) agree. It is fine if a more detailed discussion of the results will be reserved for another paper, but nonetheless at least a preliminary tree should be included to complete the description for G. pabsti. This is also critical for the revised diagnoses presented here, and how they compare with those in Tschopp et al. 2015. Discrepancies must be addressed.

·

Basic reporting

Paper is structured soundly, very well figured, etc. etc. etc. meeting all requirements for basic reporting.

Experimental design

I believe it meets the standard, with limited exception (see attached document).

Validity of the findings

I believe the core science of the paper is valid, although I have some major and minor concerns with some of the secondary conclusions (see attached document)

Additional comments

Please see attached document for comments.

·

Basic reporting

The paper is basically clearly written and well illustrated. There are some typos, grammatical errors, and other minor issues which I have listed below.

1. Abstract - ‘Diplodocids belong the best known sauropod dinosaurs. Numerous specimens of currently…’

I think this should be ‘Diplodocids are among…’

2. Line 52 ‘…report of a diplodocine occurs in the Oxfordian (Late Jurassic) of Georgia.’

This needs a supporting reference.

3. Line 64 ‘…another genus, typified by a species previously included into Diplodocus: „D.“ hayi was found as…’

‘included into Diplodocus’ - ‘into’ should be ‘in’ or ‘within’.

4. Line 218 ‘The current study allowed to recognize two more autapomorphies of the genus: (6…

‘allowed to’ should be ‘allowed us to’

5. The authors write - ‘…of autapomorphies of G. hayi includes the following autapomorphies: (1) frontals form a pointed…’

Rephrase - ‘autapomorphies’ is said twice and makes this unnecessarily repetitive.

6. Lines 566-567 ‘…anteroventrally. The prootic bears the well-developed crista prootica, which extends…’

The crista prootica should really be called the otosphenoidal ridge - e.g. see Royo-Torres and upchurch 2012 JSP paper and references therein.

7. Lines 610-611 ‘Worn teeth usually have a single wear facet at a low angle to the long axis of the tooth, but some teeth also show two facets that are conjoined medially. In these teeth, the lingual facet is more…’

‘medially’ is not a term normally used with teeth - do you mean ‘lingually’?

8. Line 676 ‘Giraffatitan (Janensch, 1950). The spol is strongly concave, becoming vertical on the upper part.’

This description is unclear unless you specify the way in which the SPOL is concave - i.e. in which view? And are we talking about a surface or a margin?

9. Line 678 ‘A large rugose area is present on the lateral side of spine, slightly above…’

Insert ‘the’ before ‘spine’

10. Line 934 ‘…such that centra and neural arches got detached easily. The neural arch is high, with highly…’

‘got detached’ should be’ ‘could be detached’

11. Line 1095 ‘…titanosaurids (Wilson, 2002; Curry Rogers, 2005). The crest is concave laterally, but…’

The family Titanosauridae should no longer be used (see Wilson and Upchurch 2003 JSP paper). So use ‘titanosaurs’ or ‘titanosaurians’ instead.

12. Line 1292 ‘There is no indication for a large foramen opening at the center of the anterior…’

‘indication for’ should be ‘indication of’

13. Lines 1319-1320 ‘In particular the distomedial edge expands to articulate with the fibular fact in the astragalus. The distal articular surface has an oval outline.’

‘fact in’ should be ‘facet on’

14. Lines 1359-1360 medial and lateral margings. The anterior surface is less trapezoidal than in mts I’

‘margings’ should be ‘margins’

15. Lines 1379-1381 ‘in mts III. The shaft is smooth, and maintains the subtriangular shape of the proximal aricular surface. It is concave transversely on its lateroplantar surface, and does not bear any distinct…’

‘aricular’ should be ‘articular'

16. Line 1434 ‘…the same pes. The groove might be caused by taphonomy, because according the quarry map,…’

‘because according the’ should be ‘because, according to the'

17. Line 1475 ‘…posterodostal ridges for articulation with the ulna; (5) the strongly beveled distal articular surface….’

‘posterodostal’ should be ‘posterodistal’?

18. Line 1644 ‘Mateus (2013b) were right in identifying SMA 0004 as at least subadult specimen, which…’

insert ‘a’ before ‘subadult’.

19. Lines 1677-1678 (In any case, it seems that the vertebrae alone are not possible to consistently identify as either cervical or dorsal elements, a fact that is also exemplified by the…’

This needs to be rephrased. It should be something like - ‘It seems that it is not possible to identify the vertebrae alone…’

Experimental design

I have no comments to make here. The paper follows the standard practices used in descriptions of fossil vertebrates and cites a previous study for support of the phylogenetic position of Galeomopus.

Validity of the findings

The Results/Conclusions of this paper are generally valid. However, I have a few comments and suggestions for minor revisions:

1. Lines 249-250 ‘…(2) the sagittal nuchal crest on the supraoccipital is marked by a vertical midline groove; (3) mid- and posterior cervical vertebrae have two vertical, posteriorly facing, accessory…’

The authors should note that the groove on the sagittal supraoccipital crest also occurs in several titanosaurs, such as Saltasaurus, Rapetosaurus etc. (e.g. see Curry Rogers’ paper on the Rapetosaurus skull).

2. Lines 569-571 ‘…the crista prootica extends to the base of the paroccipital processes, where it separates foramina IX to XI from XII (Janensch, 1935; Harris, 2006a).’

Are you sure? - this should be the crista interfenestralis, lying below the otosphenoidal ridge - it would be odd to have these openings on either side of the otosphenoidal ridge.

3. The section on high Morrison formation diversity discusses spatial and temporal separation of sauropod faunas, and then briefly mentions a few specialisations of diplodocids. However, there is a missing element here which relates to the latter point - niche partitioning buy sauropods reflected in different tooth shapes, skull shapes, neck lengths, browsing strategies etc. This is touched upon, but only by citing two papers, both of which focus specifically on diplodocids. I would suggest adding in brief discussion of the wider literature on niche partitioning on sauropods, e.g. the work of Calvo, Fiorillo, Barrett, and Upchurch etc. from the 1990s and early 2000s.

Additional comments

This is a useful paper that I believe should be published after minor revisions.

·

Basic reporting

No Comments

Experimental design

No Comments

Validity of the findings

No Comments

Additional comments

First, please accept my apologies for the very, very late submission of this review. I have excuses, but they will be of no interest. I only hope that the long, long wait will seem worthwhile to the authors, as I have tried to do a very detailed job in this review.

Overview
* * *
This is a magnificently detailed and careful piece of descriptive work. One of the joys of online open access publications like PLOS ONE and PeerJ is that they have greatly raised the bar for morphological description in palaeontology, and ths paper is a fine example of that trend. Whereas even the most well-respected traditional journals would illustrate a new taxon like _Galeamopus pabsti_ with a few half-page black-and white composite figures showing half a dozen bones in each, this contribution provides beautiful colour plates of each element, most of them from multiple angles, not to mention helpful comparative illustrations showing how _Galeamopus pabsti_ differs from other diplodocids. The comprehensiveness of the authors' work is similarly apparent in the exhaustive tables of measurements -- the most complete response I have ever seen to Matt Wedel's exhortation "Measure your damned dinosaur" (Wedel 2009). This is how all descriptions should be in 2016.

The overall standard of the work is high, and it represents a significant advance in knowledge of the important group of Morrison-Formation diplodocids. **I recommend publication subject only to the status of the holotype** -- see below.

I do feel that the title is a little cumbersome. If this were my paper, I would remove the trailing clause "with comments on neurocentral closure timing"; but that is a decision for the authors to make.

I have annotated the manuscript with numerous picky comments. **The
authors should not feel criticised or intimidated by these.** Many of these are for the authors to take or leave as they choose -- I'm trying to be helpful. In the places where I intend to be more directive, I hope I've nade it clear in the text of the comments. The authors should not interpret the density of these comments as a criticism -- most of them refer to trivial points that are easily fixed or can simply be ignored if the authors prefer.

Specifics
* * *
With 76 figures, this is a magnificently illustrated paper. But if the authors wanted to go yet another mile, they might consider also illustrated the individual bones of the skull: premaxillae, maxilla, nasals, prefrontal-frontal-braincase complex, jugals, quadratojugals, lacrimal, quadrate, dentaries, surangular and angulars. This would be a lot of extra work, and I will fully understand if the authors prefer not to do this. But if they do, it will likely make this work the primary reference for sauropod skulls, and greatly increase citations. I leave it absolutely up to them.

The one thing that I really would like them to do is rework the figures illustrating the elements of the manus and pes, which I found confusing and difficult to interpret. In many cases, I could not determine the orientation of the depicted elements (as comments on the manuscript make clear). Most of these problems can be easily solved simply with more explicit figure captions -- or, better, by adding more information directly to the figures themselves. Illustrations should where possible directly document aspects (e.g. anterior, medial) rather than using letters to be looked up elsewhere.

Related to this, directional terminology is used inconsistently through the sections describing the manus and pes, with "plantar" and "posterior" seemingly used interchangeably. The authors should pick one set of anatomical direction and stick with them.

In general, all features described in text should be highlighted on illustrations.

Caveats
* * *
I am not well versed in cranial anatomy. I hope that one of the other reviewers is better equipped then I am to judge the part of the manuscript that deals with the skull. I've done the best I can with them (and learned a lot about sauropod skulls in the process) but I will feel more comfortable knowing that an expert has also looked at it.

The manus and pes are also outside of my primary expertise. I have done the best I can with these parts of the description, but extra attention should be paid to the comments of the other reviewers in these areas.

Status of the holotype
* * *
The new species _Galeamopus pabsti_ is based on a specimen, SMA 0011, which is owned and held by the as-yet unaccredited privately owned Sauriermuseum in Aathal, new Zurich, Switzerland. This presents a problem. The owner, Kirby Sieber, is held in high regard by the palaeontological community, and it's reasonable to expect that the specimen will remain available for study just as it would at an accredited museum. The museum is making every effort to be a good citizen, and has this statement in its website at http://www.sauriermuseum.ch/de/museum/wissenschaft/

> **Declaration Concern: Holotypes of the fossil-collection of the Sauriermuseum Aathal.**
> The Sauriermuseum Aathal, Switzerland (SMA), is being recognized more and more as valuable scientific institution. We hereby state publicly the SMA policy concerning holotype specimens. We recognize the importance of these reference specimens for science, and strongly agree that they have to be available for science in perpetuity. Therefore, we declare that all holotypes present at the Sauriermuseum Aathal, Switzerland (and all new holotypes that will be described in future), will always be publicly accessible to all bona fide researchers, and will never be allowed to be sold to any private collection.

# Confidential text redacted #

To help me clarify my thinking on this complex issue, I discussed it with Paul Barrett of the Natural History Museum, London, who is generally sympathetic to Kirby and his collection. His analysis was helpful: "One way of looking at this is that it is essentially an Open Access issue -- the data are either there for everyone (long term guarantee of access) or they’re not (private museum with no concrete plans in to the future).”

I see three possible courses of action here.

1. Consider the website statement sufficient, and proceed with publication.
2. # Confidential text redacted #
3. # Confidential text redacted #

I leave it to the handling editor, perhaps in consultation with the editor-in-chief, to make a decision on this matter. # Confidential text redacted # (For whatever it may be worth, my own inclination is to trust that the Sauriermuseum, which has until now handled is specimens very well, will continue to do so -- and so publication can proceed as in option 1 above.)

In any case, I hope (for the sake of this specimen and others at SMA) that this issue is rapidly resolved. The SMA collections contain many important specimens, and will make a major contribution to the understanding of sauropods once these rather abstruse matters are sorted out.

References
* * *
* Wedel, Mathew J. 2009. MYDD! _Sauropod Vertebra Picture of the Week_, April 23, 2009. <https://svpow.com/2009/04/23/mydd/>

---

## Round 0.2 · Minor Revisions

Thank you for your patience in awaiting a decision, and for your close attention to the comments on the previous round of reviews. I have included a number of minor changes below--once they are addressed, I should be able to come to a favorable decision quite quickly.

SPECIFIC COMMENTS:
For the title, I would use the phrasing "Osteology of Galeamopus pabsti sp. nov. (Sauropoda: Diplodocidae), with implications for neurocentral closure timing, and the cervico-dorsal transition in diplodocids"

line 36: use regular quotes around “D.” (rather than „D.“)
line 499-500: "The ventrolateral edges are only laterally indicated." -- what is meant by this? They are only preserved at their lateral extent?
line 615: "No articular, splenial, and coronoid are preserved." - rephrase to "No articular, splenial, or coronoid is preserved."
line 661: "was recovered at a distance of about 1 m (Fig. 3)" - rephrase to "was recovered at a distance of about 1 m from the next nearest vertebrae (Fig. 3)"
line 663-664: add "the" before posterior cotyle and anterior condyle -- so that it reads "Also, measurements of the posterior cotyle of CV 8 and anterior condyle of CV 9..."
line 708: remove comma after "rounded"
line 734: remove comma after "4"
line 961: "The neural arch of the dorsal vertebrae is higher" -- are you referring to a single vertebra? If so, then change "vertebrae" to "vertebra". Otherwise, reword to "The neural arches of the dorsal vertebrae are higher..."
line 1058: Remove "The" from the beginning of the sentence
line 1208: Add "so" to change wording to "but less so than in Camarasaurus..."
line 1234: Remove comma after "subrectangular"
line 1237: Add comma after "III"
line 1326: Remove comma after "outline"
line 1340: Remove "and" before "concave"
line 1416-1417: Rephrase sentence to "Metatarsals III and IV are the longest elements, and mts I and II the stoutest (Table 7)."
line 1418: Add comma after "robust"
line 1448: Rephrase to "Metatarsal IV is elongate like mts III"
line 1699: Change "rare finds" to "rareity"
line 1793: Remove comma after ")"
line 1806, 1807: Remove comma after "diplodocids" and "preserved"
line 1809: Change "Implications on" to "Implications for"
line 1812: Change "in" --> "into"
line 1818: Add tab at beginning of paragraph
line 1819: add comma after "vertebrae"
line 2226-2227: Journal title italicized here, but not elsewhere

Throughout: Write out "Table" rather than abbreviating as "Tab." or "Tab"). Similarly, write out Figure/Figures rather than abbreviating.

The supplemental files should be referenced specifically, particularly for the phylogenetic matrix. See the guidelines at: https://peerj.com/about/author-instructions/#supplemental-information

---

## Round 0.3 · accepted · Accept

Thank you for your attention to the last round of revisions!